

# Multipartite edge modes and tensor networks

**Chris Akers[1]⋆, Ronak M. Soni[2]† and Annie Y. Wei[3]‡**

**1** Institute for Advanced Study, 1 Einstein Dr, Princeton, NJ 08540 USA
**2** Department of Applied Mathematics and Theoretical Physics, University of Cambridge,
Wilberforce Road, Cambridge, CB3 0WA, United Kingdom
**3** Center for Theoretical Physics, Massachusetts Institute of Technology,
Cambridge, MA 02139, USA

⋆ cakers@ias.edu , † ronakmsoni@gmail.com , ‡ anniewei@mit.edu

## Abstract

Holographic tensor networks model AdS/CFT, but so far they have been limited by involving only systems that are very different from gravity. Unfortunately, we cannot straightforwardly discretize gravity to incorporate it, because that would break diffeomorphism invariance. In this note, we explore a resolution. In low dimensions gravity can be written as a topological gauge theory, which *can* be discretized without breaking gauge-invariance. However, new problems arise. Foremost, we now need a qualitatively new kind of "area operator," which has no relation to the number of links along the cut and is instead topological. Secondly, the inclusion of matter becomes trickier. We successfully construct a tensor network both including matter and with this new type of area. Notably, while this area is still related to the entanglement in "edge mode" degrees of freedom, the edge modes are no longer bipartite entangled pairs. Instead they are highly multipartite. Along the way, we calculate the entropy of novel subalgebras in a particular topological gauge theory. We also show that the multipartite nature of the edge modes gives rise to non-commuting area operators, a property that other tensor networks do not exhibit.

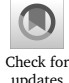

# 1 Introduction

Holographic tensor networks [1–8] are toy models of the holographic map from anti-de Sitter space (AdS) to the dual conformal field theory (CFT). See Figure 1. While imperfect models in many ways, their simplicity and concreteness have already allowed us to make rigorous statements about the emergence of spacetime [2,9], the quantum extremal surface prescription [1,3,10,11], reconstruction complexity [12], and the black hole information paradox [11].

In this note we propose a way to improve these models so that they might continue to offer insight. So far, perhaps tensor networks' biggest limitation has been their lack of time evolution. Straightforward attempts to add interesting local time evolution in the "bulk" fails to match any local time evolution of the dual "boundary" theory.[1] Long term, we would like to fix this shortcoming, adding time evolution and obtaining a completely explicit instance of holography.

In pursuit of that goal, we can ask: why have tensor networks failed to include time evolution, when the AdS/CFT duality succeeds? One glaring difference is that in gravity the diffeomorphism constraints make the physical Hamiltonian a local integral along the boundary. This leads to an easy match to a local Hamiltonian in the dual theory. Therefore, a sensible first step towards adding time evolution is to construct tensor networks that have this feature of gravity, with strong enough constraints that something similar happens, allowing us to reduce the Hamiltonian to a boundary term.

---

[1]See [13,14] for discussions of the difficulties in adding interesting time evolution. See [15,16] for one approach to a solution that does not seem to utilize gravity-like physics in the bulk.

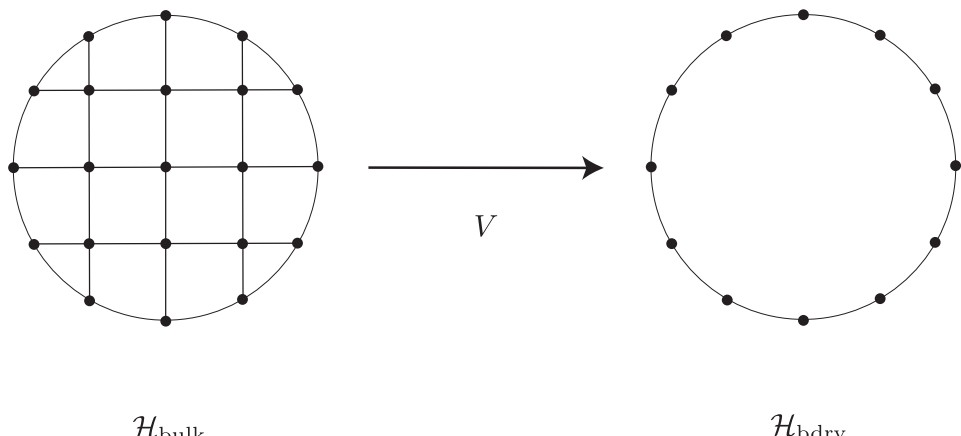

$\mathcal{H}_{\text{bulk}}$ $\qquad\qquad\qquad\qquad\qquad\qquad\qquad\qquad$ $\mathcal{H}_{\text{bdry}}$

Figure 1: An example tensor network. On the left is a graph representing the "bulk" Hilbert space $\mathcal{H}_{\text{bulk}}$, analogous to a discretized version of AdS. The links represent the geometry. On the right is the "boundary" Hilbert space $\mathcal{H}_{\text{bdry}}$ analogous to the CFT. The holographic tensor network is a linear map $V : \mathcal{H}_{\text{bulk}} \to \mathcal{H}_{\text{bdry}}$, analogous to the holographic map.

At first, however, this appears intractable. Tensor networks involve a discretization of spacetime, which inherently breaks this very diffeomorphism-invariance that we'd like to have. Nevertheless, in low enough dimensions there is a trick available to us. We can change variables and describe gravity as a certain kind of topological quantum field theory (TQFT) [17–19]. The idea is to define a gauge field as a particular combination of the vielbein and spin connection, transforming the Einstein-Hilbert action into that of an $SL(2, \mathbb{R}) \times SL(2, \mathbb{R})$ Chern-Simons theory.[2]

The advantage is that discretizing the TQFT no longer means breaking diffeomorphism-invariance. This is because the metric is not a property of the "base space" the TQFT lives on, and instead is encoded in the dynamical fields. The diffeomorphisms become "internal" gauge transformations on these fields rather than transformations of the base space itself. Hence we can try to discretize this TQFT and include it as part of the tensor network's bulk Hilbert space.[3]

We immediately run into a problem. The holographic entropy formula is different in the TQFT description, in a way that is not obviously compatible with tensor networks. Recall in AdS/CFT (in time-reversal-symmetric situations), the von Neumann entropy of a CFT subregion $B$ can be computed by [24–26]

$$S(B) = \min_b \left( \langle \hat{A}_{\eth b} \rangle + S(b) \right), \tag{1}$$

where the minimization is over AdS regions $b$ whose boundary $\eth b$ is homologous to $B$, and $\hat{A}_{\eth b}$ measures the area of $\eth b$. Traditional tensor networks satisfy a similar formula [3], where $\langle \hat{A}_{\eth b} \rangle$

---

[2]While these theories match at the level of the action, there are important known differences at the level of the path integral. For example, the natural gauge theory path integral would integrate over configurations corresponding to non-invertible metrics, which are not included in the gravitational path integral. These subtleties will not concern us, because it seems they can be addressed by using an appropriately modified TQFT [19] called the Virasoro TQFT, and our main discussion will not rely on details of any particular TQFT.

[3]Putting Chern-Simons theories on the lattice is a hard problem in general. However, pure gravity is parity-invariant. Parity-invariant Chern-Simons theories based on compact groups can be latticized as string-net models [20], which include the quantum double models we will study below. String-net models are the Hamiltonian description of Turaev-Viro models [21]. Gravity is not based on a compact group and so doesn't fall into this category; some progress for this case has been made in [22,23].

grows with the number of links cut by ð$b$. Of course, when we describe the AdS with the TQFT, the same formula (1) holds. However, in this description the area operator $\hat{A}_{ðb}$ should be understood differently! The relevant metric is now a function of the gauge fields; $\hat{A}_{ðb}$ is a certain Wilson line [27–31]. When there's no matter, the theory is topological and this Wilson line gives the same answer evaluated along any path:

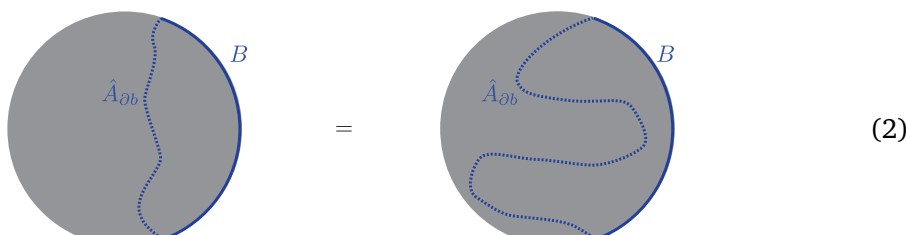

$$(2)$$

Another way to say this is that the TQFT lives on a spacetime with a metric that is irrelevant. The operator $\hat{A}_{ðb}$ is the area of a surface evaluated in the AdS metric, which is like the target space of the TQFT. This offers a challenge for tensor networks. We would not obtain an entropy formula with a property like (2) if we followed perhaps the most straightforward method to incorporate the discretized TQFT into existing tensor networks, from [4–8]. Those tensor networks lead to an entropy formula with $\langle \hat{A} \rangle$ scaling extensively with the number of links along ð$b$.[4]

The point of this note is to solve this problem with the area operator. In Section 3 we construct a tensor network with a holographic entropy formula like (1), but with an area operator that is a gauge-invariant function of the fields that encode the metric, analogous to the one in the TQFT description.

To study this problem, we will not need the full sophistication of $SL(2,\mathbb{R}) \times SL(2,\mathbb{R})$ Chern-Simons theory. Instead we will work with a toy model with the same subtlety, a much simpler topological theory that we describe in Section 2, which we call the "doubly gauged (DG) model."[5] We then define a linear map from this DG model with matter to a "boundary" Hilbert space, in Section 3. This bulk-to-boundary map (or "holographic map") is a new kind of tensor network. We explain the motivation behind the construction in Section 4.

We start with a setup as in Figure 1, like all holographic tensor network models. There are two tweaks. First, the bulk Hilbert space now includes a topological lattice gauge theory on the links. Second, the holographic map (the tensor network) is defined in a different, more topological way. The boundary Hilbert space is essentially the same as before. The result is that now boundary entropies satisfy (1) but with a different, topological $\hat{A}$. The minimization is over where to put the cut ð$b$ relative to the matter.

This new area operator leads to two striking properties of this model, which we now describe. The first striking property of our model is that its area operators do not commute, which is a desirable match to gravity [36]. In previous tensor networks, given two overlapping boundary subregions $B$ and $C$, one could generally find a bulk state such that the area operators associated to both $B$ and $C$ had arbitrarily small fluctuations. This is impossible in real AdS/CFT, because of the gravitational constraints. It is also impossible in our tensor networks, also because of the constraints.

---

[4]This sort of extensive contribution is related to the one that appears in the conventional calculations of entanglement in TQFTs [32–34], in which entropy is calculated by introducing a lattice regulator, leading to a subregion entropy with a term proportional to the area of the boundary of the subregion. We do not want to compute entropies this way, because the gravitational entropy should be independent of the way we choose to regulate the auxiliary space the TQFT lives on [27–31].

[5]Our doubly gauged models are Kitaev's quantum double models [35], but with projection onto the ground space enforced as a constraint.

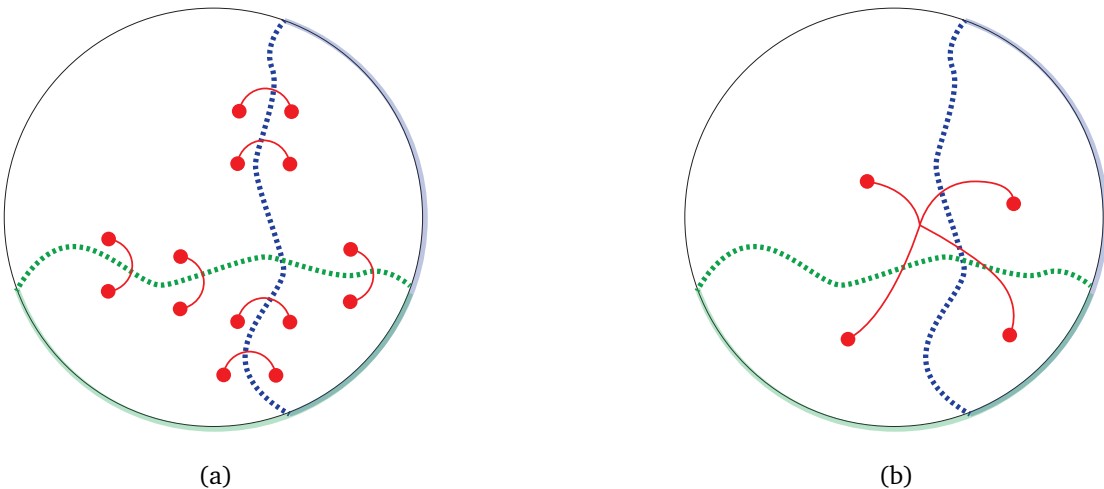

(a)          (b)

Figure 2: A difference in the entanglement structure of "edge modes" in traditional tensor networks (Figure 2a) and the tensor networks from this paper (Figure 2b). We consider two overlapping boundary regions, a blue one and a green one, and we have drawn their homologous bulk minimal surfaces as dashed lines in their respective color. In 2a, there are pairs of red dots, each pair associated to a link and representing bipartite entangled degrees of freedom. The entanglement entropy across each dashed line grows extensively with the number of links cut. Moreover, the "area operators" associated to each dotted line generally commute, because they are associated to the entanglement between different pairs of red dots. In 2b, the entanglement is no longer bipartite. Instead, the red dots are in one multipartite entangled state, and the entanglement entropy does not grow extensively across the dashed lines. Furthermore, the two area operators do not generally commute, because the allowed four party entangled states with fixed spectrum across the blue cut will not also have fixed spectrum across the green cut.

The second, related property is that this area term is the entanglement of naturally multipartite-entangled edge modes. As in all tensor networks, the "area" term in the holographic entropy formula quantifies the amount of entanglement in the "edge modes" across the cut. Historically, the edge modes in tensor networks have been local and bipartite: each link is a projected entangled *pair*. The area term simply counted the number of bipartite entangled pairs that were separated by the cut (this is why the area grew extensively with the number of cut links). See Figure 2a. In our new model, this part of the story is completely different. The degrees of freedom entangled across a cut are not in spatially localized, bipartite entangled pairs, but are in multiparty-entangled states. See Figure 2b.

These multipartite edge modes arise from a choice of factorization. Given a lattice gauge theory and a cut defining a subregion, there are many prescriptions for embedding the Hilbert space into one that factorizes across that cut, see e.g. [37–40]. The conventional choice, introduced in [41,42], leads to the insertion of a number of degrees of freedom scaling extensively with the area of the cut. However, there is one prescription that works differently, discovered by Delcamp, Dittrich, and Riello (DDR) in [43], see also [44,45]. We utilize this prescription, along the way generalizing it to new contexts.

This note is organized as follows. In Section 2 we introduce the topological gauge theory. In Section 3 we turn to tensor networks, explaining our new construction. We also explain the "commuting areas problem" and how this new construction avoids it. In Section 4 we discuss factorization of gauge theories and argue that the choices made in the construction of

the tensor network are fairly rigid. In Section 5 we explain the gravitational description of our networks (which is somewhat obscure in the gauge theory description), and connect it to other work such as [46]. In Section 6 we conclude and discuss future directions.

While this manuscript was in preparation, the work [23] appeared. They also discuss the topological description of gravity in the context of a tensor network, and find a similar non-extensive area operator. Our works agree qualitatively but explore different aspects. In particular, in this paper we have a bulk Hilbert space with matter and consider the physics of overlapping area operators. In [23], while they do not include matter, they use a more realistic TQFT. It would be interesting future work to combine these constructions.

**Notation and conventions**

A lattice $\Lambda = \{V, L, P\}$ is a set of a collection of vertices $V$, a collection of links $L$, and a collection of plaquettes. A subregion $b = \{V_b, L_b, P_b\}$ is a set such that $V_b \subseteq V$, $L_b \subseteq L$, and $P_b \subseteq P$. We will use $\eth b$ to denote the set of links connecting a vertex in $b$ to a vertex in the complement of $b$.

# 2 Doubly gauged lattice models

The goal of this note involves incorporating a topological gauge theory into a tensor network. This section introduces the topological lattice model we will use, and then discusses important properties, including its algebra of operators and insensitivity to the lattice.

## 2.1 Hilbert space

First we consider the case without matter. The model is essentially Kitaev's quantum double model [35] restricted to the ground space. Let $G$ be a finite group, $\Sigma$ an oriented 2D surface (possibly with boundary), and $\Lambda = (V, L, P)$ be an arbitrary oriented lattice on $\Sigma$, where $V, L$, and $P$ are the sets of vertices, oriented links, and plaquettes of the lattice respectively. We restrict to $\Sigma = D^2$ for this work, though we expect it to straightforwardly generalize to the cylinder $\Sigma = S^1 \times I$ as well. For every $\ell \in L$, let $\mathcal{H}_\ell := \mathcal{H}_G := L^2(G)$ be a Hilbert space associated to that edge, spanned by the basis $\{|g\rangle : g \in G\}$, which we call the group basis.[6] Note that there is another basis for $\mathcal{H}_\ell$ that will be convenient later, called the "representation basis": By the Peter-Weyl theorem (see e.g. Appendix A of [47] for an introduction), the Hilbert space decomposes as

$$\mathcal{H}_G = \bigoplus_{\mu \in \hat{G}} \mathcal{H}_\mu \otimes \mathcal{H}_{\mu^*}, \tag{3}$$

where $\hat{G}$ is the set of irreducible representations (irreps) of $G$. The representation basis is spanned by orthonormal states $|\mu, ij\rangle$ where $i, j$ index the states in $\mathcal{H}_\mu, \mathcal{H}_{\mu^*}$ respectively. The Hilbert space associated to the collection of all the links is

$$\mathcal{H}_{\text{pre}} := \bigotimes_{\ell \in L} \mathcal{H}_\ell, \tag{4}$$

which we call the "pre-gauged" Hilbert space. This $\mathcal{H}_{\text{pre}}$ has a natural basis of states of the form

$$\left| g_1, \dots, g_{|L|} \right\rangle, \tag{5}$$

which we will use often.

---

[6]It will sometimes be convenient to allow ourselves to reverse the orientation of a link while leaving the physics unchanged. In general we will refer to the reverse of the link $\ell$ as $\bar{\ell}$, and use the isomorphism between $\mathcal{H}_{\bar{\ell}}$ and $\mathcal{H}_\ell$ given by $|g\rangle_{\bar{\ell}} \cong \left| g^{-1} \right\rangle_\ell$. Note that a given set $L$ is only allowed to contain one of $\ell$ and $\bar{\ell}$.

We define the following operators. The shift operators $L_\ell(h)$ (respectively $R_\ell(h)$) act on $\mathcal{H}_\ell$ by left (right) multiplying by $h$ ($h^{-1}$), i.e.

$$L_\ell(h)\big|g_1,\ldots,g_\ell,\ldots,g_{|L|}\big\rangle = \big|g_1,\ldots,hg_\ell,\ldots,g_{|L|}\big\rangle,$$
$$R_\ell(h)\big|g_1,\ldots,g_\ell,\ldots,g_{|L|}\big\rangle = \big|g_1,\ldots,g_\ell h^{-1},\ldots,g_{|L|}\big\rangle. \tag{6}$$

These are sometimes also called the 'electric' operators.[7] The 'magnetic' operators are defined as follows. Let $\rho$ be a path through $\Lambda$, i.e. an ordered collection of vertices $\{v_1, v_2, \ldots, v_{|\rho|}\}$, each vertex connected by a link to the one before and after. Let $\ell_i$ be the link connecting $v_i$ and $v_{i+1}$. Let $W_\rho(f)$ be defined to compute the product of the group elements of the edges connecting the vertices in $\rho$ and then apply the function $f : G \to \mathbb{C}$ to the product. The prescription for computing the product is to start at the first vertex and then move along the edge connecting it to the next vertex, right multiplying by the associated group element, and inverting that group element if that edge is oriented opposite relative to the direction of travel. If we call this product $g_\rho \in G$, then we can write

$$W_\rho(f)\big|g_1,\ldots,g_{|L|}\big\rangle = f(g_\rho)\big|g_1,\ldots,g_{|L|}\big\rangle. \tag{7}$$

One useful function $f$ is the Kronecker delta $\delta_h(g)$ which equals 1 if $g = h$ and 0 otherwise.

We use these to define the operators that appear in the gauge constraints as follows. Define $A_v(g)$ to act on edges that touch $v \in V$ by $L_\ell(g)$ (or $R_\ell(g)$) if the link is oriented away (towards) $v$. Let $(v, p)$ denote the counterclockwise path around plaquette $p$ starting at vertex $v$. Define $B_{(v,p)}(h) = W_{(v,p)}(\delta_h)$ to annihilate a state where the group element around $(v,p)$ is not $h$, and to be 1 on states where it is. For example,

$$A_v(h)\left|\begin{array}{c} \end{array}\right\rangle := \left|\begin{array}{c} \end{array}\right\rangle \tag{8}$$

$$B_{(v,p)}(h)\left|\begin{array}{c} \end{array}\right\rangle := \delta_h(h_1 h_2 h_3^{-1} h_4)\left|\begin{array}{c} \end{array}\right\rangle$$

These operators can easily be shown to satisfy the algebra (known as the quantum double algebra),

$$A_v(g^{-1}) = A_v(g)^\dagger,$$
$$A_v(g)A_v(h) = A_v(gh),$$
$$B_{(v,p)}(g)B_{(v,p)}(h) = \delta_{g,h}B_{(v,p)}(h), \tag{9}$$
$$A_v(g)B_{(v,p)}(h) = B_{(v,p)}(ghg^{-1})A_v(g).$$

To define the physical Hilbert space, we will need the projectors

$$A_v := \frac{1}{|G|}\sum_{g\in G}A_v(g),$$
$$B_p := B_{(v,p)}(e), \tag{10}$$

---

[7]Note that these two shift operators are related by reversing the orientation of the link, i.e. $L_\ell(h) = R_{\bar{\ell}}(h)$.

where $v$ is any vertex adjacent to $p$ and $e \in G$ is the identity group element. (Note that when $h = e$, $B_{(v,p)}(h)$ depends only on $p$ and not on the choice of $v$.) These satisfy

$$
\begin{aligned}
A_v(g)A_v &= A_v \,, \\
B_{(v,p)}(h)B_p &= \delta(h,e)B_p \,,
\end{aligned}
\tag{11}
$$

(10) are both projectors by the following argument. By the above equation, $A_v A_v = A_v$, and by the invariance of $\sum_{g \in G}$ under $g \to g^{-1}$ we have $A_v = A_v^\dagger$. Likewise, $B_p B_p = B_p$ and manifestly $B_p = B_p^\dagger$. Using (9), one can check that for all $v \in V$ and $p \in P$, $[A_v, B_p] = 0$.

We now will use these to build projectors onto the "gauge-invariant subspace." First, for generality let there be a subset $V_{\text{bdry}} \subset V$ of vertices and $P_{\text{bdry}} \subset P$ of plaquettes that we will *not* impose constraints on. These include plaquettes and vertices at the boundary of $\Sigma$ and also any plaquettes that encircle non-trivial cycles of $\Sigma$. Let the complements of these sets be $V_{\text{bulk}}$ and $P_{\text{bulk}}$. Define the projectors onto the gauge-invariant subspace

$$
\begin{aligned}
A &:= \bigotimes_{v \in V_{\text{bulk}}} A_v \,, \\
B &:= \bigotimes_{p \in P_{\text{bulk}}} B_p \,.
\end{aligned}
\tag{12}
$$

$A$ projects onto the subspace satisfying Gauss's law at each (non-boundary) vertex, and $B$ projects onto the subspace with a trivial holonomy – i.e. flat connection – around each (non-boundary) plaquette. Define the physical, "gauged," Hilbert space

$$
\mathcal{H}_{\text{phys}} := \frac{\mathcal{H}_{\text{pre}}}{\text{Gauss} \times \text{Flatness}} := AB\mathcal{H}_{\text{pre}} \,.
\tag{13}
$$

This equation reflects a very important difference in our perspective compared to much previous work. In the lattice gauge theory literature, only the $A$-type, Gauss's law, constraints are imposed in the definition of the physical Hilbert space. Similarly, in the literature on topological phases, it is common to identify our $\mathcal{H}_{\text{pre}}$ and $\mathcal{H}_{\text{phys}}$ as the physical and ground state spaces respectively. That is natural from a condensed matter perspective, since there are no materials whose fundamental theory is topological. However, in the comparison to (the gauge theory description of) $2+1d$ general relativity, both the Gauss's law and flatness constraints are toy models for the diffeomorphism constraints, and so it is important for us that they are both used to define the physical Hilbert space.[8]

Including matter changes things as follows. Let 'site' denote a pair $(v, p)$ of a vertex and a plaquette, such that the vertex is on the bottom-left of the plaquette (this is a convention). Denote by $S$ the collection of sites. To each site we associate a Hilbert space $\mathcal{H}_{(v,p)}$, carrying a representation of the quantum double algebra (9). The pre-gauged Hilbert space is now

$$
\mathcal{H}_{\text{pre}} = \bigotimes_{\ell \in L} \mathcal{H}_\ell \bigotimes_{(v,p) \in S} \mathcal{H}_{(v,p)} \,.
\tag{14}
$$

The constraints are modified to

$$
\begin{aligned}
A_v(g) &\to A_v(g)A_{\text{mat},(v,p)}(g) \,, \\
B_{(v,p)}(g) &\to \frac{1}{|G|} \sum_{h \in G} B_{(v,p)}(gh^{-1})B_{\text{mat},(v,p)}(h) \,,
\end{aligned}
\tag{15}
$$

---

[8]Readers familiar with the Chern-Simons description of 3d gravity might find this comment a little confusing, since in that case the diffeomorphism constraints map to flatness constraints on the gauge field. Flatness constraints in continuum $G_k \times G_{-k}$ Chern-Simons theory become both types of constraints in the lattice model [20].

where the operators $A_{\text{mat}}, B_{\text{mat}}$ act on $\mathcal{H}_{(v,p)}$ and satisfy the algebra (9). The constraints are (10), with these new operators on the right hand side. We allow $A_{\text{mat}}, B_{\text{mat}}$ to be the identity operators at some sites, in which case the constraints at those sites are not modified; for simplicity we also assume that at these sites the matter Hilbert space is trivial, $\mathcal{H}_{(v,p)} = \mathbb{C}$.

Lattices that we will consider look for example like

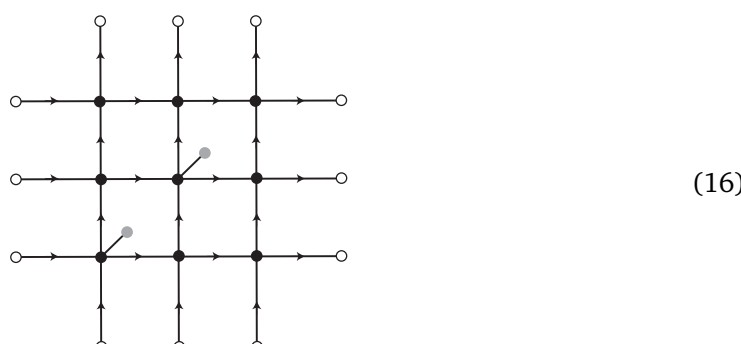

(16)

Here, black circles denote bulk vertices and white circles denote boundary vertices. Diagonal lines connected to gray circles denote which bulk sites come with matter degrees of freedom [43, 48] – the associated vertex is the one connected to the gray circle by a line, and the associated plaquette is the one that contains the gray circle. We can think of these gray circles as being where the matter lives – any site without one has no matter degree of freedom.

It will be important to note how Gauss's law manifests in the representation basis. Say we have $n$ links connected to a vertex, all oriented outwards for simplicity. Let there be matter as well. Recall that each link is spanned by a basis of the form $|\mu, ij\rangle$, as in (3), and the matter has some Hilbert space also in general decomposing as a direct sum over Hilbert spaces associated to irreps, which we might write as spanned by $|\mu, i, c\rangle$ where $i$ is a representation index (like the $i, j$ for links) and $c$ is a multiplicity index allowed for generality. The subspace invariant under the action of $A$ is the one for which all the $i$ indices "fuse together" such that the joint representation is the trivial irrep. There are only particular combinations of the $\mu$ that can fuse appropriately, and the entanglement in the $i$ indices is greatly constrained. For example, if $n = 3$ and there's no matter, a general state takes the form

$$\sum_{\substack{\mu_1\mu_2\mu_3 \\ i_1 i_2 i_3 \\ j_1 j_2 j_3}} R^{\mu_1\mu_2\mu_3}_{j_1 j_2 j_3} C^{\mu_1\mu_2\mu_3}_{i_1 i_2 i_3} |\mu_1, i_1 j_1\rangle |\mu_2, i_2 j_2\rangle |\mu_3, i_3 j_3\rangle \,, \tag{17}$$

where $R^{\mu_1\mu_2\mu_3}_{j_1 j_2 j_3}$ are free parameters, but $C^{\mu_1\mu_2\mu_3}_{i_1 i_2 i_3}$ are the Clebsch-Gordan coefficients, and are *completely fixed*, depending only on the group $G$. The fusion of more than three legs also has qualitatively similar restrictions, except there tend to be more than one way to fuse the $i$ indices given the set of $\mu$ indices.

## 2.2 Physical operators

Given $\mathcal{H}_{\text{phys}}$, what do the physical operators look like? The operators (6) are not gauge-invariant when acting on bulk links. If $|\psi\rangle \in \mathcal{H}_{\text{phys}}$, $L_\ell(h)|\psi\rangle$ violates Gauss's law at a vertex adjacent to $\ell$, since $[L_\ell(h), A_v] \neq 0$. We can construct the gauge-invariant operators as follows.

First note that a slight generalization of $L_\ell(h)$ will violate Gauss's law at a different vertex instead. Given a link $\ell$ and a path $\rho = \{v_{|\rho|}, \ldots, v_1\}$ with $\ell$ oriented away from $v_1$, let $T_{\ell,\rho}(h)$ shift the element assigned to link $\ell$ by $h$ conjugated by the product of group elements along $\rho$,

for example for $\rho = \{v_3, v_4, v_1\}$ and $\ell$ the link between $v_1$ and $v_2$,

$$T_{\ell,\rho}(h) \left| \begin{array}{c} v_4 \xrightarrow{h_3} v_3 \\ h_4 \;\rho\; h_2 \\ v_1 \xrightarrow{h_1} v_2 \end{array} \right\rangle := \left| \begin{array}{c} v_4 \xrightarrow{h_3} v_3 \\ h_4 \;\rho\; h_2 \\ v_1 \xrightarrow{(h_3^{-1}h_4)^{-1}h(h_3^{-1}h_4)h_1} v_2 \end{array} \right\rangle . \tag{18}$$

We call these *transported shift* operators. Morally, we are picking an element $h \in G$ in the frame of $v_3$ and then transporting it along $\rho$ to $v_1$. Then at $v_1$ we left multiply the element on $\ell$. We can confirm that $T_{\ell,\rho}$ fails to commute with $A_v$ only for the $v$ at the start of $\rho$. Transported versions of the $R_\ell(h)$ can also be constructed. Note the usefulness of these transported shifts: we can define a shift operator on an arbitrary edge $\ell$ that commutes with $A$ by starting $\rho$ at a vertex in $V_{\text{bdry}}$. However, while boundary anchored transported shifts commute with $A$, it is straightforward to show they do *not* commute with $B$ as long as $\ell$ borders some $p \in P_{\text{bulk}}$.

We will now define operators that commute with both, called *ribbon operators* [35, 49]. A *ribbon* is a set of two paths, one through the graph (the "spine"), the other an adjacent path through the dual graph (the links intersected by this dual graph path are called the "spokes"). We draw ribbons with an oriented dashed line along the dual graph path, and shade the space between the two paths, as below. Given a ribbon $\gamma$, we define a ribbon operator as follows. Let $g, h \in G$. The ribbon operator $F_\gamma(h, g)$ acts as

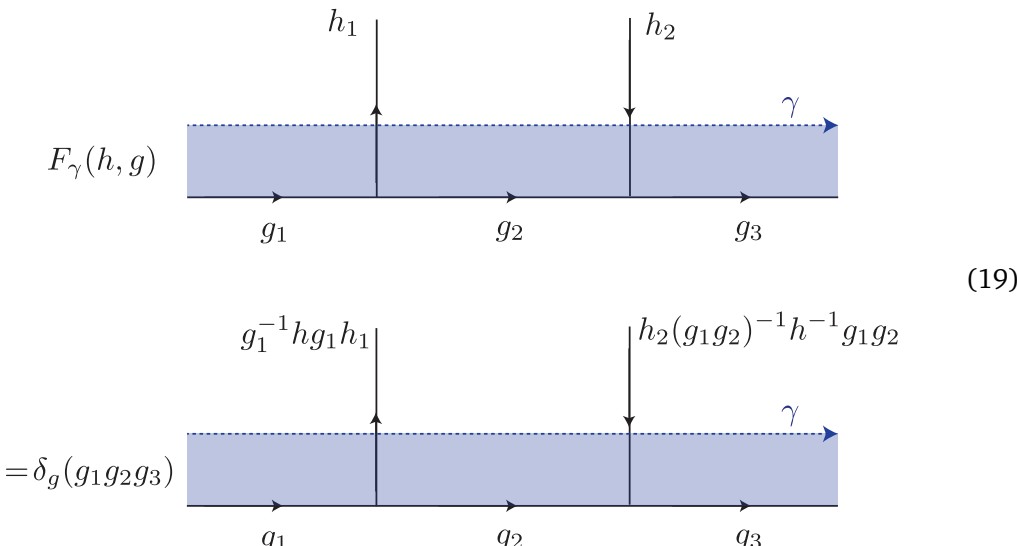

$$\tag{19}$$

One can confirm this commutes with both $A, B$ except possibly at the end points of $\gamma$. Therefore this operator is gauge invariant if its endpoints are at the boundary.

In the presence of matter degrees of freedom, a ribbon can also end on a site with matter. Any charged matter has to be dressed with the appropriate ribbon operator, either to another charge or to the boundary.

An important property of ribbon operators is that they are topological. If two ribbons $\gamma, \gamma'$ share the same end-points and $\gamma$ can be continuously deformed to $\gamma'$ without crossing any matter excitations, then

$$F_\gamma(h, g) = F_{\gamma'}(h, g) . \tag{20}$$

See [49] for a detailed proof.

## 2.3 Lattice independence

We now describe a powerful idea that we will use heavily: lattice independence. Above, we started from a lattice $\Lambda$ which defined a $\mathcal{H}_{\text{pre}}$ and then by extension a $\mathcal{H}_{\text{phys}}$. But ultimately, we only care about $\mathcal{H}_{\text{phys}}$ – the lattice and its associated pre-gauged Hilbert space are just tools helping us visualize the physical Hilbert space. This is a handy realization because many lattices lead to the same $\mathcal{H}_{\text{phys}}$! Given a physical Hilbert space, we might as well use whichever lattice makes it easiest to answer the question at hand.

We will think about lattice independence as follows. Say we start with a lattice $\Lambda_1$, defining $\mathcal{H}_{\text{pre}}^{(1)}$, projectors $A^{(1)}$ and $B^{(1)}$, and physical Hilbert space $\mathcal{H}_{\text{phys}} = A^{(1)}B^{(1)}\mathcal{H}_{\text{pre}}^{(1)}$. There are two "elementary moves" that change the lattice but leave the physical Hilbert space unchanged, see e.g. [50, 51]. That is, applying one of these elementary moves would give us a $\Lambda_2$, such that $\Lambda_2$ defines $\mathcal{H}_{\text{pre}}^{(2)}$ and projectors $A^{(2)}$ and $B^{(2)}$ with

$$A^{(2)}B^{(2)}\mathcal{H}_{\text{pre}}^{(2)} = A^{(1)}B^{(1)}\mathcal{H}_{\text{pre}}^{(1)}. \tag{21}$$

We describe the moves visually here. See Appendix A for a mathematical description.

**Move 1: Add (or remove) a vertex**  An example of this move is

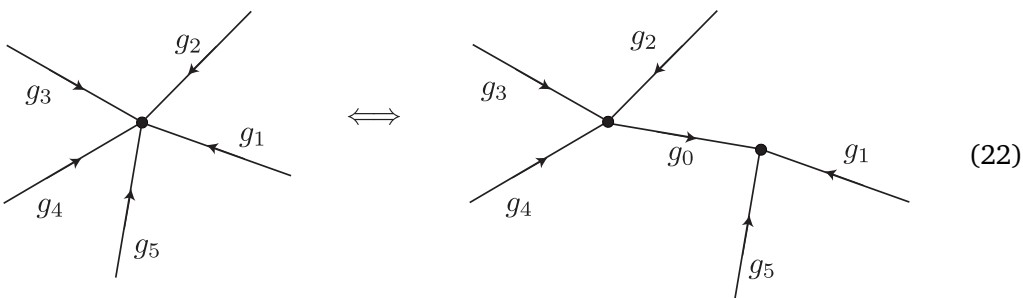

$$\tag{22}$$

Note that we can move in either direction.

This isomorphism between physical Hilbert spaces can be understood as follows. Consider a state where all five links on the left lattice are carrying fixed irreps $\mu_{1,\dots 5}$. Gauss' law requires that the five irreps on the five links fuse to the identity irrep, $\mu_1 \otimes \cdots \otimes \mu_5 \to \mathbf{1}$. The key fact is that fusion of irreps is associative. If $\mu_{2,3,4}$ fuse to $\mu_0$ and $\mu_{1,5}$ to $\mu_0'$, then Gauss' law requires that $\mu_0, \mu_0'$ also fuse to the identity. This is only possible if they are conjugate irreps, exactly like the two ends of a link, $\mu_0' = \mu_0^*$. The new link then carries the irrep $\mu_0$.

In general, the state might be in a superposition of many $\mu_0$ (or even many copies of the same irrep), but this map extends linearly. The new link carries the total electric flux propagating out of $\ell_{2,3,4}$, which can be stated mathematically as

$$R_{\ell_2}(h)R_{\ell_3}(h)R_{\ell_4}(h)L_{\ell_0}(h)\Big|_{\mathcal{H}_{\text{phys}}} = \mathbb{1}, \tag{23}$$

which is exactly the Gauss's law constraint on the right lattice. Similarly with the other end of the new link. To go the other way, we just run the above argument backwards: since fusion is associative, we don't need to separately fuse $\mu_{2,3,4}$ and $\mu_{1,5}$.

An illustrative special case of this move is to split one link into two:

$$\tag{24}$$

In the irrep basis the map takes the form

$$V_{\text{vertex}}\,|\mu;ij\rangle_1 = \frac{1}{\sqrt{d_\mu}}\sum_{k=1}^{d_\mu}|\mu;ik\rangle_0\,|\mu;kj\rangle_1. \tag{25}$$

The flux emanating out of $\ell_1$ (or, more properly, $\bar{\ell}_1$) is $\mu$, and that is what the new link carries.

**Move 2: Add (or remove a plaquette)** The move is simply

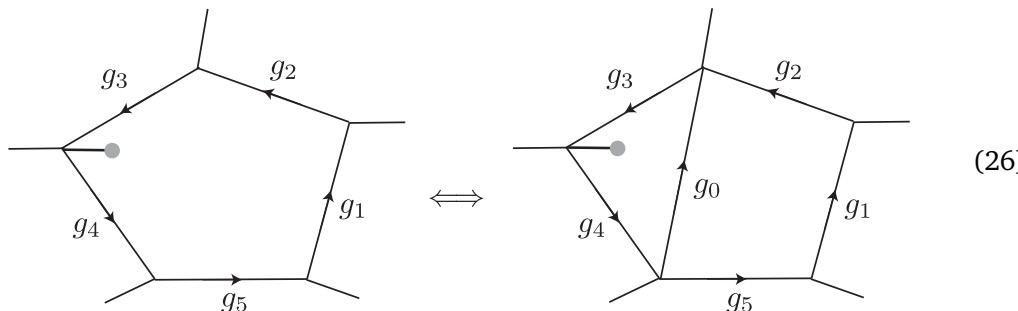

$$(26)$$

If there is a matter degree of freedom in the original plaquette, then we need to make the decision of which of the two new plaquettes it lives in.

Importantly, it does not matter that we added a link inside of a plaquette that already existed. We can take an unclosed set of links – which do not form a plaquette and therefore do not satisfy any flatness constraint – and close them by adding a new link. The new plaquette satisfies the flatness constraint regardless. The reverse operation is also important: we can take a plaquette on the edge of a lattice, then remove the outermost link, removing exactly one plaquette.

**Ribbon operator transformation**

A ribbon operator $F$ acts on $\mathcal{H}_{\text{phys}}$ and therefore must be represented on any associated lattice. We can ask: given $F_\gamma(h, g)$ acting on $\mathcal{H}_{\text{pre}}^{(1)}$, what is the associated operator on $\mathcal{H}_{\text{pre}}^{(2)}$? The answer is that it is also a ribbon operator, now including the new link if a plaquette was added along its path. For example:

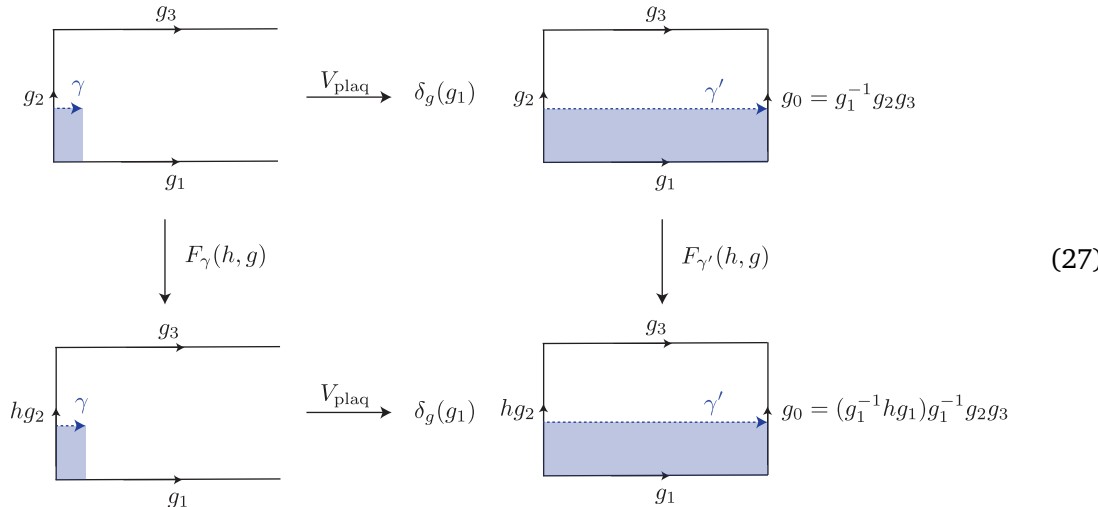

$$(27)$$

In general, the rule is as follows. The ribbon $\gamma$ is completely specified by its topological properties, i.e. its end-points, orientation, and position relative to matter degrees of freedom. The equivalent ribbon on the new lattice is simply the one that has the same properties.

## 2.4 Subalgebras and their centers

Now that we have understood the properties of the global system, we turn to subregions and subalgebras. Given a lattice $\Lambda$, a subregion $b = \{V_b, L_b, P_b\}$ is a subset of vertices, links, and plaquettes, with $V_b \subseteq V, L_b \subseteq L, P_b \subseteq P$.[9] We wish to associate to $b$ an algebra of physical operators $\mathcal{A}_b$. The feature of the subalgebra that will interest us most is the center $\mathcal{Z}_b \subseteq \mathcal{A}_b$, since that is the part associated to the area operator [10]. (The center is the subalgebra of $\mathcal{A}_b$ that commutes with all of $\mathcal{A}_b$.)

It turns out there are multiple types of subalgebras we will be interested in. In this section we will explain the simplest, most natural kind of subalgebra. In Section 2.7 and Appendix C we will explain the other types, and why we consider them. Physically, all of these subalgebras have in common that their center includes the operator that measures the net electric flux out of $b$. This is important, and means we can always find in the center an area operator with this same physical interpretation.

Given a region $b$, perhaps the most natural subalgebra to associate to it is all operators on $\mathcal{H}_{\mathrm{pre}}$ that commute with $A$ and $B$ and act trivially on the complementary set of links ($\mathcal{H}_\ell$) and matter ($\mathcal{H}_{(v,p)}$). This is the type of algebra we consider in this subsection.

We furthermore impose the following restrictions on $b$ for simplicity, in this subsection. We will specialize to lattices $\Lambda$ associated to 2D surfaces $\Sigma$ with the topology of a disk, $D^2$. These are analogous to Cauchy slices of global AdS$_3$.[10] As mentioned in the introduction, $\eth b$ is the set of links connecting $V_b$ to $V \setminus V_b$. The set $\eth b$ forms a dual path in the lattice, which intersects some plaquettes $P_{\eth b}$. We impose the restriction that $\eth b$ is topologically an interval, dividing the $D^2$ into two pieces. We also require that no plaquette in $P_{\eth b}$ contain a matter degree of freedom. It is possible to make this the case using elementary lattice moves, so there is no loss of generality, and the subsequent discussion will be simplified with this requirement.

Let $\gamma$ be a ribbon whose spokes are $\eth b$ and whose spine is the path connecting the vertices in $V_b$ adjacent to links in $\eth b$. The center is generated by the following operators that live on this ribbon:

$$F_{\eth b}([h]) := \frac{1}{|[h]|} \sum_{w \in [h]} \sum_{g \in G} F_\gamma(w, g), \tag{28}$$

where $[h] := \{w \in G : \exists g \in G \text{ s.t. } g^{-1}wg = h\}$ is the conjugacy class of $h$. A different basis will be convenient:

$$F_{\eth b}(\mu) := \frac{d_\mu}{|G|} \sum_{h \in G} \chi_\mu(h) F_{\eth b}([h]). \tag{29}$$

Here $\mu$ labels irreducible representations (irreps) of $G$, and $\chi_\mu(h)$ is the character of irrep $\mu$ and element $h$. A simple calculation shows that these are a set of orthogonal projectors,

$$F_{\eth b}(\mu) F_{\eth b}(\mu') = \delta_{\mu,\mu'} F_{\eth b}(\mu). \tag{30}$$

We prove these are central in Appendix B, along with other properties, with a straightforward argument: we write down all operators in $\mathcal{A}_b$ and then check which commute. Physically, these operators measure the *total* electric flux out of a region. $\mu$ with larger dimensions corresponds to more net flux. Intuitively, these are central because no gauge-invariant operator confined to a region can change the net flux.

---

[9]Note that we can also define a region by drawing a dual path. Just use this definition after adding new vertices wherever the dual path intersects the lattice.

[10]It would be straightforward to generalize our discussion to the case where $\Sigma$ is a cylinder, analogous to the two-sided black hole. In this setting we can consider subregions bounded by cuts $\eth b$ that are topologically $S^1$, and the center for such subregions was written down in [43]. Much like $F_\gamma(\mu)$ projects onto a sector of fixed electric flux, the central ribbon operators in this case project onto fixed irrep of the quantum double $D(G)$.

Let's convince ourselves that these operators measure the net electric flux using the lattice independence tools from Section 2.3. Consider as indicated here a subregion $b$ and the ribbon acting on ð$b$ (note $b$ includes all vertices and links that are even partially inside the circled region),

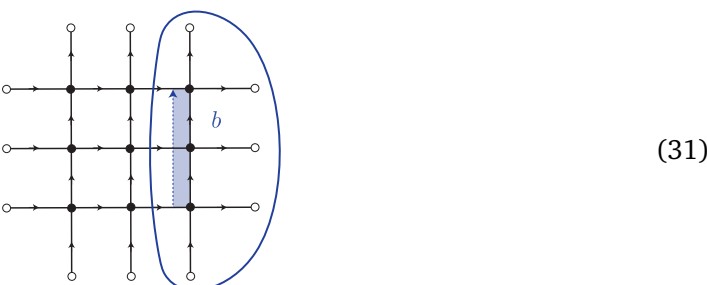

$$\tag{31}$$

Note that we did not draw the ribbon extending all the way to the boundary vertex. The rules for these central ribbon operators are that they can end on spokes; the part outside the spokes is irrelevant because it is summed over. See Proposition B.4.

As explained in Section 2.3, we change nothing by removing plaquettes along the divide (in the right way). After two applications of (26), we obtain a lattice with just one link along the path of this ribbon,

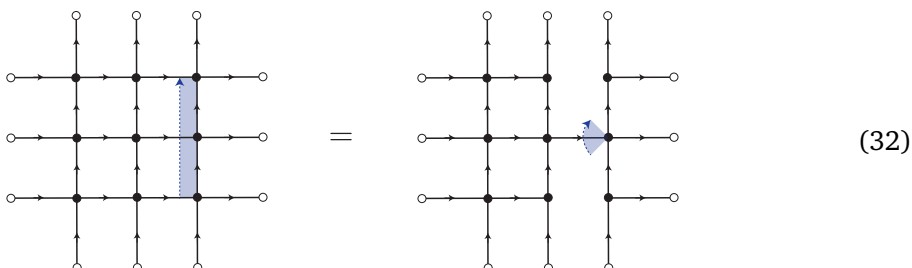

$$\tag{32}$$

Now we see: the central ribbon operator on the original lattice acts an electric operator (6) on the single link at the edge of the subregion on the new lattice. Again, nothing physical changed under each lattice manipulation. All that changed was how we represented the physical Hilbert space. Therefore the physical interpretation of these central ribbon operators is always the total electric flux, independent of which (equivalent) lattice we use.

We have explained the central operators of the simplest kind of subalgebra we might associate to a region $b$. As mentioned, we will also consider other types of subalgebras to assign to regions. These we discuss in Section 2.7 (and in more detail in Appendix C). The basic reason is that we want to associate to all $b$ an algebra in which the center includes operators measuring the total electric flux out of $b$, *but not operators measuring the flux out of individual parts*. This can make the subalgebra complicated. For example, say we are given a $b$ with two connected parts $b_1$ and $b_2$, but with $b_1$ and $b_2$ far away from each other. Say we associate to $b_1$ and $b_2$ the natural algebra described above, and furthermore say we associate to $b$ the algebraic union of these two subalgebras, $\mathcal{A}_b = \mathcal{A}_{b_1} \vee \mathcal{A}_{b_2}$. This is not what we want. In the center of $\mathcal{A}_b$ are operators measuring the net flux out of $b_1$ and $b_2$ individually. We will instead consider $\mathcal{A}_b$ with even more operators, some of which will fail to commute with the individual centers of $b_1$ and $b_2$. The only electric flux measurement in the center will be the net flux out of all of $b$.

## 2.5 Overlapping central ribbons don't commute

One important fact about the central ribbon operators is that they generally fail to commute with the central ribbon operators of other, overlapping regions. This is important for the

following reason. In future sections, the entropy we will assign to (some) $b$ will have the form[11]

$$S(b)_{|\psi\rangle} = \langle\psi|\hat{A}_b|\psi\rangle + S(b;\mathrm{alg})_{|\psi\rangle}, \qquad \hat{A}_b := \sum_\mu \log(d_\mu)F_{\delta b}(\mu). \tag{33}$$

The first term is the expectation value of a state-independent operator, and we will refer to it as the area operator, and the second term is the "algebraic von Neumann entropy" which we will define later. Two crossing area operators generally fail to commute, which we will interpret as analogous to the "non-commuting areas" property [36] in gravity. In Section 3.3 we explain this aspect of our tensor network.

We prove that suitably overlapping area operators fail to commute in Appendix B. Here we show an example. Consider this $a$ and $b$:

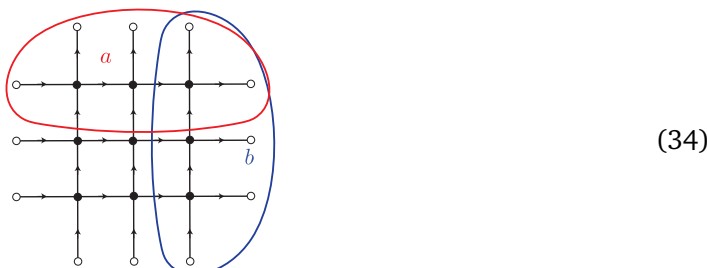

$$\tag{34}$$

Say we fix the net flux out of the $a$ region. What happens to the net flux out of $b$? Can we simultaneously fix it? For a non-abelian $G$, the answer is no. Fixing the flux out of $a$ means projecting onto a state of definite $\mu$ for the $a$ region, where $\mu$ is the label for the *joint representation* of all links in $\delta a$. In general we cannot simultaneously fix the joint representation of all links in both $\delta a$ and $\delta b$ if $a$ and $b$ are distinct but overlap.

For example, consider a simple lattice with four links connected at one vertex, with regions $a$ and $b$ each two of the links (remember, they include the entire link if it is even partially circled):

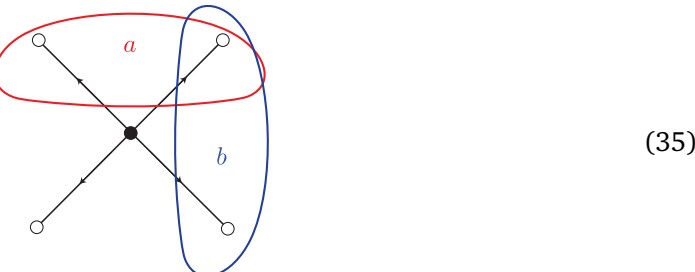

$$\tag{35}$$

Say $G = SU(2)$. Gauge-invariance tells us that all four links must fuse to the trivial irrep, but there are multiple ways to do this. Consider the case that all four links are in the spin $1/2$ representation. This is the familiar setting of four spin $1/2$ particles that together are in a singlet state. To fuse to the spin $0$ representation, the two links in $a$ could fuse to $\mu_a = 0$ or $\mu_a = 1$, and in either case the two complementary links have to do the same. But fixing $\mu_a$ either way gives a singlet state with $\mu_b$ very *not* fixed. There's no total spin $0$ state with both $\mu_a$ and $\mu_b$ fixed. The operators that measure them fail to commute.

## 2.6 Reduced lattices

We can use the lattice deformations described in Section 2.3 to make a 'minimal' lattice, which we call the reduced lattice. We describe the reduced lattice for the disk $D^2$, then argue that any lattice (embedded in $D^2$) can be deformed to it, and finally describe what the ribbon operators (and fused ribbon operators) in $\mathcal{L}(\mathcal{H}_{\mathrm{phys}})$ look like in this reduced lattice.

---

[11]More generally, the entropy will still take this form but with an operator $\hat{A}_b$ of a slightly different form. Physically, this $\hat{A}_b$ still measures the net electric flux out of $b$.

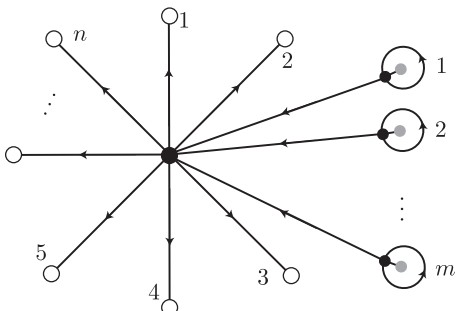

Figure 3: Reduced lattice for $D^2$. There are $n$ boundary points denoted by white circles, and $m$ lollipops. Each lollipop consists of two links connected at a vertex, with some matter living at that vertex.

The reduced lattice for $D^2$ is as follows.[12] It consists of

1. A central vertex that all boundary points are connected to by links.

2. A "lollipop" for every matter degree of freedom, also connected to the central vertex.

See Figure 3.

We change the original lattice to the reduced lattice by using the elementary moves. In particular, for any plaquette we contract all but one of the links so that the plaquette consists of one link starting and ending on the same vertex, see Figure 4. If the holonomy around the plaquette is flat, then the flatness constraint implies that the state on this link is $|e\rangle$. Gauss's law at this vertex leaves this link invariant, since $e \to heh^{-1} = e$. Thus, this link is a one-dimensional tensor factor and we can drop it. We do this for all contractible plaquettes, resulting in a new lattice where all plaquettes are inequivalent.[13] In the case when the plaquette contains a matter degree of freedom, we add a link to separate out a lollipop.[14] This gives us the reduced lattice described in Figure 3 for $D^2$.

Let us see an explicit example. Begin with

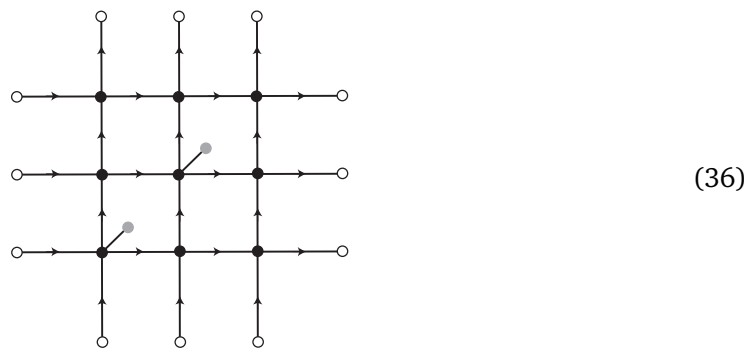

(36)

---

[12]The reduced lattice for the cylinder is similar, with one extra ingredient. There are two more links, starting as well as ending on the central vertex; all lollipops are between these two links. These two links are both representatives of the non-contractible loop of the cylinder, one for each boundary of the cylinder.

[13]Another way to arrive at the reduced lattice is via the fusion basis lattice of [43, 48]. For $D^2$, they find a tree lattice with one node for every boundary vertex and one lollipop for every plaquette with a matter degree of freedom. They show that the different assignments of irreps for links on the lattice specifies a complete basis for the physical Hilbert space. Our reduced lattice can be obtained from this tree by removing all but one bulk vertices on the 'trunk.'

[14]When constructing the reduced lattice for a more general manifold, some plaquettes may be non-contractible because it surrounds a hole in the manifold. In that case, do not remove it.

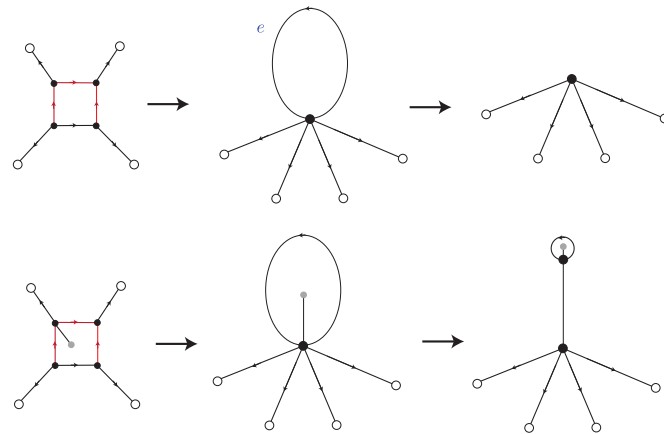

Figure 4: We contract all but one link of any plaquette, so that the plaquette consists of one link. If there is no matter inside, we can get rid of the link. If there isn't, we split it off into a 'lollipop.'

As before, black circles denote bulk vertices, white circles denote boundary vertices, and diagonal lines connected to gray circles denote which bulk sites come with matter degrees of freedom. First,

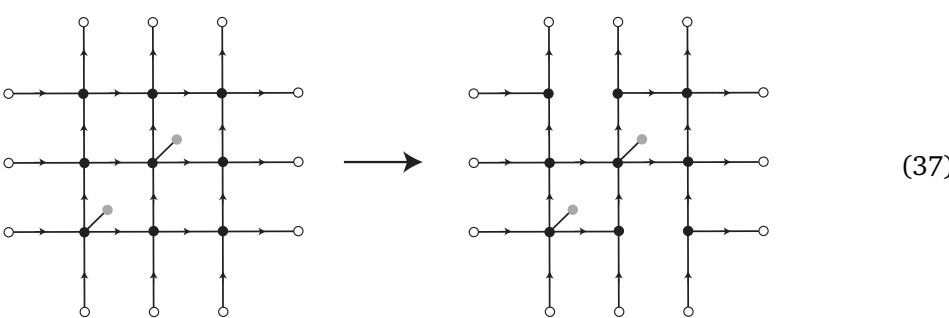 (37)

Here we have used the move (26) to remove one link from each of the plaquettes without matter. Next,

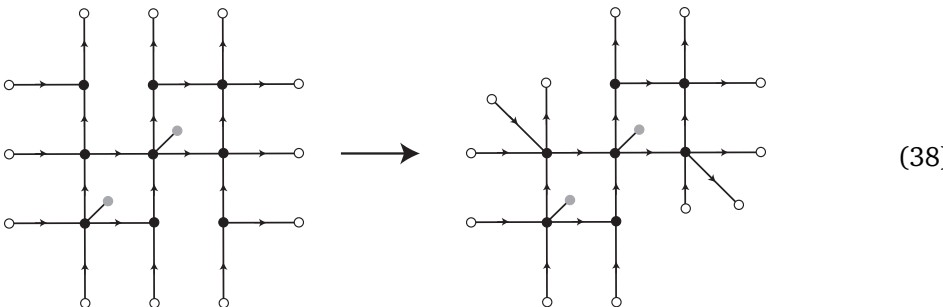 (38)

We have used (22) to remove two vertices, consolidating the graph. Next,

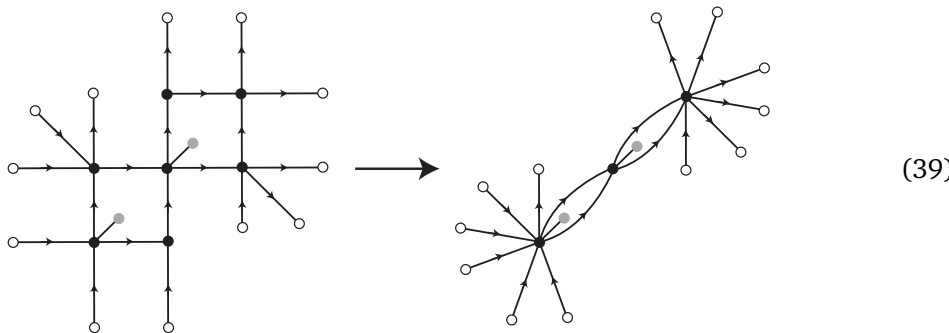

$$(39)$$

Here we have removed four more vertices with (22), two from each remaining plaquette. Next,

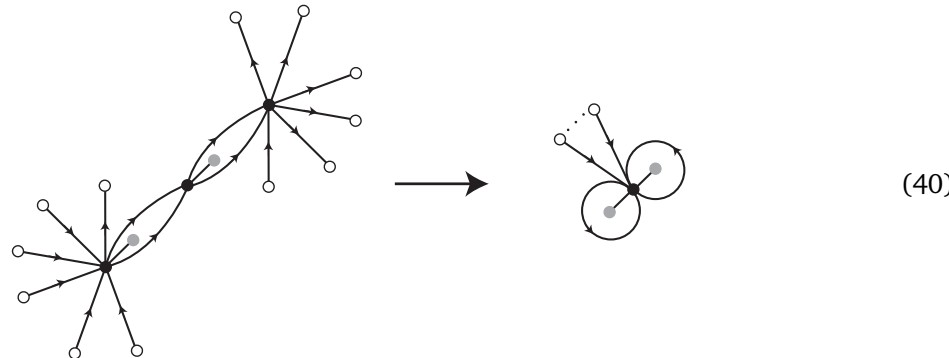

$$(40)$$

We again used (22) to remove two vertices, one from each plaquette. To reduce clutter we have suppressed 10 of the 12 boundary vertices and each of their links, indicated by "$\cdots$". Finally,

$$(41)$$

We added in two vertices using (22). This graph is now in the form (46).

## 2.7 Subalgebras revisited

We are now in a position to discuss the general kinds of subalgebras we might assign to a subregion $b$. For better or for worse, our tensor network construction will not allow us to consider only subalgebras of the simple kind from Section 2.4. Indeed, the tensor network will satisfy a holographic entropy formula like

$$S(B) = \min_b \left( \langle \psi | \hat{A}_b | \psi \rangle + S(b; \mathrm{alg}) \right), \tag{42}$$

where the minimization is over a set of bulk subregions $b$, each candidate $b$ including a different set of matter legs. What's important is that given each $b$, there is an associated subalgebra (determined by the details of the tensor network). The particular subalgebra is important, and for example affects the precise value of the algebraic entropy $S(b; \mathrm{alg})$.

The general subalgebras we'll consider are defined as follows. Say we are given a reduced lattice as in Figure 3. We pick some subset of "boundary links" (connected to white circles) and lollipops to be a subregion $\tilde{b}$. To this $\tilde{b}$, assign the natural kind of algebra from Section 2.4, which we'll call $\mathcal{A}_{\tilde{b}}$. Now, convert the reduced lattice to a more regular "full" lattice. The algebra $\mathcal{A}_{\tilde{b}}$ becomes an isomorphic algebra we'll call $\mathcal{A}_b$ acting on this full lattice. $\mathcal{A}_b$ can be associated to a subregion, which we can call $b$ – indeed it still involves operators acting on a particular set of matter legs, for example. However, it is not generally just the set of physical operators acting trivially outside $b$. We explore these algebras in more detail in Appendix C. What is important is this: the center consists of operators measuring the net electric flux out of $b$, and does *not* include operators measuring the electric flux out of subregions of $b$.

## 3 The tensor network

We are now prepared to present our main result: a tensor network with a novel, topological kind of area operator in its holographic entropy formula. This is desirable because it permits the interpretation that the lattice of the tensor network is analogous to the discretized geometry on which the TQFT description of gravity lives (which should be irrelevant to physical quantities, like the CFT entropy). One concrete advantage of this area operator is that it does not suffer from the "commuting areas problem" of other tensor networks, as we'll explain. A related noteworthy feature is that – because it is topological – this area operator's expectation value need not grow with the number of cut links, indicative of the fact that the entanglement accounted for by this area operator is not that of bipartite pairs associated to each link, a point we will discuss in detail in Section 4.

### 3.1 The model

The setup is as follows. Say we are given a system as in Section 2, with some $\mathcal{H}_{\text{phys}}$ defined on some lattice. We regard this as the bulk Hilbert space. We define the boundary Hilbert space as the set of links with one end in $V_{\text{bdry}}$, and let the tensor product of these links be the boundary Hilbert space. In other words, letting $(xy)$ denote the link connecting vertices $x$ and $y$,

$$
\begin{aligned}
\mathcal{H}_{\text{bulk}} &:= \mathcal{H}_{\text{phys}}, \\
\mathcal{H}_{\text{bdry}} &:= \bigotimes_{y \in V_{\text{bdry}}} \mathcal{H}_{(xy)}.
\end{aligned}
\tag{43}
$$

Our goal is to define a map $V : \mathcal{H}_{\text{bulk}} \to \mathcal{H}_{\text{bdry}}$. For example,

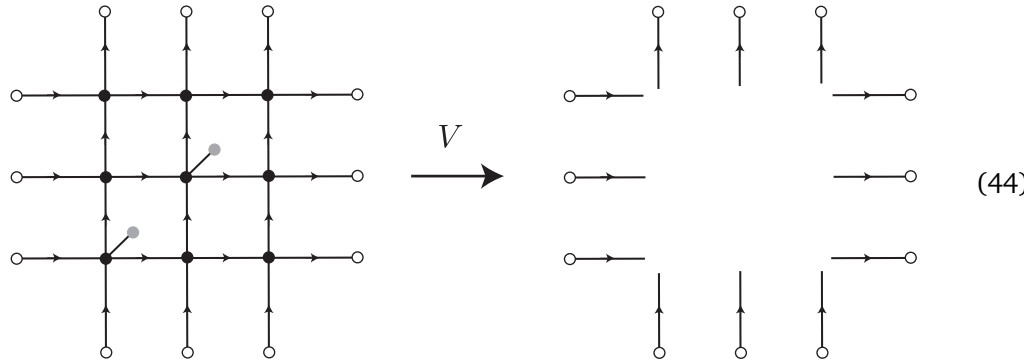

$$\tag{44}$$

The $V$ we define has three steps, which we'll list and then explain:

1. It fully reduces the lattice as in Section 2.6.

2. It (isometrically) embeds $\mathcal{H}_{\text{phys}}$ into the pre-gauged Hilbert space associated to this reduced lattice.

3. It acts random tensors $\langle T|$ on each lollipop factor.

We can draw this sequence of steps as

$$\tag{45}$$

Note these steps are schematic – for example, the true reduced lattice of the starting lattice would only have two lollipops in the next stage. We now explain the steps in detail.

First, without loss of generality we can imagine $\mathcal{H}_{\text{phys}}$ described by a fully reduced lattice, as explained in Section 2.6. This requires no physical operation on $\mathcal{H}_{\text{phys}}$; it simply requires using a particular $\mathcal{H}_{\text{pre}}$. There are in general multiple ways to reduce the lattice which do not correspond to the same $\mathcal{H}_{\text{pre}}$. However, any choice will work, and there is a finite amount of data involved in specifying which reduced lattice we wish to use and which steps we take to obtain it from the original lattice, and so we will proceed as though some choice has been made, and we have a lattice of the following form:

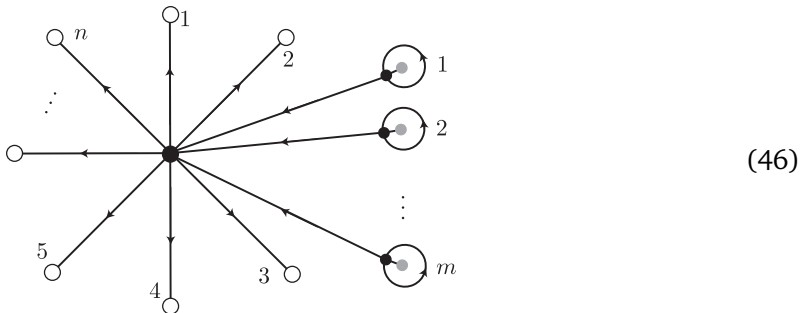

$$\tag{46}$$

Second, we embed this lattice into the pregauged Hilbert space,

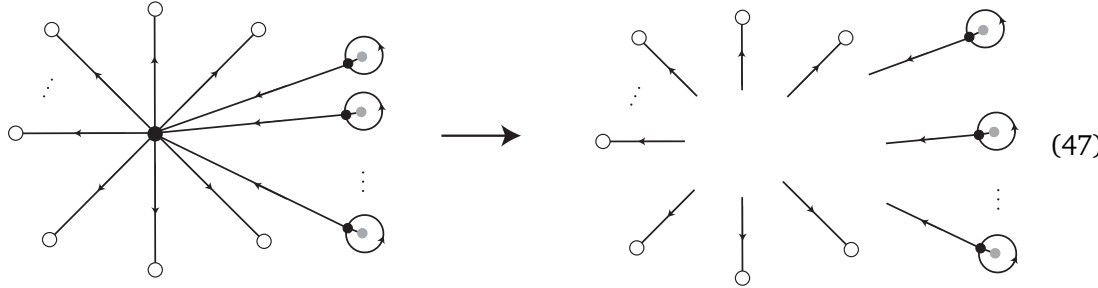

$$\tag{47}$$

(Strictly speaking, $\mathcal{H}_{\text{pre}}$ also lifts the Gauss constraint within each lollipop, so we should really not draw them still connected at their black circles. However, it will not make a difference in the later steps, and so we will continue to draw them as though they satisfy Gauss' law at their respective vertices.) Let us give a simple example to illustrate what this means. Say there is no matter, $m = 0$. In this case, that means a map $\mathcal{H}_G^{\otimes n}/\text{Gauss} \rightarrow \mathcal{H}_G^{\otimes n}$. Embedding into the pre-gauged Hilbert space now simply means that we lift the Gauss constraint – and now the

Hilbert space factorizes. For example, if $n = 2$ the bulk Hilbert space would be spanned by states $|\mu, ij\rangle$, and the embedding into the pre-gauged Hilbert space would mean the map

$$|\mu, ij\rangle \longmapsto \sum_{k=1}^{d_\mu} |\mu, ik\rangle \, |\mu, kj\rangle \, / \sqrt{d_\mu}. \tag{48}$$

Now let's reintroduce matter to the bulk Hilbert space. Then $\mathcal{H}_{\text{pre}}$ is not the same as $\mathcal{H}_{\text{bdry}}$, because it also includes the lollipop factors. We need to get rid of them, and we would like to do so in a way that is conducive to obtaining a holographic entropy formula (for example, we do not want to simply destroy the information contained in those factors). We accomplish this by acting random tensors on the extra factors. This is the third and final step of the map.[15]

We define the random tensors and their action as follows. After the embedding into $\mathcal{H}_{\text{pre}}$, we have $n+m$ factors: the $n$ "boundary" links which form a set we'll call $\mathfrak{f}_\partial$ and the $m$ lollipops which form a set we'll call $\mathfrak{f}_{\text{lol}}$. Call the set of all such factors $\mathfrak{f} = \mathfrak{f}_\partial \sqcup \mathfrak{f}_{\text{lol}}$. To act with random tensors means to act with the operator $\otimes_i \langle T_i |$ for $i \in \mathfrak{f}_{\text{lol}}$ indexing the lollipops. This eliminates the lollipop factors. These $\langle T_i |$ are each "gaussian random tensors" that we define as follows, following [52]:[16] given some fixed basis, every entry of the dual vector $\langle T_i |$ is an independent complex Gaussian random variable, i.e. can be written as $(x + iy)/\sqrt{2}$ where $x$ and $y$ are independent real Gaussian random variables of mean 0 and variance 1.[17]

## 3.2 Holographic entropy formula

Having defined our tensor network, we now argue it has a holographic entropy formula

$$S(B)_{V|\psi\rangle} = \min_b \left( \langle \psi | \hat{A}_b | \psi \rangle + S(b; \text{alg})_{|\psi\rangle} \right), \tag{50}$$

where the second term is the algebraic von Neumann entropy defined below. This is similar to traditional random tensor networks [3], but novel in three ways.

The first novelty is that the minimization over bulk regions $b$ is slightly different. We do not consider all possible cuts through the lattice homologous to $B$. Instead, each candidate $b$ is a different collection of matter legs. The minimization is really over which matter legs are included. This roughly translates to a minimization over bulk regions.

The second novelty is that given a subregion $b$, the subalgebra $\mathcal{A}_b$ we associate to it is not always the natural one described in Section 2.4. This is for reasons discussed in Sections 2.4 and 2.7 and Appendix C.

The third novelty is that the area operator $\hat{A}_b$ is quite different than in traditional tensor networks. It is no longer sensitive to the geometry of the lattice. It is now a certain physical operator in the DG model of Section 2, in the center of the algebra $\mathcal{A}_b$. In particular, it is

---

[15]As we will mention when deriving the holographic entropy formula, this only preserves the information if the original state was sufficiently nice. In particular, it needs to have a large amount of electric flux (relative to the amount of bulk entropy) from each lollipop to the boundary legs. This is like the usual random tensor network requirement that the bond dimension of in-plane legs be sufficiently large relative to the amount of bulk entropy.

[16]This is different from [3], which did not use Gaussian random tensors but instead chose tensors at random from the Haar measure. These are the same distribution up to a normalization. The Gaussian random vectors $\langle T_i |$ have norm $\| \langle T_i | \|$ that is independent of the normalized vector $\langle T_i | / \| \langle T_i | \|$, and these normalized vectors are distributed uniformly. Hence the models agree up to normalization.

[17]Explicitly, each lollipop Hilbert space is a sum over irreps $R$ of the quantum double $\mathfrak{f}_{\text{lol},i} = \oplus_R \mathcal{H}_{R,i}$. Denoting a basis as $|R, I\rangle$, we are taking $\langle T_i | R, I \rangle$ to be a Gaussian random variable. We can decompose $\langle T_i |$ into an irrep probability and a tensor in each irrep as

$$\langle T_i | := \sum_R \sqrt{p_R} \langle t_{R,i} |, \qquad \langle t_{R,i} | \in \mathcal{H}_{R,i}^*. \tag{49}$$

The prescription outlined above is equivalent to averaging over both $p_R$ as well as $\langle t_{R,i} |$ with a correlated weight.

the operator that measures the net electric flux flowing out of $b$. Therefore it is topological, only caring about its placement relative to matter degrees of freedom. Let us be more specific. When $\mathcal{A}_b$ happens to be of the simple kind described in Section 2.4, $\hat{A}_b$ is the ribbon operator

$$\hat{A}_b = \sum_\mu \log(d_\mu) F_{\delta b}(\mu), \tag{51}$$

where $F_{\delta b}(\mu)$ is the projector onto the fixed $\mu$ state defined in (29). More generally, $\hat{A}_b$ is a different kind of operator that we call a "fused ribbon operator",

$$\hat{A}_b = \sum_\mu \log(d_\mu) F_{\delta b}^{\text{fused}}(\mu), \tag{52}$$

where $F_{\delta b}^{\text{fused}}(\mu)$ is defined in Appendix C.

Let us now derive the holographic entropy formula (50). As a warmup, consider the lattice (46) with $n = 2, m = 0$. That is, two links attached at a vertex. Recall that we obtain the boundary Hilbert space simply by isometrically embedding this into the pre-gauged Hilbert space, $\mathcal{H}_G \to \mathcal{H}_G \otimes \mathcal{H}_G$. Simple as it is, this embedding of $\mathcal{H}_{\text{phys}} \to \mathcal{H}_{\text{pre}}$ already exhibits a holographic entropy formula.[18] Say we have a state $|\psi\rangle \in \mathcal{H}_{\text{phys}} \otimes \mathcal{H}_R$ for this two link Hilbert space and an arbitrary reference system $R$. We have a corresponding state $|\widetilde{\psi}\rangle \in \mathcal{H}_{\text{pre}} \otimes \mathcal{H}_R$. Say we select one of the $\mathcal{H}_G$ factors in $\mathcal{H}_{\text{pre}}$ and call it $B$, and we wish to compute the entropy of $B$ in the state $|\widetilde{\psi}\rangle$. As we know, $\mathcal{H}_{\text{phys}}$ does not factorize, instead taking the form $\mathcal{H}_G \otimes \mathcal{H}_G / \text{Gauss}$, which we can decompose as

$$\mathcal{H}_{\text{phys}} = \bigoplus_\mu \left( \mathcal{H}_{b_\mu} \otimes \mathcal{H}_{\bar{b}_\mu} \right), \tag{53}$$

where $\mu$ labels eigenvalues of the "electric" operators. Hence a general state in the bulk Hilbert space takes the form

$$|\psi\rangle = \sum_\mu \sqrt{p_\mu} |\psi_\mu\rangle, \tag{54}$$

where $\sum_\mu p_\mu = 1$ and

$$|\psi_\mu\rangle = \sum_{i,j=1}^{d_\mu} c_{ij}^\mu |\mu; ij\rangle_{b_\mu \bar{b}_\mu}, \tag{55}$$

with $\sum_{ij} |c_{ij}^\mu| = 1$. The state in the (factorizing) boundary Hilbert space $\mathcal{H}_G \otimes \mathcal{H}_G$ takes the form

$$J |\psi\rangle = \sum_\mu \sqrt{p_\mu} \sum_{i,j,k=1}^{d_\mu} \frac{c_{ij}^\mu}{\sqrt{d_\mu}} |\mu; ik\rangle_B |\mu; kj\rangle_{\bar{B}}. \tag{56}$$

By direct computation we see that the entropy of $B$ equals

$$S(B)_{J|\psi\rangle} = \sum_\mu p_\mu \log d_\mu - \sum_\mu p_\mu \log p_\mu + \sum_\mu p_\mu S(b_\mu)_{|\psi_\mu\rangle}. \tag{57}$$

We combine these last two terms into the "algebraic von Neumann entropy" $S(b; \text{alg})_{|\psi\rangle}$, for algebra $\mathcal{A}_b = \oplus_\mu \left( \mathcal{L}(\mathcal{H}_{b_\mu}) \otimes \mathbb{1}_{\bar{b}_\mu} \right)$. Then we see this takes the form

$$S(B)_{J|\psi\rangle} = \langle\psi|\hat{A}_b|\psi\rangle + S(b; \text{alg})_{|\psi\rangle}, \tag{58}$$

where

$$\hat{A}_b = \sum_\mu \log(d_\mu) F_{\delta b}(\mu). \tag{59}$$

---

[18]This is not surprising in light of [10]. This bulk to boundary map is an isometry with complementary recovery.

Here $F_{\eth b}(\mu)$ is the central ribbon projector (29) in this case acting only on the one link intersected by $\eth b$, which is the only link in $\mathcal{H}_{\text{bulk}}$.

The case with $n > 2$ links is completely analogous. The only difference is that the blocks in the decomposition (53) are now related to eigenvalues of the central ribbon operator (29). It is the total electric flux out of $B$ that matters in both cases – in the two link case that just happens to be measured by a single link operator. Therefore, for a connected region $B$ like the three links indicated here:

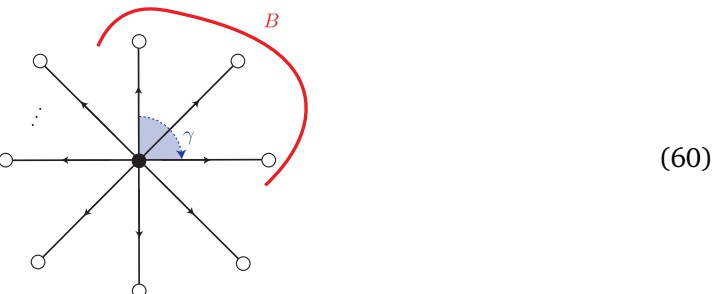

(60)

the formula becomes (58) with area operator (51), and the path $\eth b$ labelled as $\gamma$ in (60). If $B$ is disconnected, the formula is still (58) but the area operator is (52). The difference arises from the topology: when $B$ is disconnected there isn't one normal ribbon operator that acts on the links in $B$ but not its complement. Nonetheless, it is still physical to ask what is the net electric flux out of $B$, and that is what the fused ribbon operator (52) does.

Now we consider the case with matter. It will be convenient to write explicitly the division of the map $V$ into multiple parts, say $V = TJ$. The implicit first step is to map the given lattice into the reduced lattice – this does not require an explicit operator in $V$ because both lattices represent the same $\mathcal{H}_{\text{phys}}$. The second step is to act $J$, which embeds $\mathcal{H}_{\text{phys}} \to \mathcal{H}_{\text{pre}}$. The final step is $T$, which acts the random tensors.

Say we are given a state $|\psi\rangle \in \mathcal{H}_{\text{bulk}}$. Letting $\langle T| = \otimes_{i \in F_{\text{lol}}} \langle T_i|$, we can write $T = \mathbb{1}_\partial \otimes \langle T|$. The factor $\mathbb{1}_\partial$ indicates that $T$ acts trivially on $\mathfrak{f}_\partial$. The state on $\mathcal{H}_{\text{bdry}}$ is

$$V|\psi\rangle = (\mathbb{1}_\partial \otimes \langle T|)(J|\psi\rangle) = (\mathbb{1}_\partial \otimes \langle T|)|J\psi\rangle , \tag{61}$$

and we can write

$$
\begin{aligned}
V|\psi\rangle\langle\psi|V^\dagger &= (\mathbb{1}_\partial \otimes \langle T|)|J\psi\rangle\langle J\psi|(\mathbb{1}_\partial \otimes |T\rangle) \\
&= \text{tr}\left[(\mathbb{1}_\partial \otimes |T\rangle\langle T|)|J\psi\rangle\langle J\psi|\right] .
\end{aligned}
\tag{62}
$$

Now given a boundary subregion $B \subseteq \mathfrak{f}_\partial$ let's compute the $k$th Renyi entropy $S_k(B)_{V|\psi\rangle}$. This is defined as follows: given a state $|\phi\rangle \in \mathcal{H}_B \otimes \mathcal{H}_{\overline{B}}$, with $\rho := \text{tr}_{\overline{B}}|\phi\rangle\langle\phi|$ the density matrix of $B$, we have

$$S_k(B)_{|\phi\rangle} := \frac{1}{1-k}\log\frac{\text{tr}[\rho^k]}{\text{tr}[\rho]^k} , \tag{63}$$

for $k \in (0,1) \cup (1,\infty)$. We care about this because there is a way to compute it in random tensor networks using standard techniques [3, 52], and the limit $k \to 1$ is the von Neumann entropy.

Let $S_k$ be the symmetric group on $k$ elements, and let $R(\pi)$ denote the representation of $\pi \in S_k$ on $\mathcal{H}^{\otimes k}$ which acts by permuting the kets according to $\pi$. We will write $R_i(\pi)$ for $i \in \mathfrak{f}$ when it acts on ($k$ copies of) factor $i$, and also $R_B(\pi)$ when it acts on ($k$ copies of) $B$ (similarly for $\overline{B}$). Let $\tau$ denote the cyclic $k$-cycle

$$\tau = (12\ldots k) . \tag{64}$$

Now, notice

$$\text{tr}_B[\rho^k] = \text{tr}\big[(R_B(\tau) \otimes R_{\overline{B}}(e)) |\phi\rangle\langle\phi|^{\otimes k}\big]. \tag{65}$$

Furthermore, note an important property of Gaussian random tensors:

$$\mathbb{E}\big[|T_i\rangle\langle T_i|^{\otimes k}\big] = \sum_{\pi \in S_k} R_i(\pi). \tag{66}$$

Here $\mathbb{E}$ denotes the expectation value over the ensemble of tensors. Now let $|\phi\rangle = V|\psi\rangle$ and compute

$$\begin{aligned}
\mathbb{E}\,\text{tr}[\rho^k] &= \mathbb{E}\,\text{tr}\big[(R_B(\tau) \otimes R_{\overline{B}}(e)) |\phi\rangle\langle\phi|^{\otimes k}\big] \\
&= \mathbb{E}\,\text{tr}\big[(R_B(\tau) \otimes R_{\overline{B}}(e))(\mathbb{1}_\partial \otimes |T\rangle\langle T|)^{\otimes k} |J\psi\rangle\langle J\psi|^{\otimes k}\big] \\
&= \text{tr}\big[(R_B(\tau) \otimes R_{\overline{B}}(e))\mathbb{E}\big[(\mathbb{1}_\partial \otimes |T\rangle\langle T|)^{\otimes k}\big] |J\psi\rangle\langle J\psi|^{\otimes k}\big].
\end{aligned} \tag{67}$$

To simplify further, we introduce the set

$$S_{B,\sigma} := \Big\{\{\pi_i\}_{i \in F} : \pi_i \in S_k, \text{ where } \pi_i = \sigma, \text{ for } i \in B \text{ and } \pi_i = e, \text{ for } i \in \overline{B}\Big\}, \tag{68}$$

for any $\sigma \in S_k$. An element of $S_{B,\sigma}$ is an assignment of $\pi_i$ to each $i \in \mathfrak{f}$, subject to the constraint that all $i \in \mathfrak{f}_\partial$ are fixed: $i \in B$ have $\pi_i = \sigma$ and $i \in \overline{B}$ have $\pi_i = e$, the identity element. Using (66), we have

$$\mathbb{E}\,\text{tr}[\rho^k] = \sum_{\{\pi_i\} \in S_{B,\tau}} \text{tr}\Big[\bigotimes_{i \in \mathfrak{f}} R_i(\pi_i) |J\psi\rangle\langle J\psi|^{\otimes k}\Big]. \tag{69}$$

Now we will make an assumption to simplify the calculation:[19] *replica symmetry*. Under the assumption of replica symmetry, every $i \in \mathfrak{f}$ is assigned either the element $\tau$ or $e$. Let us denote by $\Delta$ the set assigned $\tau$ (which always includes $B$), and $\overline{\Delta}$ the set assigned $e$ (which always includes $\overline{B}$). Let $C(B)$ denote the set of all assignments $\Delta$. Then we can further simplify

$$\mathbb{E}\,\text{tr}[\rho^k] = \sum_{\Delta \in C(B)} \text{tr}\big[(R_\Delta(\tau) \otimes \mathbb{1}_{\overline{\Delta}}) |J\psi\rangle\langle J\psi|^{\otimes k}\big] = \sum_{\Delta \in C(B)} e^{-(1-k)S_k(\Delta)_{|J\psi\rangle}}. \tag{70}$$

We can plug this into (63) to obtain the average Renyi entropy. A typical selection of tensors $\langle T|$ will lead to an answer very close to this average [3], and so we have effectively computed the Renyi entropy for a given draw of $\langle T|$ with high probability. This is as much as we need to say about computing the Renyi entropy.

Now we turn to computing the von Neumann entropy $S = \lim_{k \to 1} S_k$. We simplify again by making a second assumption: the validity of the *saddle point approximation*,

$$\mathbb{E}\,\text{tr}[\rho^k] \overset{!}{=} e^{-(1-k)S_k(\Delta_m)_{|J\psi\rangle}}, \tag{71}$$

where $\Delta_m \subseteq \mathfrak{f}$ is some fixed set of factors, and the symbol $\overset{!}{=}$ denotes the assumption. Specifically, we assume that (70) is well-enough approximated by a single $\Delta$ (which we call $\Delta_m$) for all $k$ such that we get approximately the right von Neumann entropy by neglecting all of the others:

$$S(B)_{V|\psi\rangle} \overset{!}{=} S(\Delta)_{|J\psi\rangle} = \lim_{k \to 1} S_k(\Delta)_{|J\psi\rangle}. \tag{72}$$

---

[19]This assumption will be valid for some states $|\psi\rangle$ but not all, see e.g. [53]. In our model, nice states include those with large amounts of electric flux relative to matter entropy, and fairly simple flux patterns. We choose to make this assumption because it neglects subtleties that are not special to this model, and it allows us to more concisely demonstrate what's special about this model.

This assumption is valid for many states and choices of $B$, as in traditional random tensor networks [3], and in this note we will not attempt a full discussion of when it is valid. From now on we will drop the ! above the =, leaving the saddle point approximation implicit.

To finish computing $S(B)_{V|\psi\rangle}$, we must evaluate $S(\Delta)_{|J\psi\rangle}$ for a given configuration $\Delta$. This $\Delta$ has two parts: the degrees of freedom that are also in $\mathcal{H}_{\text{bulk}}$ which we'll call $b_1$, and the part that's introduced by the embedding into the pre-gauged Hilbert space, which we'll call $b_2$. Exactly as in (53), the bulk Hilbert space decomposes into blocks, once again with $\mu$ the eigenvalue of the ribbon operator acting on $\partial b$, associated to the total electric flux between $b$ and its complement. So, we can again write

$$|\psi\rangle = \sum_{\mu} \sqrt{p_\mu} |\psi_\mu\rangle_{b_{1,\mu}\bar{b}_{1,\mu}} . \tag{73}$$

In the embedding into the pre-gauged Hilbert space, we tack on a factor that we'll write as $|\chi_\mu\rangle \in \mathcal{H}_{b_{2,\mu}} \otimes \mathcal{H}_{\bar{b}_{2,\mu}}$, giving a state

$$|S\psi\rangle = \sum_{\mu} \sqrt{p_\mu} |\psi_\mu\rangle_{b_{1,\mu}\bar{b}_{1,\mu}} |\chi_\mu\rangle_{b_{2,\mu}\bar{b}_{2,\mu}} . \tag{74}$$

Computing $S(\Delta)_{|J\psi\rangle}$ hence gives

$$S(b)_{|\widetilde{\psi}\rangle} = \sum_{\mu} p_\mu S(b_{2,\mu})_{\chi_\mu} + \sum_{\mu} p_\mu S(b_{1,\mu})_{\psi_\mu} - \sum_{\mu} p_\mu \log p_\mu . \tag{75}$$

Recalling that $\text{tr}_{b_{2,\mu}} |\chi_\mu\rangle\langle\chi_\mu| = \mathbb{1}/d_\mu$, and that under the saddle point approximation $S(B)_{V|\psi\rangle} = S(\Delta)_{|J\psi\rangle}$ for the $\Delta$ minimizing the right hand side, we finally arrive at

$$S(B)_{V|\psi\rangle} = \langle\psi|\hat{A}|\psi\rangle + S(b; \text{alg})_{|\psi\rangle} , \tag{76}$$

where $\hat{A} = \oplus_\mu \log(d_\mu)\mathbb{1}_{b_{1,\mu}}$ and $S(b; \text{alg})_{|\psi\rangle} = \sum_\mu p_\mu S(b_{1,\mu})_{\psi_\mu} - \sum_\mu p_\mu \log p_\mu$. This completes the argument that our tensor network satisfies the holographic entropy formula (50), if we start from the reduced lattice, i.e. neglecting the first step of (45).

We now argue that the holographic entropy formula continues to hold if we start from a general lattice. The first step of (45), changing to the reduced lattice, does not change anything physical about $\mathcal{H}_{\text{bulk}}$ or the fact that the minimization in (50) is over which matter legs get included in the region $b$. All that changes is what the physical operators look like on the lattice. When $b$ is a single connected region in the reduced lattice, its algebra $\mathcal{A}_b$ is straightforward, and maps to the natural subalgebra of a single connected region in the full lattice. However, in the more general case that $b$ in the reduced lattice is disconnected, the algebra $\mathcal{A}_b$ is different. We explain the details in Appendix C. Intuitively, we have defined the algebra to include the net electric flux out of the region $b$ but not out of its sub-parts.

## 3.3 Non-commuting area operators

As pointed out in [36], traditional tensor networks fail to match the "non-commuting area operators" property of AdS/CFT. Say in AdS/CFT we consider two boundary regions $A$ and $B$ that overlap, and a state $|\psi\rangle$ of the AdS bulk. We can consider the holographic entropy formula for each:

$$\begin{aligned} S(A) &= \langle\psi|\hat{A}_a|\psi\rangle + S(a)_{|\psi\rangle} , \\ S(B) &= \langle\psi|\hat{A}_b|\psi\rangle + S(b)_{|\psi\rangle} , \end{aligned} \tag{77}$$

where $a$ and $b$ are bulk regions that minimize the respective right hand sides. It turns out that one can in general find a bulk state $|\psi\rangle$ such that $\hat{A}_a$ has very small fluctuations [54,55] (or

in which $\hat{A}_b$ has very small fluctuations). Specifically, given some $\varepsilon > 0$ we can in general find a state such that $\langle\psi|\hat{A}_a^2|\psi\rangle - |\langle\psi|\hat{A}_a|\psi\rangle|^2 < \varepsilon$.[20] This was a fortunate discovery for traditional tensor networks, because their area operators have very small fluctuations (in fact zero, for most tensor networks). One can imagine these "fixed-area" states of gravity are in this limited sense the correct AdS analog of traditional tensor networks.

However, it was pointed out in [36] that this tensor network / fixed-area state analogy only goes so far. In gravity, one can argue (using the gravitational constraint equations) that there does not exist a state $|\psi\rangle$ with very small fluctuations for the area operator of every boundary region simultaneously. In particular, the area operators of overlapping regions cannot both have small fluctuations. We can't find states with arbitrarily small fluctuations in both $\hat{A}_a$ and $\hat{A}_b$. This is different from traditional tensor networks, which can have small fluctuations across all cuts simultaneously.

Our tensor network improves this situation. In it, it is not possible to find a non-trivial bulk state that is an eigenstate of overlapping boundary subregions. This is because the area operators of overlapping boundary regions are overlapping ribbon operators (or fused ribbon operators) and will not commute, as explained in Section 2.

## 4 Multipartite edge modes

In this section, we attempt to clarify the choices made in the construction of the tensor network in Section 3. One might wonder, for example, why we chose (unconventionally) to construct the tensor network based on the reduced lattice rather than the original extended lattice. Or where is the beloved relation between geometric area and the length of the cut?

Here, we motivate our choices by studying entanglement in the DG model. As in all gauge theories, there are multiple prescriptions for defining the entanglement of a subregion. Of all these prescriptions, we are specifically interested in those that involve a factorization map, i.e. embedding the gauge theory into a larger, factorizable Hilbert space. This is because that's what tensor networks do! The boundary Hilbert space in Section 3 was the product of factors, $\mathcal{H}_G^{\otimes n}$. The holographic entropy formula computes the entropy of these factors. Therefore it is by definition computing the entropy using a factorization map – the entropy of a subregion in a factorizable Hilbert space that the DG model has been embedded into.

*Edge modes* are what we call the new degrees of freedom present in the factorized Hilbert space but not the original gauge theory Hilbert space. There are multiple known ways to define such factorization maps, each of which can be said to introduce different kinds of edge modes. However, as we'll argue, most factorization maps will fail to match certain properties we need in the holographic entropy formula. Their edge modes will have the wrong entanglement structure. In fact, we can essentially narrow down which factorizations of the DG model could give certain desired properties, down to the particular factorization map we employ in Section 3. This argument is the point of this section – with the goal of motivating the perhaps surprising choices we made in Section 3.

Let us summarize our reasons for two of the choices we made in Section 3:

1. Usually, entanglement in quantum double models is defined by a *local* factorization map on the original lattice. We do not do this.

   The reasons are twofold. Using this local factorization map, the von Neumann entropy of a region $b$ contains the two terms $(|\eth b| - 1)\log|G|$. The first term is a problem because – as we argued in Section 1 – the size of the cut in the original lattice does not have

---

[20]To be safe, one should really keep $\varepsilon$ sufficiently large relative to $\exp(-O(1/G))$, to stay within the regime of semiclassical gravity.

a relevant gravitational interpretation. The second term is *also* a problem because it makes the entanglement growth sub-extensive in a way that poorly matches AdS/CFT. We explain this in Section 4.1.

2. We define the holographic map by first deforming the original lattice to the reduced one.

    We motivate this in Section 4.2 with a tension between the non-local bipartite factorization and crossing cuts.

These do not appear as distinct steps in our construction, as the first choice is implemented by the second, but we motivate them separately. There is also a third, conventional, choice, which is that we factorized the bulk by embedding $\mathcal{H}_{\text{phys}}$ into the $\mathcal{H}_{\text{pre}}$ of the reduced lattice. This is a particular choice of edge modes on the reduced lattice. Surprisingly, it turns out that this is forced upon us by the above two choices, as we outline in Section 4.3 and prove in Appendix E.

Let us begin by giving some more careful definitions. Suppose we have many subalgebras $\mathcal{A}_{1\ldots n} \subseteq \mathcal{L}(\mathcal{H}_{\text{phys}})$ with some network of inclusion relations (which could be fairly complicated). In general, many of these subalgebras may have non-trivial centers.

To calculate the entropy of an algebra with center, say $\mathcal{A}_1$, we first calculate a reduced density matrix by embedding $\mathcal{H}_{\text{phys}} \hookrightarrow \mathcal{H}_1 \otimes \mathcal{H}_{1'}$, using a *factorization map $J_1$* [56, 57]. The entanglement entropy takes the form [10, 38].[21]

$$S(\mathcal{H}_1)_{J_1|\psi\rangle} = \langle\psi|A(J_1)|\psi\rangle + S(\mathcal{A}_1; \text{alg})_{|\psi\rangle}, \tag{78}$$

for some operator $A(J_1)$ in the center $\mathcal{A}_1 \cap \mathcal{A}_1'$.

A multipartite factorization map $J$ is an embedding of $\mathcal{H}_{\text{phys}} \hookrightarrow \mathcal{H}_{\text{fact}}$ such that *all* the algebras $J\mathcal{A}_{1\ldots n}J^\dagger$ act on tensor factors of the Hilbert space $\mathcal{H}_{\text{fact}}$. As we will see below, there can be multipartite factorization maps that are not built out of products of bipartite factorizations. When this is the case, we say that the map introduced 'multipartite edge modes.' One point of this section is to argue that if we want to incorporate a DG model into a tensor network as a toy model for gravity, then the edge modes we introduce should be multipartite.

## 4.1 Bipartite factorization

**Issue with local factorization**

Previous work on entanglement in gauge theories has introduced a factorization map [32, 33, 37–39, 41, 42], which consists of (48) for every link in $\eth b$. Let us call this the local factorization map. We now explain why this map has undesirable properties for building toy models of gravity, expanding on the discussion in [43].

Let $b$ be a subregion of $D^2$, and let there be no bulk charges. The von Neumann entropy with the local factorization map is [32–34, 59]

$$S_{\text{loc}}(b) = |\eth b| \log|G| - \log|G| + S_{\text{nonloc}}(b), \tag{79}$$

where $S_{\text{nonloc}}(b)$ is going to be the entropy in our factorization defined momentarily.[22]

Both of the first two terms are problematic for appearing in a holographic entropy formula. The extensive first term in (79) depends on the number of links in the lattice. This is the length of $\eth b$ in the discrete metric we have introduced to regulate the topological field theory.

---

[21]Alternatively, we can work abstractly in the language of generalized traces [58]. All of the literature on bipartite entanglement with centers [10, 37–39, 41, 42, 48] can be translated into this language. We have not attempted to translate the multipartite story below into this language; it is not straightforward.

[22]In the absence of matter, $S_{\text{nonloc}}$ can be thought of as the entropy of boundary degrees of freedom [34], since the central ribbon operator at $\eth b$ can be deformed to hug the boundary $\tilde{\partial} b$.

$$B_1 \quad B_2 \quad B_3 \qquad\qquad B_1 \quad B_2 \quad B_3$$

Figure 5: The different bulk regions that appear for the calculation of the tripartite information. Positive contributions on the left and negative ones on the right. The entanglement wedge and its boundaries are colour-coded. While the boundary $\partial \Sigma$ is drawn as a line for simplicity of notation, in the actual calculation, we take it to be $S^1$.

However, the area of the extremal surface is just a specific Wilson line in the Chern-Simons formulation [28, 29].[23] The central ribbon measuring the amount of electric flux flowing out of $b$ is such an operator, and its contributions to the entropy show up only in $S_{\text{nonloc}}$.

Suppose you are not convinced by this argument, taking the perspective that the whole reason tensor networks have been useful is the area law entanglement. But then you run into a second problem, which is the second term in (79). If we want to interpret the first term as $A/4G_N$, the second term is an entirely unwelcome $-1/4G_N$. Furthermore, this negative term is the famous topological entanglement entropy [32, 33], and its value is a state-independent constant that depends only on the anyon fusion algebra, so we cannot even get rid of it. There is no analog of this violation of extensivity in the HRT formula.

This leads to unwelcome behaviour not just in the von Neumann entropy, but also in other entropic quantities, as pointed out for example in [46]. Consider three contiguous boundary intervals $B_{1,2,3}$ in a 2d CFT. The tripartite information is

$$I_3(1:2:3) := S_1 + S_2 + S_3 - S_{12} - S_{23} - S_{13} + S_{123}, \tag{80}$$

where $S_1 = S(B_1)$ etc. The classic calculation of this quantity in AdS/CFT [60] shows that[24]

$$S_2 \ll S_{1,3} \quad \implies \quad I_3 \propto S_2. \tag{81}$$

Now assume that we have a tensor network with a holographic entropy formula where we minimize (79) over bulk regions of the correct homology class, using for example the construction of [7]. Assume that the entanglement wedges of the various regions are topologically the same as you would find in AdS/CFT, see Figure 5. Then, the tripartite information is

$$I_3 = -\log|G| + I_{3,\eth} + I_{3,\text{nonloc}}, \tag{82}$$

---

[23]For completeness, let us briefly describe the Wilson line introduced in these works. Denote by $\partial B := \partial \eth b = \partial \tilde{\eth} b$ the corners of $b$. $\partial B$ consists of two points on the boundary, and the Wilson line stretches between them. Then, the Wilson line is the Euclidean quantum mechanical path integral of a particle on $SL(2,\mathbb{R}) \times SL(2,\mathbb{R})$, propagating along $\eth b$. The irrep of the Wilson line is encoded in the mass and spin of the particle, and the initial and final states of the particle at the two points on $\partial B$ are defined by Ishibashi states within the highest-weight irrep the particle lives in. This particle localizes to a saddle-point in the large mass limit, and the saddle-point corresponds to a bulk geodesic; the on-shell action is proportional to the length of the geodesic.

Our central ribbons are a little different from the Wilson lines, in that they project onto certain values of the irrep flowing through $\eth b$. Furthermore, our central ribbons should be valued not in highest-weight irreps but in principal series irreps [27, 30, 31]. However, both our central ribbon and their Wilson line measure the same quantity; in the presence of matter, the HRT formula with both constructions become quantum minimal surface formulas as in Section 3. So, the construction of [28, 29] is enough to show that area is an operator in the algebra of the topological field theory, though the precise operator might be harder to pin down at the full quantum level.

[24]If the three regions have lengths $l_1, l_2, l_3$, the tripartite information in the vacuum is

$$I_3 = \frac{c}{3} \log \frac{l_1 l_2 l_3 (l_1 + l_2 + l_3)}{(l_1 + l_2)(l_2 + l_3) l_2 (l_1 + l_2 + l_3)} \xrightarrow{l_2 \to 0} -\frac{c}{3}\left(\frac{1}{l_1} + \frac{1}{l_3}\right) l_2.$$

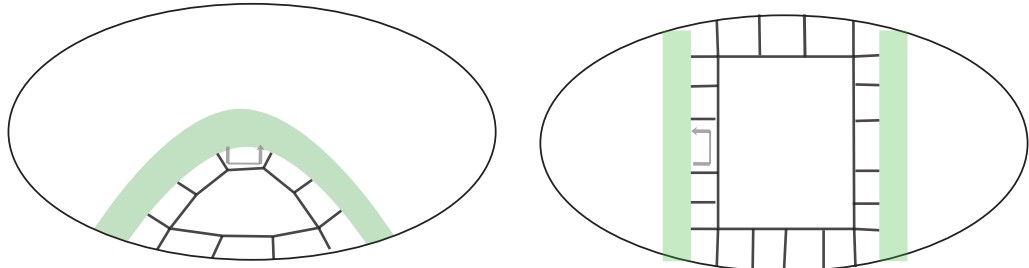

Figure 6: A lattice that makes the calculation of entanglement of the region bounded by the thick green bars and the boundary easy. We have used lattice deformations to make the bulk of the region a single plaquette. The flatness constraint is imposed for this plaquette, since it is in the bulk. We decompose the state in terms of the 'half-loop' holonomies, shown as grey arrows. Left: the subregion intersects the physical boundary once. Right: it intersects the physical boundary twice, but that doesn't change the argument.

where $I_{3,\eth}$ is the contribution of the extensive term (which behaves similarly to the area term in gravity) and the last term is the same combination of $S_{\text{nonloc}}$. Both of these last two terms satisfy (81).[25] However, the first term does not, so (82) overall fails to satisfy (81).

To prove (82), we can use (79) for the regions $B_{1,2,3}, B_{12}, B_{23}, B_{123}$, since all of their entanglement wedges have the same topology. While the formula was not originally proven for the topology of $b_{13}$, which is a strip connecting $B_1$ to $B_3$, it remains true by the following argument, following [33, 38]. Let us recall the derivation of the form (79), say for $b_2$. We deform the lattice as in the left of Figure 6, so that there is one plaquette in the bulk; the controlled unitaries from Appendix A that implement this deformation act only within $b_2$. The Hilbert space is labelled in terms of the group element around the half-loops, labelled as grey arrows. Each half-loop at $\eth b_2$ is completely unconstrained in the density matrix, except that the product of all of them is $e$, by flatness of the plaquette. Going to a sector of fixed holonomies at $\partial \Sigma$, the reduced density matrix is maximally mixed over a $|G|^{|\eth b_2|-1}$-dimensional subspace, labelled by half-loop configurations satisfying the single constraint. The last term in (79) is the entropy of these boundary holonomies. This argument only relies on the bulk region having topology $D^2$, so it goes over to the strip of $b_{13}$.

**A non-local factorization map**

We define a new factorization map, following [43], that is non-local on $\eth b$. Let us describe it in some detail for a connected subregion $b$ on $D^2$. Use the elementary lattice moves in Section 2.3 to make $\eth b$ cut a single link as in Figure 7. In that case, the central ribbon $F(\mu)$ is just a projector onto that link being in the irrep $\mu$. And then the factorization map is simply

$$J |\mu, ij\rangle = \frac{1}{\sqrt{d_\mu}} \sum_k |\mu, ik\rangle |\mu, kj\rangle \,, \tag{83}$$

on this link. This is the usual (local) factorization map of lattice gauge theory [41, 42] for the single link; but the lattice moves we did first make it a different way to factorize the original lattice.

Another way to think about this is in terms of the original lattice. In that case, the edge modes we introduce are collective modes that live on the entire cut rather than local degrees

---

[25]The second satisfies (81) for the same reason as the holographic entropy. For the third term, note that as we shrink $B_2$ the volume of $b_2$, and therefore its maximum entropy shrinks also.

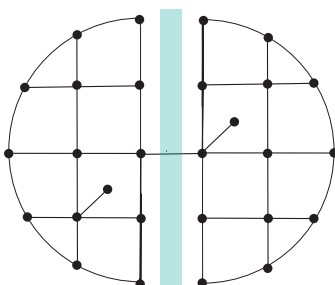

Figure 7: We deform a lattice so that $\eth b$ is a single link.

of freedom at different points on the cut, in sharp contrast to the local factorization map. The lattice deformation is helpful because it allows us to give a local description of this non-local edge mode.

We show in Appendix D that this factorization map, unlike the local one, has the property (81) (assuming that the tripartite information is negative).

## 4.2 Bipartite factorizations can fail to commute

Another issue with bipartite factorization maps is that they generally don't combine uniquely into a multipartite factorization map.

Let $\mathcal{A}$ be an algebra acting on a finite dimensional Hilbert space $\mathcal{H}$. Assume $\mathcal{A}$ has a non-trivial center, i.e. $\mathcal{A} \cap \mathcal{A}'$ contains more than multiples of the identity operator on $\mathcal{H}$. Any operator in this center can be written as a linear combination of a set of commuting projectors $P_\alpha$ (see e.g. [10]). They must commute, since the center is a commutative algebra.

Let $J : \mathcal{H} \to \mathcal{H}_B \otimes \mathcal{H}_{\overline{B}}$ be a factorization map with respect to $\mathcal{A}$, i.e. an isometry such that $\mathcal{A}$ acts on $\mathcal{H}_B$ and its commutant $\mathcal{A}'$ acts on $\mathcal{H}_{\overline{B}}$.[26] Such a factorization map can be written in the following way. For $|\psi\rangle \in \mathcal{H}$,

$$J|\psi\rangle = \sum_\alpha P_\alpha |\psi\rangle_{B_p^\alpha \overline{B}_p^\alpha} \otimes |\chi_\alpha\rangle_{B_f^\alpha \overline{B}_f^\alpha}, \tag{84}$$

where we have used a decomposition $\mathcal{H}_B = \oplus_\alpha (\mathcal{H}_{B_p^\alpha} \otimes \mathcal{H}_{B_f^\alpha}) \oplus \mathcal{H}_{B_r}$, and similarly for $\mathcal{H}_{\overline{B}}$. The central projectors $P_\alpha$ act on $\mathcal{H}_{B_1} \otimes \mathcal{H}_{\overline{B}_1}$. What's important here is that the central projectors $P_\alpha$ enter crucially into the definition of the factorization map $J$.

Now say we have two algebras $\mathcal{A}_1$ and $\mathcal{A}_2$ both acting on $\mathcal{H}$, with central projectors $P_{\alpha_1}$ and $P_{\alpha_2}$ respectively. Let

$$\begin{aligned} J_1 &: \mathcal{H} \to \mathcal{H}_B \otimes \mathcal{H}_{\overline{B}}, \\ J_2 &: \mathcal{H} \to \mathcal{H}_C \otimes \mathcal{H}_{\overline{C}}, \end{aligned} \tag{85}$$

be factorization maps, with respect to $\mathcal{A}_1$ and $\mathcal{A}_2$ respectively. We are interested in acting both factorizations on $\mathcal{H}$. This can work as follows. Say we first act $J_1$. How does $J_2$ act on $\mathcal{H}_B \otimes \mathcal{H}_{\overline{B}}$? Because $J_1$ was an isometry, we can consider the operator $J_2$ pulled through $J_1$. More explicitly, the action of $J_2$ on $J_1 \mathcal{H}$ is defined using the projectors $J_1 P_{\alpha_2} J_1^\dagger$. This allows us to define $J_2 J_1 \mathcal{H}$, which embeds $\mathcal{H}$ into a Hilbert space with more than two factors. However, because $[P_{\alpha_1}, P_{\alpha_2}] \neq 0$, in general $J_2 J_1 \mathcal{H} \neq J_1 J_2 \mathcal{H}$! We cannot in general construct a unique multipartite factorization map by the product of all bipartite factorizations. Factorizing in a different order leads to a different final Hilbert space.

---

[26]In the language of [10], $J$ defines an operator-algebra quantum erasure code with complementary recovery, with respect to $\mathcal{A}$.

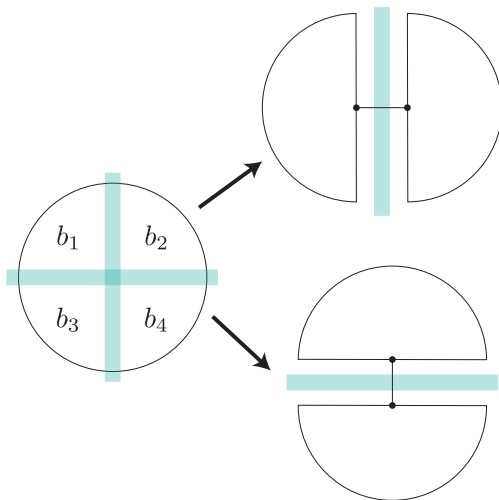

Figure 8: The lattice deformations that we need to factorize intersecting regions are incompatible.

Let us see this concretely in the DG model. Split the disk into four regions $b_{1...4}$ as in Figure 8. We are interested in the subregions $b_1 b_2$ and $b_2 b_4$. The non-commutativity of the bipartite factorization maps for these two subregions can be seen in the fact that the elementary lattice moves required to achieve each bipartite factorization are different. For example, if we change to a lattice with only one link along $\eth(b_1 b_2)$, then we have two semicircles that separately need to factorize to split $b_2$ from $b_1$ and $b_4$ from $b_3$, which would create *two* total links along $\eth(b_2 b_4)$. We cannot draw a lattice where both $\eth(b_1 b_2)$ and $\eth(b_2 b_4)$ consist of only one link.

Though it might seem cartoonish, this problem is the central one. To make it more precise, take a state where there are fixed irreps $\mu_{1...4}, \mu_{12}$ flowing out of $b_{1...4}, b_1 b_2$. Furthermore, take each of $b_1, \ldots b_4$ to consist of a single link, so we have four links total. We can decompose this state into a superposition of states with fixed irrep $\mu_{23}$ flowing out of $b_2 b_3$ using the $F$-matrices [20][27]

$$\left| \begin{matrix} \mu_4 & & \mu_3 \\ & \mu & \\ \mu_1 & & \mu_2 \end{matrix} \right\rangle = \sum_\nu F^{\mu_1 \mu_2 \mu}_{\mu_3 \mu_4 \nu} \left| \begin{matrix} \mu_4 & & \mu_3 \\ & \nu & \\ \mu_1 & & \mu_2 \end{matrix} \right\rangle . \tag{86}$$

For the quantum double model, the $F$-matrix is the $6j$-symbol of $G$; more generally, it is part of the definition of the tensor category that defines the topological phase [61]. For a non-Abelian theory, the right hand side generically has many non-zero terms. Thus, we cannot simultaneously fix the irreps flowing out of $b_1 b_2$ and $b_2 b_3$. This is a manifestation of the fact that the corresponding central ribbons don't commute, as we argued in Section 2.4.

We might try to deal with this by introducing one set of edge modes for every segment $\eth b_i \cup \eth b_{i+1}$ of the subregion boundaries. For example, in Figure 9, we could factorize all the links crossing the blue lines. But, as detailed in [44], factorizing more than one link cutting $\eth b_1$, even if it is just two links, gives rise to the negative contribution in (79). The entropy will again run afoul of (81).

This is an important obstacle to constructing a tensor network with non-commuting area operators. A tensor network acts on a bulk Hilbert space that does not factorize, and maps it to a boundary Hilbert space which is simultaneously factorized across all partitions.

---

[27]Our convention for the $F$-matrices do not exactly match those in [20]. This will not affect any of the following discussion. The only important thing is that (86) is a unitary change of basis.

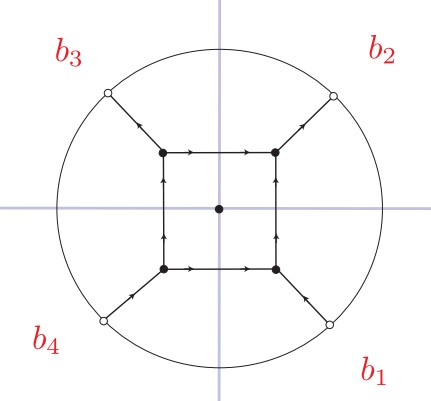

Figure 9: In local factorization maps, we can introduce edge modes for every segment of the boundary of a region.

It is implementing some multipartite factorization map – but which one? As we have argued, it cannot be some product of bipartite factorizations. It must be something more sophisticated and inherently multipartite, introducing "multipartite edge modes."[28]

We accomplish this in the tensor network of Section 3 by embedding $\mathcal{H}_{\text{phys}}$ into the $\mathcal{H}_{\text{pre}}$ of the reduced lattice. This different kind of factorization is the main idea in this work that allowed us to define tensor networks with the desired properties.

On the reduced lattice, (86) is modified in a simple way. Instead of being a relation between different lattices, it is now a relationship between different bases for $\mathcal{H}_{\text{phys}}$. The relation remains true in $\mathcal{H}_{\text{pre}}$, because the physical state is given by the fusion of the four irreps via Clebsch-Gordan coefficients. Taking the inner product of (86) with $\langle \mu_1, i_1, j_1; \dots \mu_4, i_4, j_4 | J$ makes it a relation between Clebsch-Gordan coefficients and $6j$-symbols of the group that is known to be true.

### 4.3 Bootstrapping the multipartite edge modes

The above discussion explains why our TN is constructed using the reduced lattice. Now we ask: can we further justify the factorization map we use on the reduced lattice? This is not possible for bipartite edge modes, as noted in [40]. It turns out that in the multipartite case the factorization map is much more constrained. In Appendix E, we prove that the factorization map is unique given certain assumptions. The assumptions are that the edge modes introduced by the factorization map depend only on the total irrep flowing out of the region, and that the edge mode state for the identity irrep is factorized. Let us give an overview of the logic here.

The basic idea is that we can take all possible equations of the form (86) for an arbitrary number of boundary vertices and apply the factorization map. Each of these equations becomes an equation for the matrix elements of the factorization map $J$, and the only solutions to this whole set of equations is the CG coefficients. This is related to what is known as Tannaka-Krein duality, which says that a group can be reconstructed from the fusion rules and $F$-matrices of its irreps. A helpful review is [65].

We prove a weaker statement than either of the above, but it does say that the required edge modes are (up to local unitaries) those we use in our factorization map. We take reduced lattices with $2n$ boundary links, all in the irrep $\mu$, such that they fuse to the identity in pairs.

---

[28]The connection between non-commuting modular Hamiltonians and multipartite entanglement was also explored in [62–64].

This state can be written in two ways

$$\left| \begin{matrix} \mu \underline{\quad 1 \quad} \mu \\ \mu \underline{\quad 2 \quad} \mu \\ \vdots \\ \mu \underline{\quad n \quad} \mu \end{matrix} \right\rangle = \sum_{\nu} \sqrt{\frac{d_\nu}{d_\mu^n}} \sum_{a=1}^{\left[\mathbb{N}_\mu^n\right]_\mu^\nu} \left| \begin{matrix} \mu \\ \mu \\ \vdots \\ \mu \end{matrix} \nu, a \begin{matrix} \mu \\ \mu \\ \vdots \\ \mu \end{matrix} \right\rangle . \tag{87}$$

Then, calculating the entropy in two ways, we find consistency only when the edge modes are maximally mixed with rank $d_\mu$.

This concludes our motivation for our tensor network construction. We had to transform to the reduced lattice because otherwise we would either run afoul of holographic tripartite information or not know how to uniquely factorize overlapping regions. The factorization map on the reduced lattice had to be the one we used because the multipartite edge modes in the DG model are highly constrained.

## 5 Gravity interpretation

Unlike traditional tensor networks, our tensor networks need *not* be a tiling of hyperbolic space. Instead, the connectivity in our tensor networks is analogous to the geometry on which a TQFT lives; that is, it's not fundamentally important. An advantage is what we've argued in this paper: edge modes with more gravity-like properties, leading for example to non-commuting area operators. The disadvantage is that the physical, gravitational interpretation of the state is less clear. In this (largely qualitative) section, we aim to clarify this interpretation.

We use the fact that 3d general relativity is genuinely a topological theory, albeit one that is not included in the set of DG models. However, we expect that qualitative aspects of our results do generalize (with appropriate refinements). The difficulties with overlapping bipartite factorization that we encountered in Section 4.2 are a consequence of non-trivial $F$-matrices, which exist also in other topological theories and also GR [22, 23]. There is an interesting similarity between the algebraic structure of our tensor network and that introduced in [46], as we will argue below. Finally, we view the uniqueness of edge modes formalized in theorem E.2, which holds for a large class of topological phases (though not GR), as a toy model for the reason that low-energy gravity 'knows' about the UV entropy but not its microstates. Thus, while the rest of this section has not yet been made precise, we expect that it is possible to do so.

### Comparison with conventional RTNs

The first major difference between our tensor networks and traditional ones is that in the traditional ones each link is a fixed segment of a fixed curve, and its bond dimension is interpreted as the area of the segment. Even when the area is a non-trivial operator it is a sum of areas of segments with fluctuating bond dimension. In our tensor networks, however, the area of the entire quantum minimal surface is the expectation value of a single, non-local, topological operator. As justification for this claim, we point to the works [23, 27–31], which showed that the topological Wilson lines and irrep data are related (in the semi-classical limit) to the area of quantum extremal surfaces and not to arbitrary surfaces.[29] There is nothing that corresponds to the area of a fixed segment of a QES, or the area of a non-extremal surface.

---

[29] [28, 29] showed that their Wilson lines localised to geodesics, but their Wilson lines are different from our area operator, see footnote 23. [23, 27, 30, 31] showed that factorizing across cuts of specific topological classes (roughly the same as those we have considered) gave entropy equal to the area of the QES in the same topological class. There is less work in the presence of matter; in two dimensions, [66] established the relationship between the irrep flowing across the cut and the area of the QES of the same topological class. We will also discuss an important subtlety in this statement due to matter around Figure 12.

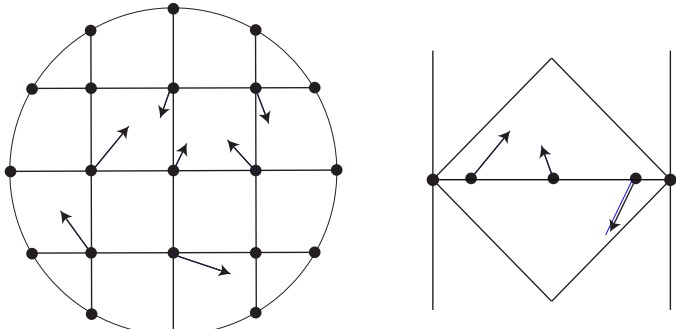

Figure 10: If we imagine the lattice as being embedded in AdS, then the diffeomorphism constraints (left) and Hamiltonian constraints (right) move the lattice around. Since the constraints in the TQFT are semi-classically these two types of constraints, we argue that our TN is a toy model for a superposition of embeddings of the lattice.

The area being a topological operator means that it takes the same value on any two topologically equivalent cuts. This can seem confusing, however, since multiple topologically equivalent cuts on the same lattice then correspond to the same geodesic in the bulk. The apparent puzzle can be resolved by remembering that the gauge field in the CS description is the metric, meaning that the group element on a link (which has non-zero fluctuations) is related to a length.[30] Secondly, any physical state has non-zero fluctuations of the group element, due to Gauss's law. Thus, a state in the topological theory on the lattice describes a *superposition* of embeddings of the lattice into the spacetime.

An important caveat with this last statement, and also the rest of this section, is that we do not have a precise gravitational interpretation of the fully quantum topological theory. So, all of these statements are true only in the semi-classical limit of the TQFT, where the coupling constant goes to zero and we restrict attention to coherent states.

This superposition presumably includes all ways of embedding the lattice into a Wheeler-deWitt patch. The reason to believe this is that the gauge constraints of the Chern-Simons action are (on-shell) equivalent to the Hamiltonian and momentum constraints of gravity, and both of these generate translations of points on a Cauchy slice; momentum constraints generate diffeomorphisms of the Cauchy slice, and the Hamiltonian constraint generates translations of the Cauchy slice within the WdW patch. See Figure 10. Since we work with gauge-invariant states, they must be dual to superpositions over all embeddings of the lattice in the WdW patch. This also explains why two different ribbons on different sets of links can correspond to the same geodesic. The expectation value of our area operator on a certain cut agrees with the expectation value of a geodesic, but that does not mean that the links that the operator is supported on are dual to the geodesic.

Let us see how this works for the simplest case, the reduced lattice for $D^2$ without matter. The central ribbon on one link $b_1$, and that on two links $b_1 b_2$, measure the areas of two spacelike-separated surfaces, as shown in Figure 11. Mathematically, this is because the irreps $\mu_1, \mu_2$ on two boundary links might be entangled to produce only a subset of $\mu_1 \otimes \mu_2$, so that $\langle \hat{A}_{12} \rangle < \langle \hat{A}_1 \rangle + \langle \hat{A}_2 \rangle$. So the links $b_1, b_2$ should neither be interpreted as the surface $X_1 \cup X_2$ nor as the surface $X_{12}$. But different operators on these two links reproduce *properties* of either of these surfaces.[31]

---

[30]It's not quite a length, since the gauge field also has the spin connection in it. It exactly becomes a length for a geodesic; more generally, it is the length of a topologically equivalent geodesic. The connection between length of a geodesic and a mixture of length and spin connection (related to extrinsic curvature) of a different curve is the subject of [67].

[31]The most 'classical' lattice is the fusion basis lattice of [43, 48]; in that case, each link can be assigned a

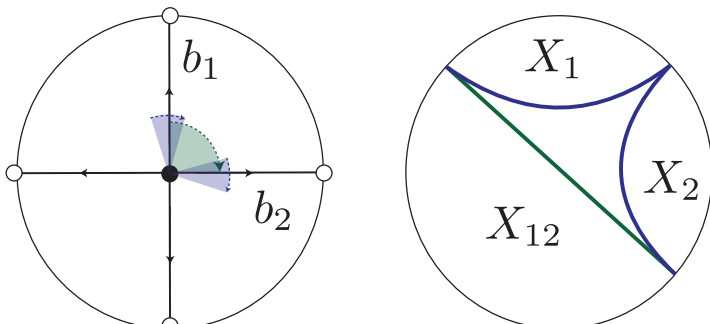

Figure 11: Central operators on individual links $b_1, b_2$ measure the area of different HRT surfaces, but the central operator on $b_1 \cup b_2$ measures the area of a completely different extremal surface that is spacelike-separated from both of them. Thus the two links should not be associated to individual extremal surfaces, even in the classical limit.

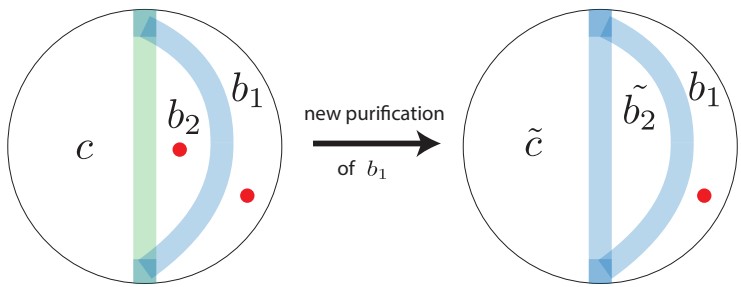

Figure 12: We can calculate the "area" of any cut, but that might not correspond to the length of any geodesic that exists in the spacetime. However there could be a purification of the bulk region where the geodesic exists. On the left, the green central ribbon is the minimal one, whose value should be related to the length of a geodesic. The blue ribbon is topologically inequivalent, and there may not be a geodesic with the right end-points in that topological class. However, there is a different purification of the region $b_1$ between the blue ribbon and the boundary, where there should be a geodesic of the same topological class.

Another potentially confusing point is that there are generically more topological classes of central ribbons than QESs, as shown in Figure 12. For example, if there are two matter excitations very close together but very far away from any QES, there is a ribbon that separates these two excitations but no QES between them. In that case, the area operator is not measuring the area of a QES in the geometry, but seems to be related to the outer entropy [68–70] or the holographic covariant entropy bound [67]. It is the area of a QES that would exist in a different geometry where one of the matter excitations has been taken away.[32]

---

particular QES. For example, in Figure 11, $b_1$ and $b_2$ would fuse to a third link, let's call it $b_{12}$, such that the irrep flowing out of $b_1 b_2$ is the irrep on $b_{12}$. Thus, a fusion basis lattice can be said to be lattice-dual to a non-intersecting network of geodesics, as pointed out also in [23]; $b_1, b_2, b_{12}$ would correspond to $X_1, X_2, X_{12}$ respectively. However, the central ribbon operator on $b_{12}$ equally can be represented on $b_1 \cup b_2$, and so there's still an element of convention to the identification.

[32]This is true when there is a normal surface between the two excitations; trapped surfaces are more confusing, and would likely require more explicit computations.

**A gauge theory of intertwiners**

On a related note, there are fascinating connections between our construction and the algebraic story of [46, 71]. They conceptualize the quantum error-correcting structure of AdS/CFT as an approximate version of Doplicher-Haag-Roberts theory, which is the algebraic approach to gapped theories with superselection sectors. The restriction of CFT operators to code subspace operators is implemented by a 'conditional expectation,' which in the DHR case is a restriction to operators that don't change the superselection sector.

Consider first the case without matter. There are two algebras acting on the links, $\mathcal{L}(\mathcal{H}_{\text{bdry}})$ and $\mathcal{L}(\mathcal{H}_{\text{phys}})$; consider the latter as the vacuum superselection sector of the former. The conditional expectation from $\mathcal{L}(\mathcal{H}_{\text{bdry}})$ to $\mathcal{L}(\mathcal{H}_{\text{phys}})$ is implemented by Gauss's law at the central vertex.[33] For two boundary regions $B_1, B_2$, the physical algebra $\mathcal{A}_{12} \in \mathcal{L}(\mathcal{H}_{\text{phys}})$ of the two regions is bigger than the algebra union $\mathcal{A}_1 \vee \mathcal{A}_2 \in \mathcal{L}(\mathcal{H}_{\text{phys}})$ of the two subregion algebras separately. The difference is made up of operators that create a charge in $B_1$ and an anti-charge in $B_2$, called intertwiners in the DHR theory. For $B_1, B_2$ adjacent, these intertwiners are made up of all ribbon operators that begin in $B_1$ and end in $B_2$. In the general case, it is the subset shown in Figure 16.[34]

Note that DHR theory deals with the boundary theory, but we are giving bulk descriptions of the objects involved: the TQFT is a _bulk_ gauge theory of _boundary_ intertwiners. This is perhaps not a surprise, since there is a reconstruction theorem relating DHR models in $1 + 1$ dimensions and tensor categories [72].

What is perhaps more novel is this. Since error-correction in AdS/CFT is approximate [73, 74], we have to back away from this limit of an exact bulk gauge theory of intertwiners, while keeping the bulk structure. This is hard, because bulk matter makes the theory non-topological and therefore harder to make holographic; adding the random tensors seems to be a way to mitigate this problem and allows us to back away from the topological limit while keeping holography. It would be interesting to understand these connections in detail.

## 6 Discussion

We have constructed a map

$$\mathcal{H}_{\text{bulk}} \rightarrow \mathcal{H}_{\text{bdry}}, \tag{88}$$

with the following properties: First, the bulk Hilbert space $\mathcal{H}_{\text{bulk}}$ consists of a topological theory coupled to matter. Second, boundary entropies satisfy a holographic entropy formula

$$S(B)_{V|\psi\rangle} = \langle\psi|\hat{A}(b)|\psi\rangle + S(b; \text{alg})_{|\psi\rangle}, \tag{89}$$

in which the minimization is over different sets of matter legs, and the area operator $\hat{A}$ is a (topological) operator measuring the net electric flux out of $B \cup b$. This is desirable if we would like a tensor network that represents a discretization of the topological quantum field theory (TQFT) description of $2+1d$ gravity, because in that description the area of the Ryu-Takayanagi (or quantum extremal) surface will correspond to some topological operator in the TQFT. As a byproduct, in this model the area operators of overlapping boundary regions fail to commute, as in gravity but not traditional tensor networks [36].

One could worry: isn't changing the area operator ruining the usefulness of tensor networks? Wasn't their whole point that boundary entropies were related to the cut through the graph that intersected the fewest links – resembling the Ryu-Takayanagi formula? A map with

---

[33]For general lattices, this is a closed ribbon around the boundary.
[34]They also define a dual set of 'twists.' In the DG model, these are also ribbon operators.

a topological area operator goes against that, so what is its point? The point is that the area operator in this model is still in line with the Ryu-Takayanagi formula. We still interpret $\hat{A}$ as measuring the geometric area, and we still are using random tensor networks to construct a map with an entropy formula minimizing (89). The only difference is that the TQFT description obscures the geometry of the gravitational description – so it is no surprise that what is geometric area in the gravitational description can be a topological operator in the TQFT description.

The reason the geometric description is obscured is twofold. First, our tensor networks do not introduce bulk-local degrees of freedom where there should be none. The matter-free theory is topological, so there is no need to add local tensors in this case. Secondly, our tensor networks are toy models for states invariant under the Hamiltonian constraint of gravity, and should be compared not with Cauchy slices but Wheeler-deWitt patches.

One oddity of our tensor networks is the particular subalgebras it minimizes over in the holographic entropy formula. While our tensor network has many desirable properties, these subalgebras seem fairly strange, as we discuss in Section 2.7 and Appendix C. It would be nice to better understand the gravitational analog of these subalgebras, or how to construct a tensor network that minimizes over more natural subalgebras.

The broader question of interest is constructing a tensor network with matching local dynamics on both sides of the map, akin to AdS/CFT. We believe the model in this paper will help with this, because it includes constraints that are more gravity-like. Perhaps one way to proceed will be to combine with this model the insights of [15, 16].

Having explained our main result and the broader question, let us summarize some important physics we learned along the way. The first lesson is that there is a qualitative difference between bipartite and multipartite edge modes. Abstractly, this arises from tensions between additivity and duality in local algebras in the code subspace [46]. The second, and most important, lesson is this: to model holography, we might need multipartite edge modes. The fact that the centers of overlapping regions fail to commute poses a challenge for purely bipartite edge modes.

These lessons lead to a number of structural questions. Remember, historically, that the bipartite edge modes were originally studied in gauge theories [37–39, 41, 42, 75], but then were shown to be a general feature of error-correcting codes with bipartite code spaces [10] and holography [76]. Perhaps we should regard this work as the first half of the same historical progression, for multipartite edge modes. Can we construct a general theory of multipartite edge modes and prove that they are needed in holography? What do multipartite edge modes look like in the gravitational phase space, analogous to [75] and follow-ups? Does the theorem in Appendix E generalize to a statement inferring a unique entropy function from a set of subalgebras? Understanding the multi-local operator that is dual to the soap-films of [77] will likely be useful to learn about these questions.

Let us end with some other interesting, but more specific, questions and directions for future work. We'd like to be able to interpret quantities in our model, like the area commutator, in terms of gravitational quantities. This could likely be done in the model of [23]. On a related note, the lattice model we have used as the base for our tensor network is related in spirit to loop quantum gravity [43, 48], and perhaps we can also import some insights from there. Another interesting question is to identify the operator analogous to our central ribbons in the Virasoro TQFT of [19], and see whether the factorization maps of [23, 30, 31] are related to these operators. As a first step, the analog of our factorization map in Chern-Simons is being studied [78]. It would also be interesting to study whether a model like ours realizes a quantum *extremal* surface formula, rather than just quantum minimality.[35] Finally, we could

---

[35]We would like to thank Jing-Yuan Chen, Bartek Czech, Alex Frenkel, Xiao-Liang Qi and Gabriel Wong for discussions on similar points.

mock up identity block domination of the CFT using specific string-net models, and see which gravity features we get from there.

Finally, we need to understand what we have learned about higher dimensions. Likely the connection to the story of [46] will be crucial in doing so.

## Acknowledgments

We thank Alejandra Castro, Jing-Yuan Chen, Shawn X. Cui, Bartek Czech, Xi Dong, Jackson R. Fliss, Alex Frenkel, Shahn Majid, Isaac Kim, Laurens Lootens, Xiao-Liang Qi, Daniel Ranard, Aron Wall, Gabriel Wong, and Bowen Yan for discussions.

**Funding information** CA is supported by National Science Foundation under the grant number PHY-2207584, the Sivian Fund, and the Corning Glass Works Foundation Fellowship. RMS is supported by the Isaac Newton Trust grant "Quantum Cosmology and Emergent Time" and the (United States) Air Force Office of Scientific Research (AFOSR) grant "Tensor Networks and Holographic Spacetime," and also partially supported by STFC consolidated grant ST/T000694/1. They also thank UC Berkeley for hospitality. AYW acknowledges funding from the US DOE center "Quantum Systems Accelerator."

## A  Lattice deformations

Recall that we think about lattice independence as follows. Say we start with a lattice $\Lambda_1$, defining $\mathcal{H}^{(1)}_{\text{pre}}$, projectors $A^{(1)}$ and $B^{(1)}$, and physical Hilbert space $\mathcal{H}_{\text{phys}} = A^{(1)}B^{(1)}\mathcal{H}^{(1)}_{\text{pre}}$. There are two "elementary moves" that change the lattice but leave the physical Hilbert space unchanged, see e.g. [50,51]. That is, applying one of these elementary moves would give us a $\Lambda_2$, such that $\Lambda_2$ defines $\mathcal{H}^{(2)}_{\text{pre}}$ and projectors $A^{(2)}$ and $B^{(2)}$ with

$$A^{(2)}B^{(2)}\mathcal{H}^{(2)}_{\text{pre}} = A^{(1)}B^{(1)}\mathcal{H}^{(1)}_{\text{pre}}. \tag{A.1}$$

Each move is an isometry (or co-isometry, in the reverse direction) that we can write down explicitly. To do so, we introduce the following controlled multiplication operators:

**Definition A.1.** Let $\mathcal{H}_C \cong \mathcal{H}_T \cong \mathcal{H}_G$. Given the bipartite Hilbert space $\mathcal{H}_C \otimes \mathcal{H}_T$, we call the first factor the "control" and the second factor the "target" when using the following four controlled multiplication operators:

$$C_{II,CT} |h\rangle_C |g\rangle_T := |h\rangle_C |gh\rangle_T , \tag{A.2}$$

$$C_{IO,CT} |h\rangle_C |g\rangle_T := |h\rangle_C \left|h^{-1}g\right\rangle_T , \tag{A.3}$$

$$C_{OI,CT} |h\rangle_C |g\rangle_T := |h\rangle_C \left|gh^{-1}\right\rangle_T , \tag{A.4}$$

$$C_{OO,CT} |h\rangle_C |g\rangle_T := |h\rangle_C |hg\rangle_T . \tag{A.5}$$

The two elementary moves are as follows.

**Move 1: Add (or remove) a vertex**

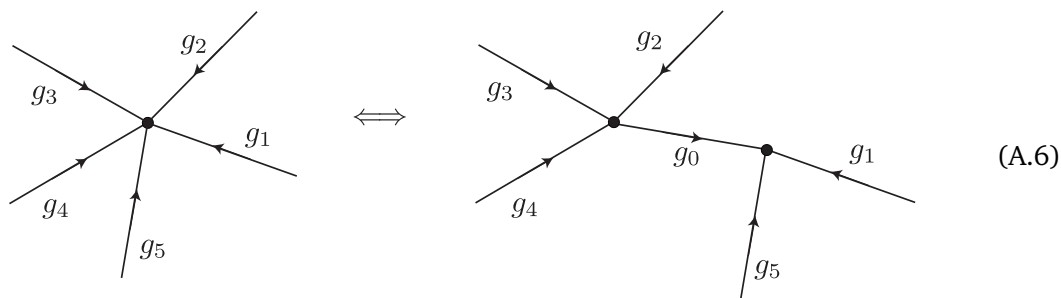

$$\Longleftrightarrow \qquad\qquad\qquad\qquad\qquad\qquad\qquad\qquad (A.6)$$

Concretely, we define the isometry $V_{\text{vertex}} : \mathcal{H}_{\text{pre}}^{(1)} \to \mathcal{H}_{\text{pre}}^{(2)}$ in a two step process. First, we tensor onto our state $|\psi\rangle \in \mathcal{H}_{\text{pre}}^{(1)}$ the state $|\mathbf{1}\rangle \in \mathcal{H}_G$, where

$$|\mathbf{1}\rangle := \frac{1}{\sqrt{|G|}} \sum_{g \in G} |g\rangle \,, \tag{A.7}$$

is the state corresponding to the trivial irrep. Second, we act controlled multiplication operators with the control being this new factor and the target being a set of links attached to this vertex and all adjacent to each other. It doesn't matter the order in which we act these. In (A.6), the targets would be the $g_5$ link and the $g_1$ link (or the $g_2, g_3, g_4$ links). Which of the four control operations we use depends on the orientations of the two links. If they are both oriented "in" towards the vertex, then we use $C_{II}$ (the $II$ stands for In-In). If the control is oriented in but target out, we use $C_{IO}$ (for In-Out), and so on. In total, in (A.6) we'd have

$$
\begin{aligned}
V_{\text{vertex}} |g_1\rangle_1 |g_2\rangle_2 |g_3\rangle_3 \big|g_4\big\rangle_4 |g_5\rangle_5 &= C_{II,05} C_{II,01} |\mathbf{1}\rangle_0 |g_1\rangle_1 |g_2\rangle_2 |g_3\rangle_3 \big|g_4\big\rangle_4 |g_5\rangle_5 \\
&= \frac{1}{\sqrt{|G|}} \sum_{g_0 \in G} |g_0\rangle_0 |g_1 g_0\rangle_1 |g_2\rangle_2 |g_3\rangle_3 \big|g_4\big\rangle_4 |g_5 g_0\rangle_5 \,.
\end{aligned}
\tag{A.8}
$$

It is easy to confirm this satisfies Gauss's law at both new vertices (if the original vertex satisfied Gauss's law), as well as any flatness constraint in any adjacent plaquettes that satisfied it in the original lattice.

Removing a vertex happens by reversing the steps. In (A.6), we would remove the $g_0$ link by first acting with $C_{II,05}^{-1} C_{II,01}^{-1}$, then acting with $\langle \mathbf{1}|_0$. Note this is *not* an isometry, so we may be surprised that this leads to a sensible one-to-one identification of physical operators. Indeed it does. The key point is that *no matter what state* we start with in $AB\mathcal{H}_{\text{pre}}$, acting $C_{II,05}^{-1} C_{II,01}^{-1}$ is sure to put link 0 in the state $|\mathbf{1}\rangle_0$. In other words, $V_{\text{vertex}}^\dagger$ might not be an isometry on $\mathcal{H}_{\text{pre}}$, but it's bijective on the physical subspace.

An important special case of this move is to split one link into two:

$$\bullet \xrightarrow{\hspace{1.5cm} g_1 \hspace{1.5cm}} \bullet \quad \Longleftrightarrow \quad \bullet \xrightarrow{\hspace{0.8cm} g_0 \hspace{0.8cm}} \bullet \xrightarrow{\hspace{0.8cm} g_1 \hspace{0.8cm}} \bullet \tag{A.9}$$

In this case,

$$V_{\text{vertex}} |g_1\rangle_1 = \frac{1}{\sqrt{|G|}} \sum_{g_0 \in G} |g_0\rangle_0 \big|g_0^{-1} g_1\big\rangle_1 \,. \tag{A.10}$$

Note that in the irrep basis this takes the form

$$V_{\text{vertex}} |\mu, ij\rangle_1 = \frac{1}{\sqrt{d_\mu}} \sum_{k=1}^{d_\mu} |\mu, ik\rangle_0 |\mu, kj\rangle_1 \,. \tag{A.11}$$

**Move 2: Add (or remove) a plaquette**

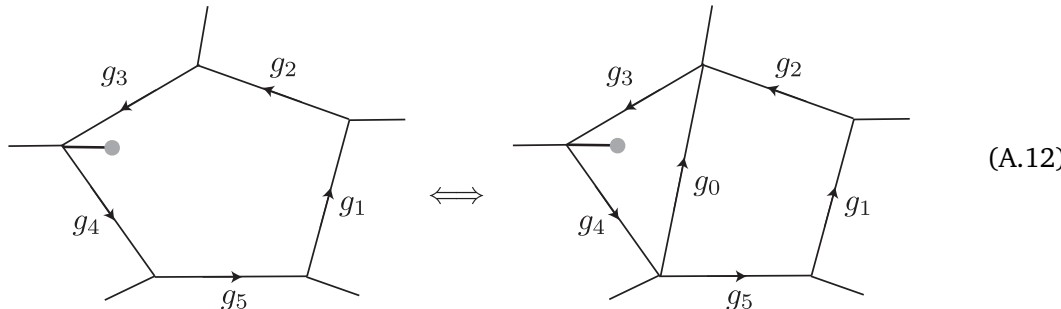

$$(A.12)$$

Concretely, we again proceed in two steps. In the first step, we append to $|\psi\rangle \in \mathcal{H}_{\text{pre}}^{(1)}$ the state $|e\rangle \in \mathcal{H}_G$, where $e$ is the identity group element. Second, we move counterclockwise around the plaquette that was just formed (if two were formed, pick one), at each link acting a controlled multiplication operator, with the new link as the target. Which controlled multiplication depends on the orientations of the two links. If the new link is oriented clockwise (respectively counterclockwise), then the target is $I$ (respectively $O$). If the other link is oriented clockwise (respectively counterclockwise), then the control is $O$ (respectively $I$). In (A.12), we would start by appending $|e\rangle_0$ and then acting $C_{II,50}$. Then we would act $C_{II,10}$ and then $C_{II,20}$. In total we'd have

$$
\begin{aligned}
V_{\text{plaq}} |g_1\rangle_1 |g_2\rangle_2 |g_3\rangle_3 |g_4\rangle_4 |g_5\rangle_5 &= C_{II,20} C_{II,10} C_{II,50} |e\rangle_0 |g_1\rangle_1 |g_2\rangle_2 |g_3\rangle_3 |g_4\rangle_4 |g_5\rangle_5 \\
&= |g_2 g_1 g_5\rangle_0 |g_1\rangle_1 |g_2\rangle_2 |g_3\rangle_3 |g_4\rangle_4 |g_5\rangle_5 \, .
\end{aligned}
\tag{A.13}
$$

It is straightforward to see that the new plaquette has trivial holonomy. Here, $g_0^{-1} g_5 g_1 g_2 = e$. We can see that Gauss's law is satisfied at the two modified vertices (if they satisfied Gauss's law before) by writing out the states before and after explicitly.

To remove a plaquette, we perform the inverse operations. Here we'd act

$$
\langle e|_0 \, C_{OO,50}^{-1} C_{OO,10}^{-1} C_{OO,20}^{-1} \, .
\tag{A.14}
$$

This is not an isometry on $\mathcal{H}_{\text{pre}}$, but it still defines a one-to-one identification of physical states and operators.

Note that it did not matter that we added a link inside of a plaquette that already existed. It would have worked to take an unclosed set of links – which do not form a plaquette and therefore do not satisfy any flatness constraint – and close them by adding a new link. The new plaquette satisfies the flatness constraint regardless. The reverse operation is also important: we can take a plaquette on the edge of a lattice, then remove the outermost link.

**Ribbon operator transformation**

A ribbon operator $F$ acts on $\mathcal{H}_{\text{phys}}$ and therefore must be represented on any associated lattice. We can ask: given $F_\gamma(h, g)$ acting on $\mathcal{H}_{\text{pre}}^{(1)}$, what is the associated operator on $\mathcal{H}_{\text{pre}}^{(2)}$? The answer is that it is also a ribbon operator, now including the new link if a plaquette was added along

its path. For example:

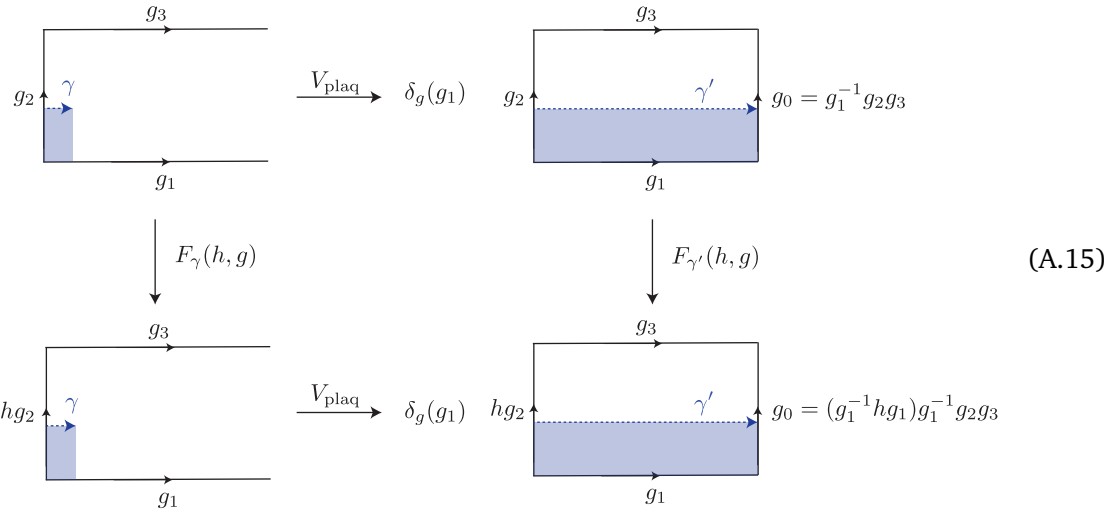

$$(A.15)$$

One way to see how $F_\gamma(h,g)$ maps is to recall that $V_{\text{plaq}} = C \left|e\right\rangle_0$, where $C$ is a unitary and $\ell_0$ is a new link introduced in the identity element state. Then we know that what we want is an operator $O$ that satisfies

$$\forall \left|\psi\right\rangle \in \mathcal{H}_{\text{phys}}, \qquad OV\left|\psi\right\rangle = VF_\gamma(h,g)\left|\psi\right\rangle. \tag{A.16}$$

This is satisfied by the anzats $O = C(\mathbb{1}_0 \otimes F_\gamma(h,g))C^\dagger$. In the plaquette example (A.15), we have $C = C_{IO,10}C_{OO,20}, C_{OO,30}$, and we compute

$$\begin{aligned}
C(\mathbb{1}_0 \otimes F_\gamma(h,g))C^\dagger \left|g_0, g_1, g_2, g_3\right\rangle &= C(\mathbb{1}_0 \otimes F_\gamma(h,g))\left|g_3^{-1}g_2^{-1}g_1 g_0, g_1, g_2, g_3\right\rangle \\
&= \delta_g(g_1)C\left|g_3^{-1}g_2^{-1}g_1 g_0, g_1, hg_2, g_3\right\rangle \\
&= \delta_g(g_1)\left|g_1^{-1}hg_1 g_0, g_1, hg_2, g_3\right\rangle \\
&= F_{\gamma'}(h,g)\left|g_0, g_1, g_2, g_3\right\rangle.
\end{aligned} \tag{A.17}$$

We see that $F_\gamma(h,g)$ has become a larger ribbon operator on $\gamma'$, which passes across the $g_1$ link to act on the $g_0$ link.

**Equivalence of the Hilbert spaces**

Now let us sketch the argument that the two elementary moves of Section 2.3 leave the physical Hilbert space unchanged, and that each physical operator maps 1-to-1 to a physical operator in the new Hilbert space. Consider $V_{\text{vertex}}$. Let $\Lambda_1 = (V_1, L_1, P_1)$ and $\Lambda_2 = (V_2, L_2, P_2)$ be two lattices related by a move as in (A.6). Without loss of generality, say in $\Lambda_1$ there is an $n$-valent vertex $v_1 \in V_1$ with incident links $\ell_1, \ell_2, \cdots \ell_n$ all pointing in toward $v$, while in $\Lambda_2$ there is an $m+1$-valent vertex $v_2$ with $m < n$ incident links $\ell_0, \ell_1, \cdots \ell_m$ with $\ell_0$ pointing out from $v_2$ while the rest point inwards, and an $n-m$-valent vertex $v_2'$ with incident links $\ell_0, \ell_{m+1}, \ell_{m+2}, \cdots, \ell_n$, with all pointing inwards toward $v_2'$.

$\Lambda_1$ and $\Lambda_2$ define pre-gauged Hilbert spaces $\mathcal{H}_{\text{pre}}^{(1)}$ and $\mathcal{H}_{\text{pre}}^{(2)}$ respectively. Moreover, in our initial lattice $\Lambda_1$, we have the constraint projectors

$$\begin{aligned}
A^{(1)} &= \bigotimes_{v \in V_{\text{bulk}}^{(1)}} A_v^{(1)}, \\
B^{(1)} &= \bigotimes_{p \in P_{\text{bulk}}^{(1)}} B_p^{(1)},
\end{aligned} \tag{A.18}$$

defining the physical Hilbert space $\mathcal{H}_{\mathrm{phys}}^{(1)} := A^{(1)}B^{(1)}\mathcal{H}_{\mathrm{pre}}^{(1)}$. Similarly, $\Lambda_2$ comes with constraint projectors $A^{(2)}$ and $B^{(2)}$. Crucially, $A^{(2)}$ is very much like $A^{(1)}$ except that $A^{(1)}$ acts on $v_1$ while $A^{(2)}$ instead acts on $v_2, v_2'$. $B^{(2)}$ is like $B^{(1)}$ except possibly one or two plaquettes now also involve the new link $\ell_0$. By construction,

$$
\begin{aligned}
V_{\mathrm{vertex}} &: \mathcal{H}_{\mathrm{pre}}^{(1)} \to \mathcal{H}_{\mathrm{pre}}^{(2)}, \\
V_{\mathrm{vertex}}^{\dagger} &: \mathcal{H}_{\mathrm{pre}}^{(2)} \to \mathcal{H}_{\mathrm{pre}}^{(1)}.
\end{aligned}
\tag{A.19}
$$

What we wish to show is that in particular

$$
V_{\mathrm{vertex}} : A^{(1)}B^{(1)}\mathcal{H}_{\mathrm{pre}}^{(1)} \to A^{(2)}B^{(2)}\mathcal{H}_{\mathrm{pre}}^{(2)},
\tag{A.20}
$$

$$
V_{\mathrm{vertex}}^{\dagger} : A^{(2)}B^{(2)}\mathcal{H}_{\mathrm{pre}}^{(2)} \to A^{(1)}B^{(1)}\mathcal{H}_{\mathrm{pre}}^{(1)}.
\tag{A.21}
$$

To show (A.20), recall that when acting $V_{\mathrm{vertex}}$ we first append an ancilla in the state $|\mathbf{1}\rangle$, and then we act unitary controlled multiplication operators. Say we start with state $|\psi\rangle \in \mathcal{H}_{\mathrm{pre}}^{(1)}$ that satisfies $A^{(1)}B^{(1)}|\psi\rangle$. After appending the ancilla $\ell_0$ but before the unitaries, we have the state

$$
|\mathbf{1}\rangle_0 \otimes |\psi\rangle_{1\cdots m} \in \mathcal{H}_G \otimes \mathcal{H}_{\mathrm{pre}}^{(1)},
\tag{A.22}
$$

which satisfies

$$
L_0(g) \otimes (\otimes_{i=1}^n R_i(g)) |\mathbf{1}\rangle_0 \otimes |\psi\rangle_{1\cdots n} = |\mathbf{1}\rangle_0 \otimes |\psi\rangle_{1\cdots n},
\tag{A.23}
$$

$$
R_g(0) \otimes \mathbb{1}_{1\cdots n} |\mathbf{1}\rangle_0 \otimes |\psi\rangle_{1\cdots n} = |\mathbf{1}\rangle_0 \otimes |\psi\rangle_{1\cdots n}.
\tag{A.24}
$$

To complete the action of $V_{\mathrm{vertex}}$ we then act with the unitary operator

$$
C := \prod_{i \in \{m+1, \cdots, n\}} C_{II,0i}.
\tag{A.25}
$$

It follows that

$$
C(R_0(g) \otimes \mathbb{1}_{1\cdots n})C^{\dagger} V_{\mathrm{vertex}}|\psi\rangle = V_{\mathrm{vertex}}|\psi\rangle.
\tag{A.26}
$$

We compute that

$$
C_{II,0i}(R_0(g) \otimes \mathbb{1}_i)C_{II,0i}^{\dagger} = R_0(g) \otimes R_i(g),
\tag{A.27}
$$

and therefore

$$
C\left(\frac{1}{|G|}\sum_{g \in G} R_0(g) \otimes \mathbb{1}_{1\cdots n}\right)C^{\dagger} = A_{v_2'}^{(2)}.
\tag{A.28}
$$

We have shown $A_{v_2'}^{(2)} V_{\mathrm{vertex}}|\psi\rangle = V_{\mathrm{vertex}}|\psi\rangle$. Now we want to show the same for $A_{v_2}^{(2)}$. We compute

$$
C_{II,0i}(L_0(g) \otimes R_i(g))C_{II,0i}^{\dagger} = L_0(g) \otimes \mathbb{1}_i.
\tag{A.29}
$$

This implies

$$
C\left(L_0(g) \otimes (\otimes_{i=1}^n R_i(g))\right)C^{\dagger} = L_0(g) \otimes R_1(g) \otimes \cdots \otimes R_m(g) \otimes \mathbb{1}_{m+1} \otimes \cdots \otimes \mathbb{1}_n.
\tag{A.30}
$$

It follows that $A_{v_2}^{(2)} V_{\mathrm{vertex}}|\psi\rangle = V_{\mathrm{vertex}}|\psi\rangle$. All that is left is to argue $B^{(2)}V_{\mathrm{vertex}}|\psi\rangle = V_{\mathrm{vertex}}|\psi\rangle$. This follows because the introduction of $|\mathbf{1}\rangle$ does not change any of the holonomies, and the action of $C$ also preserves the holonomies.

The reverse direction (A.21) and the analogs for the $B$ constraints can be shown similarly.

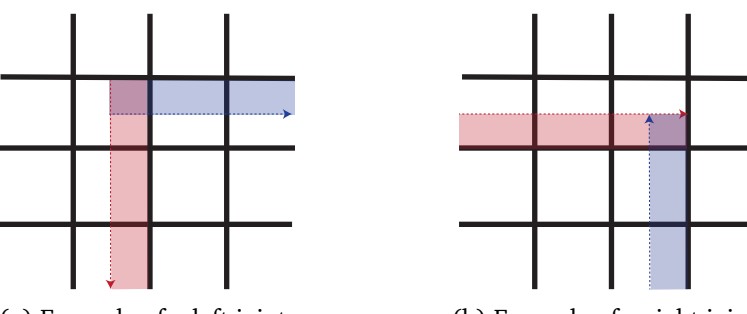

(a) Example of a left joint.   (b) Example of a right joint.

Figure 13: The two types of joints between two ribbons.

# B  Central ribbon operators

**Central ribbons**

Here we derive the form of the central ribbon operators for a bipartition of the disk into two regions $b$ and $\overline{b}$. We assume that both $b, \overline{b}$ are topologically $D^2$, and $\eth b = \eth\overline{b}$ is topologically an interval.

**Theorem B.1.** *The center of the algebra $\mathcal{A}_b$ associated to $b$ is*

$$F_{\eth b}([h]) = \frac{1}{|[h]|} \sum_{\substack{w \in [h] \\ g \in G}} F_\gamma(w, g),$$

*where $[k]$ is the conjugacy class of $k \in G$, and $\gamma$ is a ribbon whose spokes are $\eth b$, and whose spine is the path connecting the vertices in $V_b$ adjacent to links in $\eth b$.*

Note that two ribbons $F_{\gamma_1}([h])$, $F_{\gamma_2}([h])$, where $\gamma_1$ and $\gamma_2$ are anchored to the same boundary endpoints, are the same ribbon provided that it is possible to deform $\gamma_1$ into $\gamma_2$ without passing through any charges.

First, we need some results and definitions that appear in [49]. We say that we have a left joint when two ribbons diverge outwards from a common point, as shown in Figure 13a. Similarly, we have a right joint when two ribbons converge towards a common point, as shown in Figure 13b. Then we can use the following result from [49]:

**Proposition B.2.** Let $\gamma_1$ and $\gamma_2$ be ribbons satisfying a left joint relation. Then they satisfy the following commutation relation:

$$F_{\gamma_1}(h, g)F_{\gamma_2}(k, \ell) = F_{\gamma_2}(hkh^{-1}, h\ell)F_{\gamma_1}(h, g). \tag{B.1}$$

Let $\gamma_3$ and $\gamma_4$ be ribbons satisfying a right joint relation. Then they satisfy the following commutation relation:

$$F_{\gamma_3}(h, g)F_{\gamma_4}(k, \ell) = F_{\gamma_3}(k, \ell g^{-1}h^{-1}g)F_{\gamma_4}(h, g). \tag{B.2}$$

To prove our theorem, we need a number of intermediate results. The first is that the ribbons of interest are the center of a particular set of ribbon operators,

**Proposition B.3.** Consider two ribbons $\gamma, \gamma'$ that satisfy either a left-joint or a right-joint relation. Then, for $k \in G$, the ribbon operators

$$F_\gamma([k]) = \frac{1}{|[k]|} \sum_{\substack{k \in [k] \\ l \in G}} F_\gamma(k, l), \tag{B.3}$$

commute with all possible ribbon operators on $\gamma'$ and $\gamma$.

*Proof.* We look for a ribbon on $\gamma$ that has this property

$$F_{\gamma,c} = \sum_{k,l \in G} f(k,\ell) F_\gamma(k,l). \tag{B.4}$$

Imposing commutation with an arbitrary ribbon on $\gamma'$ by plugging this into (B.1) yields the condition

$$\forall h, \; f(k,l) = f(hkh^{-1}, hl). \tag{B.5}$$

The two combinations of $k, l$ invariant under this set of transformations are the conjugacy class $[k]$ and $l^{-1}kl$, so we find $f(k,l) = f([k], l^{-1}kl)$.

The product of two operators on the same ribbon can be easily worked out to be

$$F_\gamma(k,l) F_\gamma(h,g) = \delta_{l,g} F_\gamma(kh,l). \tag{B.6}$$

Imposing commutation of (B.4) with $F_\gamma(h,g)$ gives

$$\begin{aligned}
F_{\gamma,c} F_\gamma(h,g) &= \sum_{k \in G} f([k], g^{-1}kg) F_\gamma(kh,g) \\
&= \sum_{k' \in G} f([k'], g^{-1}hk'h^{-1}g) F_\gamma(hk',g) \\
&= F_\gamma(h,g) \sum_{k' \in G} f([k'], l^{-1}hk'h^{-1}l) F_\gamma(k',l).
\end{aligned} \tag{B.7}$$

In the second line, we have defined $k' = h^{-1}kh$ and used the invariance of the sum under conjugation. The last line only equals $F_\gamma(h,g) F_{\gamma,c}$ if we take

$$f(k,l) = f([k]). \tag{B.8}$$

The right joint condition doesn't yield any additional constraints. $\qquad\square$

This proposition will be sufficient to prove our theorem in the case without matter. To include matter, we need some more results.

**Proposition B.4.** Central ribbon operators live on a special type of ribbon. The ribbon $\gamma$ can end on a spoke, as long as that spoke borders only one plaquette in $P_{\text{bulk}}$, like

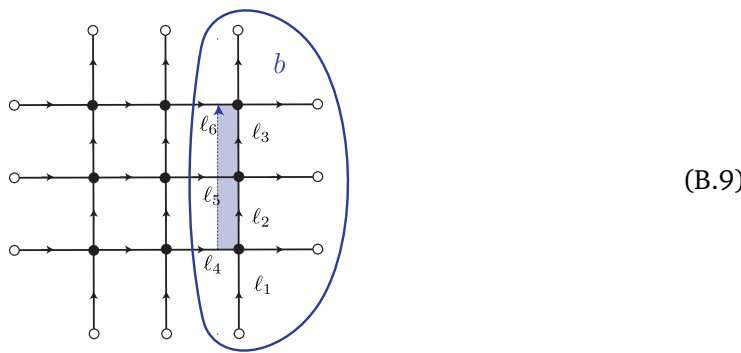

$$\tag{B.9}$$

Below, we state this concisely as "central ribbons end on a spoke."

*Proof.* We can see this directly from our definition. Focusing on $\ell_2$ in (B.9), the operator can be written as

$$F_{\delta b}([h]) = \frac{1}{|[h]|} \sum_{w \in [h], g \in G} T_{\ell_1, \bar{\ell}_4}(w) T_{\ell_1 \ell_2, \bar{\ell}_5}(w) T_{\ell_1 \ell_2 \ell_3, \bar{\ell}_6}(w). \tag{B.10}$$

Remembering that the operator $F_{\ell_1}(e, g_1)$ projects the link $\ell_1$ to the value $g_1$, we can write the summand as

$$T_{\ell_1, \bar{\ell}_4}(w) T_{\ell_1 \ell_2, \bar{\ell}_5}(w) T_{\ell_1 \ell_2 \ell_3, \bar{\ell}_6}(w) = \sum_{g_1 \in G} F_{\ell_1}(e, g_1) L_{\bar{\ell}_1}(g_1^{-1} w g_1) T_{\ell_2, \bar{\ell}_5}(g_1^{-1} w g_1) T_{\ell_2 \ell_3, \bar{\ell}_6}(g_1^{-1} w g_1). \quad \text{(B.11)}$$

We exchange the $g_1$ and $w$ sums, then define $w' = g^{-1} w g$ and use $\sum_w = \sum_{w'}$ (because we are summing over a conjugacy class) to find that

$$F_{\delta b}([h]) = \left( \sum_{g_1} F_{\ell_1}(e, g) \right) \frac{1}{|[h]|} \sum_{w \in [h], g \in G} L_{\bar{\ell}_1}(w) T_{\ell_2, \bar{\ell}_5}(w) T_{\ell_2 \ell_3, \bar{\ell}_6}(w). \quad \text{(B.12)}$$

The operator in the brackets is the identity operator, since it is a sum over a complete set of projectors. Thus, we find that the central ribbon has no action on $\ell_1$, justifying our claim.

The proof extends straightforwardly to arbitrary ribbons $\gamma$. $\qquad\square$

The qualitative reason is this: we need to end a generic ribbon operator on $V_{\text{bdry}}$ because it violates Gauss's law at the end-point. But the sum over conjugacy classes already makes it commutes with Gauss's law. We will call these special ribbons that don't end on $V_{\text{bdry}}$ while still supporting physical operators *central ribbons*.

The last intermediate result is the following:

**Proposition B.5.** The central ribbon operator $F_{\delta b}([k])$ on one side of a cut is the same as an equivalent ribbon operator $F_{\bar{\delta b}}([k^{-1}])$ on the opposite side of the cut. Thus this operator lives in both algebras.

For example, the two ribbons in

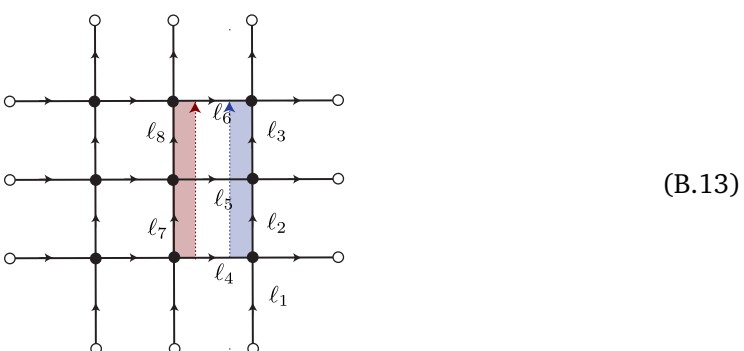

(B.13)

are related in this way.

*Proof.* The fundamental fact we need is that the electric operator on one side is the same as the transported shift acting on the other side, $R_\ell(h) = T_{\ell, \ell}(h^{-1})$. More generally, denoting by $\rho \ell$ the path obtained by adjoining $\ell$ to the end of $\rho$,

$$T_{\rho, \ell}(h) = T_{\rho \ell, \bar{\ell}}(h^{-1}). \quad \text{(B.14)}$$

Then, specializing to the example,

$$F_{\delta b}(h) = \frac{1}{|[h]|} \sum_{w \in [h], g \in G} T_{\bar{\ell}_1, \bar{\ell}_1}(w^{-1}) T_{\ell_2 \bar{\ell}_5, \ell_5}(w^{-1}) T_{\ell_2 \ell_3 \bar{\ell}_6, \ell_6}(w^{-1}). \quad \text{(B.15)}$$

But the parallel-transport along $\ell_2 \bar{\ell}_5$ ($\ell_2 \ell_3 \bar{\ell}_6$) is the same as the parallel-transport along $\bar{\ell}_4 \ell_7$ ($\bar{\ell}_4 \ell_7 \ell_8$), and so we can shift the first argument in each $T$ operator accordingly. Then, by the same argument as in the proof of Proposition B.4, we can get rid of the $\bar{\ell}_1$. We then find the red ribbon in (B.13), where the shift parameter is summed over $[h^{-1}]$ instead of $[h]$, as claimed.

Again, the proof extends to other ribbons in a straightforward way.

Note here the importance of our requirement that the dual path $\eth b$ not intersect any plaquettes containing matter. If any of them did contain matter, then we couldn't use flatness and the argument would not go through. $\qquad \square$

Now we are ready for the following proof.

*Proof of Theorem B.1.* First, consider the case without matter, since the argument is more direct. We first remember that the center commutes with every operator in $b$, and that the algebra of $b$ is generated by the ribbon operators. We consider three types of ribbons: those that share no end-points with $\gamma$, those that share one, and those that share two. Operators on ribbons that share no end-points with $\gamma$ can always be deformed so that their support has no intersection with $\gamma$. Those that share one, which we will call $\gamma'$, can be deformed so that $\gamma, \gamma'$ satisfy either a left-joint or a right-joint relation. $\gamma$ and $\gamma'$ cannot cross per se, since such a $\gamma'$ would leave $b$. Finally, any ribbon that shares both end-points with $\gamma$ is topologically the same as $\gamma$ itself. So, the non-trivial commutation relations to check are exactly those checked in Proposition B.3. This proves it in this case.

In the presence of matter, there are many more topologically inequivalent ribbons in $b$. So we opt for an indirect argument. Notice that any operator supported entirely in $\overline{b}$ is in the commutant $\mathcal{A}'_b$ of the algebra $\mathcal{A}_b$ of operators supported in $b$, for the simple reason that spacelike-separated operators commute. Furthermore, the center is defined exactly as $\mathcal{Z}_b = \mathcal{A}_b \cap \mathcal{A}'_b$. Proposition B.5 shows that our ribbons satisfy this property.

Finally, we have to show in both cases that these operators generate the full center. Firstly, note that only ribbon operators containing (a) no projection along the magnetic part and (b) a sum over the electric parameter weighted by a class function can be deformed out of $b$. We need the first because if the ribbon projects the spine then it has a non-trivial action on a link in $L_{\mathrm{bdry}}$; and we need the second to run the argument that proved Proposition B.4 to make the electric action not depend on a link in $L_{\mathrm{bdry}}$ (in our example, this link is $\ell_1$). And if an operator has a non-trivial action on a link in $L_{\mathrm{bdry}} \cup b$, then it cannot be supported entirely in $\overline{b}$.

Thus, the only candidates are operators that have the same structure as our central ribbons, but supported on some ribbon $\gamma'$ that is topologically distinct from $\gamma$. It can be topologically distinct for two reasons. The first is that there is some matter excitation between the two. In this case, the operator on $\gamma'$ cannot be deformed to $\overline{b}$ because it cannot cross the matter excitation. The second reason $\gamma'$ might be topologically distinct is that it does not share both end-points with $\gamma$. In that case also, we cannot deform it to have support only in $\overline{b}$. As an example, take the same lattice as before but a different choice for $b$,

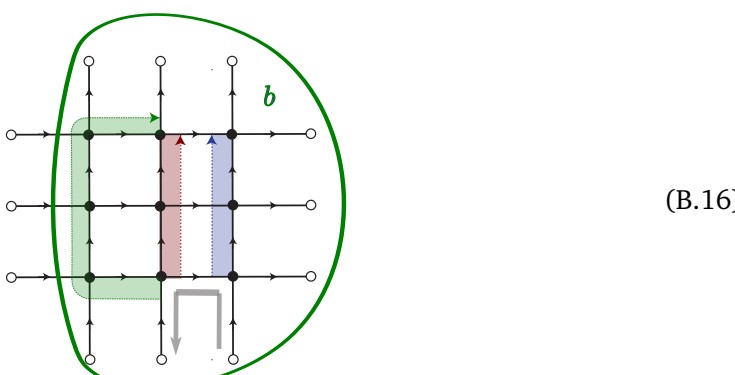

$$(B.16)$$

Now, the blue ribbon is no longer supported on $\eth b$; trying to deform it towards the boundary of $b$, we get first the red and then the green ribbon. The reason is that the operator has non-zero commutation with the holonomy shown as a thick gray arrow in (B.16). (Note that this holonomy is a boundary-anchored Wilson line, and therefore it's gauge-invariant.)[36] But this requirement causes the green ribbon to have the horizontal parts above and below, so that it is always partly supported in $b$. So only a ribbon supported on $\eth b$ itself can be deformed to $\bar{b}$. $\qquad\square$

**Irrep basis**

It turns out that a Fourier-transformed set that measures the total irrep flowing out of $b$ is more useful for us.

**Theorem B.6.** *The central ribbons,*

$$F_{\eth b}(\mu) = \frac{d_\mu}{|G|} \sum_{[k] \in G} \chi_\mu(k) F_{\eth b}([k]), \tag{B.17}$$

*are a complete, orthonormal set of projectors,*

$$F_{\eth b}(\mu) F_{\eth b}(\nu) = \delta_{\mu\nu} F_{\eth b}(\mu), \qquad \sum_\mu F_{\eth b}(\mu) = \mathbb{1}. \tag{B.18}$$

*Proof.* We first prove that they are projectors. Unpacking the definition, $F_{\eth b}(\mu)$ can be written as

$$
\begin{aligned}
F_{\eth b}(\mu) &= \frac{d_\mu}{|G|} \sum_{h,g \in G} \chi_\mu(g) \frac{1}{|[h]|} \sum_{w \in [h]} F_\gamma(w, g) \\
&= \frac{d_\mu}{|G|} \sum_{h,g \in G} \chi_\mu(g) F_\gamma(h, g).
\end{aligned} \tag{B.19}
$$

Here, $\gamma$ is the ribbon associated to $\eth b$, as defined in the main text. To go from the first line to the second, we use $\sum_{h \in G} |[h]|^{-1} f([h]) = \sum_{h \in C_G} f([h])$ and $\sum_{h \in C_G} \sum_{w \in [h]} f(w) = \sum_{h \in G} f(h)$.

Taking the product of two irrep ribbons,

$$
\begin{aligned}
F_{\eth b}(\mu) F_{\eth b}(\mu') &= \frac{d_\mu d_{\mu'}}{|G|^2} \sum_{g,h,g',h' \in G} \chi_\mu(h) \chi_{\mu'}(h') F_\gamma(h, g) F_\gamma(h', g') \\
&= \frac{d_\mu d_{\mu'}}{|G|^2} \sum_{g,h,h' \in G} \chi_\mu(h) \chi_{\mu'}(h') F_\gamma(hh', g) \\
&= \frac{d_\mu d_{\mu'}}{|G|^2} \sum_{g,\tilde{h}} \left[ \sum_{h'} \chi_\mu(\tilde{h} h'^{-1}) \chi_{\mu'}(h') \right] F_\gamma(\tilde{h}, g).
\end{aligned} \tag{B.20}
$$

In the last line, we have defined $\tilde{h} = hh'$ and used $\sum_{h,h'} = \sum_{\tilde{h},h'}$.

To evaluate the quantity in the square brackets, we need to expand the characters in terms of representation matrices $\chi_\mu(h) = D_{ii}^\mu(h)$, summation assumed. These representation matrices have the following orthogonality property,

$$\sum_h D_{ij}^\mu(h) D_{i'j'}^{\mu'}(h^{-1}) = \frac{|G|}{d_\mu} \delta^{\mu\mu'} \delta_{ij'} \delta_{ji'}. \tag{B.21}$$

---

[36]This non-commutation is the true reason such a ribbon cannot be in the center. For a ribbon supported on $\eth b$, this holonomy is contained in neither $\mathcal{A}_b$ nor $\mathcal{A}_b'$, allowing our ribbons to be central.

We apply it as follows,

$$
\begin{aligned}
\sum_{h' \in G} \chi_\mu(\tilde{h} h'^{-1}) \chi_{\mu'}(h') &= D_{ji}^\mu(\tilde{h}) \sum_{h' \in G} D_{ij}^\mu(h'^{-1}) D_{i'i'}^{\mu'}(h') \\
&= D_{ji}^\mu(\tilde{h}) \frac{|G|}{d_\mu} \delta_{\mu\mu'} \delta_{ii'} \delta_{ji'} \\
&= \frac{|G|}{d_\mu} \chi_\mu(\tilde{h}) \delta_{\mu\mu'} .
\end{aligned}
\tag{B.22}
$$

Plugging this back in to (B.20), we find the orthogonality of projectors, as advertised.

To prove completeness of the set of projectors we use character orthogonality in the following form

$$
\sum_\mu d_\mu \chi_\mu(h) = \sum_\mu \chi_{\mu^*}(e) \chi_\mu(h) = |G| \delta_{h,e} .
\tag{B.23}
$$

We use this to evaluate

$$
\begin{aligned}
\sum_\mu F_{\delta b}(\mu) &= \frac{1}{|G|} \sum_{h \in G} \sum_\mu d_\mu \chi_\mu(h) F_{\delta b}([h]) \\
&= F_{\delta b}([e]) = \mathbb{1} .
\end{aligned}
\tag{B.24}
$$

$F_{\delta b}([e])$ is the identity operator because it neither shifts nor projects any links. $\qquad\square$

## Non-commutation relation

We claim that two intersecting central ribbons, corresponding to $\gamma$ and $\gamma'$, fail to commute. Again, by intersecting we mean that $\gamma$ and $\gamma'$ intersect at one point, as all other cases can be deformed to this case assuming that the path of deformation does not pass through any charges.

**Theorem B.7.** *The central ribbons $F_\gamma([h])$ and $F_{\gamma'}([k])$ fail to commute. Here $\gamma$ and $\gamma'$ denote paths on the lattice that intersect exactly once.*

*Proof.* One way to show this is to just directly commute the two ribbons past each other using the left and right joint relations. We will instead take the following route because it has a nicer interpretation. Letting $|\psi\rangle$ be any state on the lattice, we will show this by directly demonstrating that $F_\gamma([h]) F_{\gamma'}([k]) |\psi\rangle$ is not equal to $F_{\gamma'}([k]) F_\gamma([h]) |\psi\rangle$.

Note that the only places where the operators can possibly not commute are on the two links, $a$ and $b$, located at the intersection of the two ribbons, as in the following diagram. Here both $a$ and $b$ are located on the incoming spines of their respective ribbons.

We will also label the elements on these two links as $a$ and $b$, so that $|a\rangle |b\rangle$ is the part of $|\psi\rangle$ that we are concerned with. Without loss of generality we take $a$ to be located on the spine of $\gamma'$, and $b$ to be located on the spine of $\gamma$. We can also break up the paths as $\gamma = \gamma_1 a \gamma_3$ and $\gamma' = \gamma_2 b \gamma_4$, i.e. into a piece that comes before the link and piece that comes after the link. We let $g_{\gamma_1}$ denote the product of all the elements on the links in $\gamma_1$ in the order traversed by the ribbon, and similarly we let $g_{\gamma_2}$ denote the same quantity for $\gamma_2$. Then

$$
\begin{aligned}
F_{\gamma'}([k]) F_\gamma([h]) |a\rangle |b\rangle &= F_{\gamma'}([k]) \left| b^{-1} g_{\gamma_1}^{-1} h g_{\gamma_1} b a \right\rangle |b\rangle \\
&= \left| b^{-1} g_{\gamma_1}^{-1} h g_{\gamma_1} b a \right\rangle \left| a^{-1} b^{-1} g_{\gamma_1}^{-1} h^{-1} g_{\gamma_1} b g_{\gamma_2}^{-1} k g_{\gamma_2} b^{-1} g_{\gamma_1}^{-1} h g_{\gamma_1} b a b \right\rangle \\
&= \left| b^{-1} h b a \right\rangle \left| a^{-1} b^{-1} h^{-1} b k b^{-1} h b a b \right\rangle ,
\end{aligned}
\tag{B.25}
$$

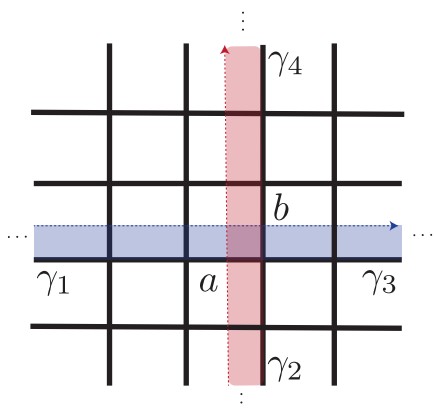

Figure 14: Two ribbons that intersect exactly once. We will call the links bordering the intersection point $a$ and $b$, so that the path of the blue ribbon can be written in the form $\gamma = \gamma_1 a \gamma_3$, and the path of the red ribbon can be written in the form $\gamma' = \gamma_2 b \gamma_4$.

where to get to the last line we took $h \to g_{\gamma_1}^{-1} h g_{\gamma_1}$ and $k \to g_{\gamma_2}^{-1} k g_{\gamma_2}$. Similarly,

$$
\begin{aligned}
F_\gamma([h]) F_{\gamma'}([k]) |a\rangle |b\rangle &= F_\gamma([h]) |a\rangle \left| a^{-1} g_{\gamma_2}^{-1} k g_{\gamma_2} a b \right\rangle \\
&= \left| b^{-1} a^{-1} g_{\gamma_2}^{-1} k g_{\gamma_2} a g_{\gamma_1}^{-1} h g_{\gamma_1} a^{-1} g_{\gamma_2}^{-1} k g_{\gamma_2} a b a \right\rangle \left| a^{-1} g_{\gamma_2}^{-1} k g_{\gamma_2} a b \right\rangle \\
&= \left| b^{-1} a^{-1} k a h a^{-1} k a b a \right\rangle \left| a^{-1} k a b \right\rangle,
\end{aligned}
\tag{B.26}
$$

where we again took $h \to g_{\gamma_1}^{-1} h g_{\gamma_1}$ and $k \to g_{\gamma_2}^{-1} k g_{\gamma_2}$. $\qquad\square$

These two expressions are not equal. We can see this clearly by taking $G$ to be some continuous Lie group. In that case, we can write $h = e^{i\epsilon H} \approx 1 + i\epsilon H$ and $k = e^{i\epsilon K} \approx 1 + i\epsilon K$. Then

$$
F_c^{[h]}(\gamma) F_c^{[k]}(\gamma') |a\rangle |b\rangle = \left| b^{-1} h b a \right\rangle \left| a^{-1} k a b + \epsilon^2 g_H \right\rangle,
\tag{B.27}
$$

for some $g_H$, and similarly

$$
F_c^{[k]}(\gamma') F_c^{[h]}(\gamma) |a\rangle |b\rangle = \left| b^{-1} h b a + \epsilon^2 g_K \right\rangle \left| a^{-1} k a b \right\rangle,
\tag{B.28}
$$

for some $g_K$.

## C  General subalgebras

As stated in Section 2.4, in general when we consider a subregion $b$ of the lattice, the sub-algebra we associate to that $b$ is *not* just the set of physical operators that act trivially on its complement $\bar{b}$. It is instead a larger subalgebra which includes some operators that act non-trivially on $\bar{b}$. In this appendix, we describe this subalgebra and its center in detail.

The fundamental principle is that these subalgebras do correspond to subregions of the reduced lattice. Once we convert the reduced lattice back to the full lattice, that subalgebra turns out to be this novel type, not the one we would most naturally ascribe to a subregion. That said, this is indeed a natural subalgebra to assign to (disconnected) $b$ if you were interested in having the center of that subalgebra include the operator that measures the net electric flux out of $b$.

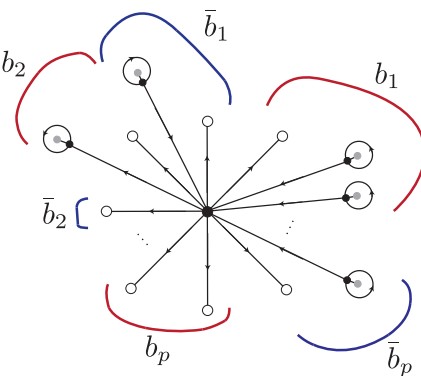

Figure 15: An example of the setup for this section.

The summary description of these subalgebras is as follows. Say $b$ has $n$ connected regions. These subalgebras are the algebraic union of the natural subalgebra associated to each connected region, along with a network of Wilson lines connecting all of these subregions in pairs. These Wilson lines allow us to compare the electric flux leaving each subregion. The center is generated by "fused ribbon" operators that measure the total electric flux leaving all the subregions.

## Subregion subalgebras on the reduced lattice

We use the notation of Section 3 for the parts of the reduced lattice. It has $n$ vertices $v_1 \ldots v_n$, with corresponding links $\ell_1 \ldots \ell_n$ oriented out of the central vertex; together these links form the set $\mathfrak{f}_\partial$. There are also $m$ lollipops $l_1 \ldots l_m$, which form the set $\mathfrak{f}_{\text{lol}}$. In our figures, we will label $\ell_r$ with $r$ and $l_r$ with $r'$. Together, $\mathfrak{f} = \mathfrak{f}_\partial \cup \mathfrak{f}_{\text{lol}}$ consists of $\mathfrak{f}_1 \ldots \mathfrak{f}_{n+m}$, numbered clockwise around the central vertex.

The set of subregions we minimize over in the TN are subsets $b \subseteq \mathfrak{f}$. We split $b$ into contiguous sets $b_1 \ldots b_{\mathsf{p}}$. By convention, $b_1 = \mathfrak{f}_1 \cup \ldots \mathfrak{f}_{|b_1|}$. Similarly, we split the complement $\overline{b}$ into $\overline{b}_1 \ldots \overline{b}_{\mathsf{p}}$, also arranged clockwise.

## Subalgebras for each component

For a factor $\mathfrak{f}_i \in \mathfrak{f}_\partial$, the algebra $\mathcal{A}_{\mathfrak{f}_i}$ (projected to the gauge-invariant subspace) is the set of operators that act on the boundary vertex,

$$\mathcal{A}_{\mathfrak{f}_i} = \left\{ R_{\mathfrak{f}_i}(h) | h \in G \right\}'' = \bigoplus_\mu \mathcal{L}(\mathcal{H}_\mu). \tag{C.1}$$

Note that, when we factorize, $\mathfrak{f}_i$ will actually be the Hilbert space of the entire link, but the gauge-invariant algebra on a single link is that on just one end of the link.

For a factor $\mathfrak{f}_i \in \mathfrak{f}_{\text{lol}}$, the algebra consists of the closed ribbon operator that measures the quantum double charge of the matter, and the operator that measures the irrep of the stem. Call the algebra of closed ribbons $\mathcal{A}_{i,\text{cl}}$; the details will not be important to us. To describe the rest, name the three links on the lollipop $\int^i, \lfloor_i, \langle_i$ (the stem, boundary and heart),

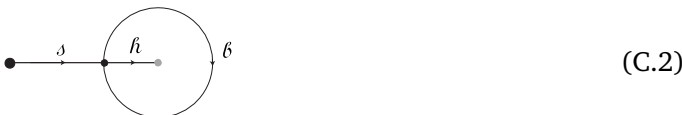

$$\tag{C.2}$$

The other set of operators measures the total electric flux leaving the lollipop,

$$L_{\mathfrak{f}^i}(\mu) := \frac{1}{|G|} \sum_h \chi_\mu(h) L_{\mathfrak{f}^i}(h).$$  (C.3)

We can use Gauss's law to rewrite the electric operator as a ribbon

$$L_{\mathfrak{f}^i}(\mu) = R_{\mathfrak{f}^i}(\mu^*) = \frac{1}{|G|} \sum_{h \in G} \chi_\mu(h) T_{\lfloor_i, \mathfrak{f}^i}(h) T_{\langle_i, \mathfrak{f}^i}(h) T_{\bar{\lfloor}_i, \mathfrak{f}^i}(h)$$

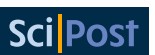
$$=$$  (C.4)

Along with $\mathcal{A}_{i,\mathrm{cl}}$, these operators generate $\mathcal{A}_{\mathfrak{f}_i}$.

Now consider a contiguous region $b_i$. Apart from the operators on each factor, there are now also ribbon operators $F_\gamma(h, g)$, $\gamma \subseteq b_i$,

$$\mathcal{A}_{b_i} = \left\{ F_\gamma(h, g) \big| \gamma \subseteq b_i \right\}'' \bigvee_{\mathfrak{f} \in b_i} \mathcal{A}_\mathfrak{f}.$$  (C.5)

Under lattice deformations, these contiguous regions map to those considered in Section 2.4, and the center $\mathcal{Z}_{b_i}$ is generated by irrep ribbons $F_{\delta b_i}(\mu)$ of the sort defined there.

**The full subalgebra**

With multiple components, the algebra $\mathcal{A}_b$ is made out of three types of operators.

1. The additive algebra $\mathcal{A}_{b_1} \vee \cdots \vee \mathcal{A}_{b_\mathsf{p}}$.

2. The Wilson lines $F_{\bar{\ell}_i \ell_j}(e, g)$, $\forall \{\mathfrak{f}_i, \mathfrak{f}_j\} \subseteq \mathfrak{f}_\partial \cap b$. An example is shown in blue in Figure 16.

3. The transported shifts $T_{\mathfrak{f}^i, \bar{\ell}_1}(h)$ acting on the stem of lollipop $\mathfrak{f}_i$, with group element transported from $v_1$. An example is shown in green in Figure 16. As in (C.4), we can transport it to shift $\langle_i$ and $\lfloor_i$ (on both ends) similarly to (C.4).

The crucial fact is that we have arbitrary ribbons in each component, but only a subclass of those that cross between components. This subclass is the one that doesn't contain any electric action; we call these the magnetic ribbon operators and the others electric ribbon operators.[37] These can be used to parallel-transport all electric actions to any boundary vertex in $b$. We denote a group element parallel transported along $\gamma$ as $h_\gamma$, so

$$L_\ell(h_\gamma) = T_{\gamma, \ell}(h).$$  (C.6)

The central operator is the fused ribbon operator

$$F_{\delta b}(\mu) = \frac{1}{|G|} \sum_{h \in G} \chi_\mu(h) \prod_{\mathfrak{f}_i \in b} L_{\mathfrak{f}_i}(h).$$  (C.7)

---

[37]A more accurate name would be 'non–purely magnetic,' but we opt for the shorter name.

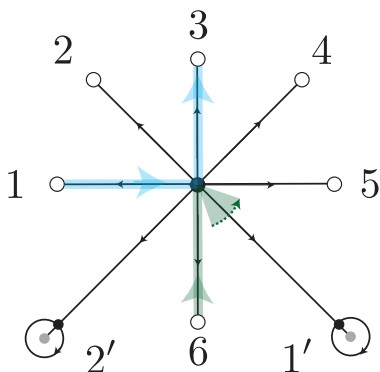

Figure 16: Examples of non-local operators in a region consisting of the links numbered $1, 3, 6$ and the lollipop labelled $1'$. In blue, a Wilson line from 1 to 3. In green, a shift acting on the stem of $1'$, transported to the boundary end of link 6.

If $\mathfrak{f}_i \in \mathfrak{f}_{\text{lol}}$, then $L_{\mathfrak{f}_i} = L_{\int^i}$. The reason we call it a fused ribbon operator is as follows. We can see from the definition that it measures the total electric flux leaving $b$. This irrep arises in the fusion of the fluxes leaving individual components, $\mu_1 \otimes \dots \mu_{\mathsf{p}} \to \mu$. This can be seen at the level of the operator by defining the ribbons in each component

$$F_{\delta b_r; \nu_1}(h, 1) := \prod_{\mathfrak{f}_j \in b_r} L_{\mathfrak{f}_j}(h_{\bar{\ell}_1}). \tag{C.8}$$

Here, 1 denotes the function $1(h) = 1$. We can use the fact that the character is a class function to parallel transport the group element to $\nu_1$ in (C.7); as a result, we can write

$$F_{\delta b}(\mu) = \frac{1}{|G|} \sum_{h \in G} \chi_\mu(h) \prod_{r=1}^{\mathsf{p}} F_{\delta b_r; \nu_1}(h, 1), \qquad F_{\delta b_r}(\mu) = \frac{1}{|G|} \sum_{h \in G} \chi_\mu(h) F_{\delta b_r; \nu_1}(h). \tag{C.9}$$

This is the sense in which the central operator for $b$ is a fusion of the central ribbons in the separate components.

### Mapping to the original lattice

We are also interested in the algebra and center on the original lattice. It will be instructive to begin with some examples.

### Basic examples

First, consider $n = 4, m = 0$, and take $b = \ell_1 \cup \ell_3$, as shown in Figure 17. Do the "add a vertex" move, such that $\ell_1$ and $\ell_3$ are separated by the new link, which we call $\ell_0$. The constraints on the new lattice are $L_{\ell_1}(h) L_{\ell_4}(h) L_{\ell_0}(h) = L_{\ell_2}(h) L_{\ell_3}(h) R_{\ell_0}(h) = \mathbb{1}$, and so the electric operator on this new link is neither in $\mathcal{A}_b$ nor in $\mathcal{A}_{\bar{b}}$. As a result, the magnetic operator is in both algebras (or rather, since it is not gauge-invariant on its own, it appears as a component of some operator in both algebras). This is important, since this magnetic operator is required to parallel transport a shift on $\ell_3$ to $\nu_1$ or parallel transport a shift on $\ell_4$ to $\nu_2$. Mathematically,

$$\mathcal{A}_b = \mathcal{A}_{\ell_1} \vee \mathcal{A}_{\ell_3} \vee \left\{ F_{\bar{\ell}_1 \ell_0 \ell_3}(e, g) \right\}, \qquad \mathcal{A}_{\bar{b}} = \mathcal{A}'_b = \mathcal{A}_{\ell_2} \vee \mathcal{A}_{\ell_4} \vee \left\{ F_{\bar{\ell}_2 \ell_0 \ell_4}(e, g) \right\}. \tag{C.10}$$

We can also derive this explicitly using the unitaries in appendix A; the Wilson line in the reduced lattice maps to that in the bigger lattice. To find the central operator in the new lattice, we write the one on the reduced lattice as in (C.9), with $F_{\delta \ell_1; \nu_1} = R_{\ell_1}(h)$ and $F_{\delta \ell_3; \nu_1} = T_{\ell_3, \bar{\ell}_1}(h)$.

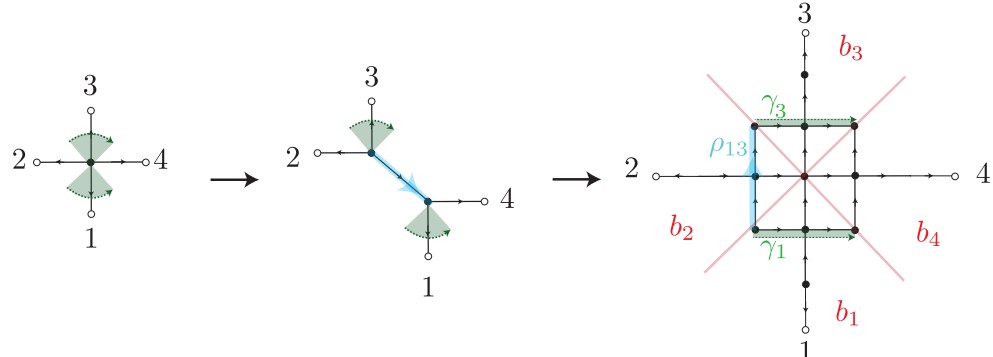

Figure 17: An example where the reduced lattice consists of four links.

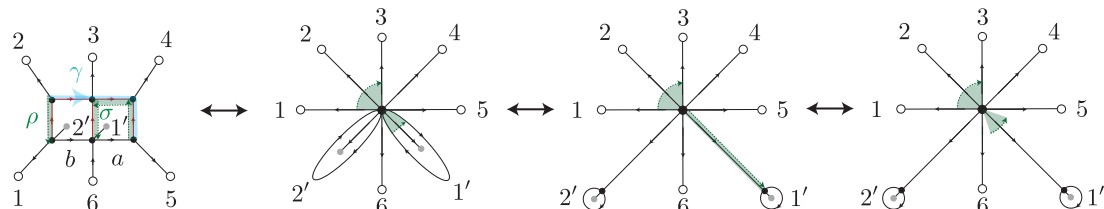

Figure 18: Second example.

This generalises to larger lattices; we define $b_{1\ldots4}$ to be regions on the larger lattice as in Figure 17. We include a Wilson line on the path $\rho_{13}$ that connects $b_1$ to $b_3$. The central operator straightforwardly generalises (C.9)

$$F_{12}^{\mu} = \frac{1}{|G|} \sum_{h \in G} \chi_{\mu}(h) F_{\delta b_1}(h, 1) F_{\delta b_2}\left(T_{\rho_{13}}(h), 1\right). \tag{C.11}$$

As a second example, consider two matter degrees of freedom, as shown in Figure 18. $b$ consists of $b_1 = \ell_1 \cup \ell_2$ and $b_2 = l_1$, labelled $1'$ in the figure. For $F_{\delta b_2; \nu_1}$ in the fused ribbon operator, we use (C.4) to deform it as in the rightmost arrow of the figure. This is the ribbon deformation shown in the rightmost arrow of Figure 18. Following this through the lattice deformations, we find

$$F_{\delta b}^{\mu} = \frac{1}{|G|} \sum_{h \in G} \chi_{\mu}(h) F_{\delta b_1}(h, 1) F_{\delta b_2}\left(h_\gamma, 1\right). \tag{C.12}$$

First we remove a vertex (second arrow) and add five vertices (all but one of the ones adjacent to the red links). The add/remove a vertex move acts on a transported shift by transporting a shift to the same origin vertex ($\nu_1$ in this case); the path of the parallel transport includes the original path along with a subset of the new links.

This can be found without explicit computation. Remember that the central operator is a fusion of central ribbon operators, all parallel-transported to the same vertex. Notice that $b$ on the reduced lattice separates $\overline{b}_1 = \ell_3 \cup \ell_4 \cup \ell_5$ from $\overline{b}_2 = \ell_6 \cup l_2$ (labelled $2'$ in the figure). Similar to how electric ribbons can't cross from $b_1$ to $b_2$, they can't cross from $\overline{b}_1$ to $\overline{b}_2$. Thus, (a) the central ribbons that make up the fused ribbon must surround $b_1$, $b_2$ respectively without surrounding anything else, and (b) the parallel-transport path should separate $\overline{b}_1$ from $\overline{b}_2$. The central ribbon surrounding $b_1$ is no different from that considered in Section 2.4. To find the one surrounding $b_2$, we first note that a central ribbon must end on links which border only one plaquette in $P_{\text{bulk}}$. There are three such ribbons surrounding $l_1$ on the larger lattice,

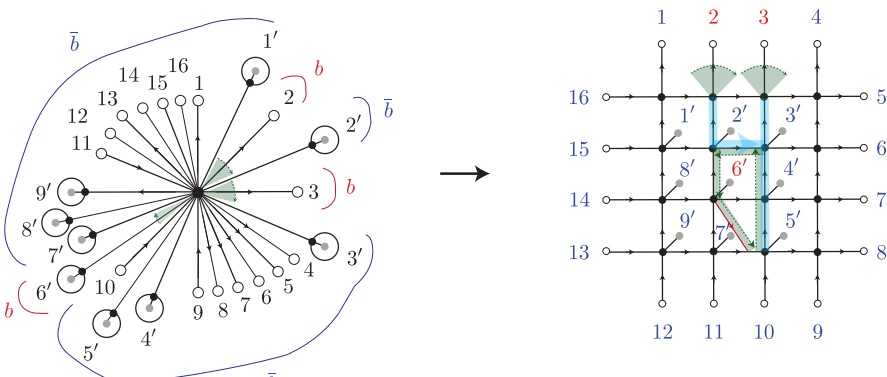

Figure 19: A big lattice where $b$ contains a matter degree of freedom deep in the lattice, along with its corresponding reduced lattice.

$\sigma$ in Figure 18 along with the following ribbons,

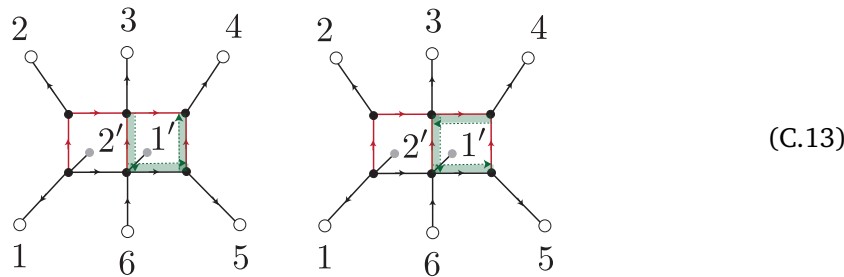
$$(C.13)$$

but only one of them does not separate $\ell_5$ from $\ell_{1...3}$. Here, it is important to remember that the green ribbon shifts the group element on the link it ends on, and that link cannot be used for parallel-transporting in the complement region. Similarly, there is only one path from the end of this ribbon to $v_1$ that separates $\overline{b}_1$ from $\overline{b}_2$.

**A complicated example**

Finally, we should consider the possibility that the matter degree of freedom is deep in the original lattice. Unlike the above case, the central ribbon passing through the matter link cannot live on just one plaquette, since commutation with the plaquette constraints requires that central ribbons end on links bordering only one element of $P_{\text{bulk}}$. On the reduced lattice, it does live on just the one plaquette in the lollipop. When we go back to the original lattice, we have to use (A.15) to extend the ribbon while modifying the lattice.

Let us see an example, shown in Figure 19. On the reduced lattice, $b$ has three components, $b_1 = \ell_2, b_2 = \ell_3$ and $b_3 = l_6$.[38] Similarly, $\overline{b}$ has three components $\overline{b}_1 = l_2$, $\overline{b}_2 = l_3 \ldots \ell_{10}$ and $\overline{b}_3 = \{l_7 \ldots \ell_1\}$. The relations between the numberings was found by explicit lattice transformations (not shown here, in order to preserve the reader's sanity). Note that the correspondence is not unique.

Because $\ell_2$ and $\ell_3$ separate $l_2$ from the rest of $\overline{b}$, the Wilson line connecting them must go *around* the corresponding matter vertex. This tells us how to parallel transport $F_{\delta b_2}(h, 1) = L_3(h)$ to $v_2$ to fuse the ribbons.

---

[38]The numbering is off from previous conventions by 1, due to a clerical error that propagated till it was too late to change it. Please bear with us.

The situation with $b_3 = l_6$ is more interesting. We have to be very careful that the central ribbon avoids all matter links, while ending at the right place on the boundary. To find the right end-point, note first that $l_6$ lies between $\ell_{10}$ and $l_7$. So the ribbon should separate these two. However, it must not separate $\ell_{10}$ and $l_5$. Visually, it is clear that the only ribbon which satisfies this property must change orientation! The simplest way to deal with this to add new links like the red on in Figure 19. Then the central ribbon as drawn has the right properties.[39] The central operator is the fusion of the three ribbons, parallel-transported along the purple paths.

Finally, there is the issue of the physical interpretation of the central ribbon operator for $b_3$, $F_{\delta b_3}(\mu)$. In the reduced lattice, it clearly measures the electric flux leaving $b_3$; but this is not so clear in the original lattice.

**The lessons**

The algorithm to map the central operator is as follows. First, remember that there is a unique bijective correspondence between matter degrees of freedom and boundary vertices on the two lattices. Labelling these physical degrees of freedom in the same way on both lattices, $\mathcal{A}_b$ on the original lattice is not the algebra of all operators in some subregion. Each $\mathcal{A}_{b_i}$ is, but there are also some magnetic operators connecting the different components. The set of extra magnetic operators are the ones that don't commute with electric ribbons that cross from $\overline{b}_i$ to $\overline{b}_j$.

The central ribbon for each component can be fixed using the topological properties as above. We might need to add a small number of new links to easily describe the corresponding operator. The area operator is the fusion, like in (C.9), of ribbon operators surrounding each component $b_i$.

**A subtlety with our tensor network**

Let us look again at Figure 19, keeping in mind that we are building a tensor network for the boundary links. Suppose we define the tensor network using the lattice deformations shown there and find upon minimization that the entanglement wedge of the boundary region $B = \ell_2 \cup \ell_3$ is the region $b$. Notice the following oddity: the region $\overline{B}$ is a contiguous set of boundary links in the original lattice (and it is also contiguous in the reduced lattice once we project out the lollipops). However, two subregions of $\overline{B}$ are topologically separated by the dressing of the matter $l_6$.

This never happens in AdS/CFT. If $\overline{B}$ is an interval, then any two points in $\overline{B}$ can be connected through the bulk by a path that doesn't leave the entanglement wedge. In particular, in the continuum TQFT description, all Wilson lines stretching between any pair of points in $\overline{B}$ which are homotopic to a sub-interval of $\overline{B}$ are included in the entanglement wedge subalgebra. (They measure things like two-point functions and entanglement of subintervals [28,29].) So, our tensor network is a bad toy model for gravity if subalgebras like this form the entanglement wedge. It will be important to address this in future work.

There is a second oddity. Suppose, in the example of Figure 19, that the lollipop $l_2$ was also included in $b$. In that case, even though the plaquettes containing $l_2$ and $l_6$ are adjacent to each other, the algebra does not contain ribbons stretching from $l_2$ to $l_6$. The central ribbon

---

[39]Alternatively, one could mathematically define a twisted ribbon on the original lattice. Just use the relation between left actions and right actions on the links being shifted twice, along with flatness of the new plaquettes bounded by red links.

operator is then

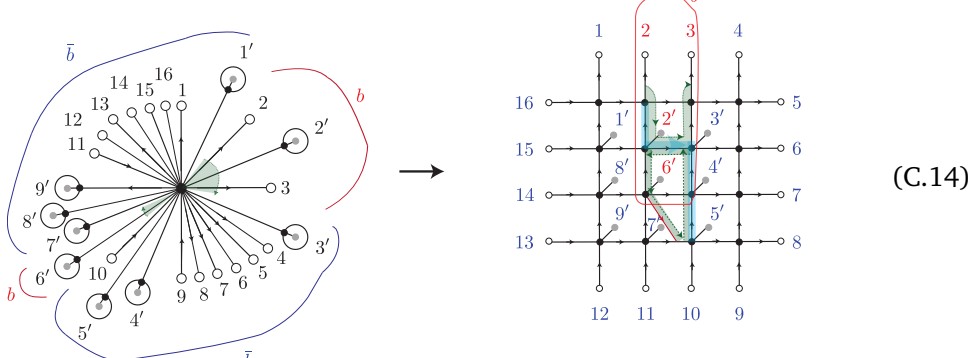

$$(C.14)$$

Despite the bulk region being contiguous, the algebra is still a fused subalgebra of two subalgebras dressed to different parts of the boundary.

We look forward to dealing with these subtleties in future work. For now, let us note that we can get around this by restricting our matter to be electric. That means that the flatness constraints are not modified. In the example of Figure 19 (where we return to the case where $l_2$ is excluded from $b$), note that the green ribbon around $l_6$ shifts the links below it twice, once with a left-multiplication and once with a right-multiplication. If the matter doesn't have magnetic charge, the parallel transport around the plaquette containing $l_6$ is equivalent to a parallel transport along the link itself (by the flatness constraint). Now, note that for any link $\ell$

$$L_\ell(h)T_{\ell,\bar{\ell}}(h)\,|g\rangle_\ell = \left|hg(g^{-1}h^{-1}g)\right\rangle = |g\rangle\,.\tag{C.15}$$

Thus, the green ribbon only acts as a shift on the matter,

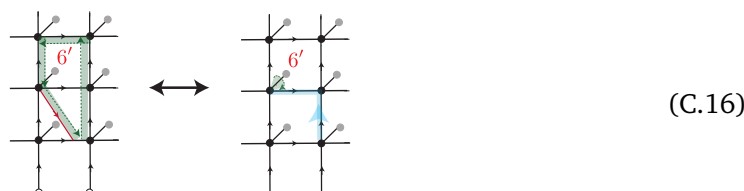

$$(C.16)$$

The only role of its spine is to parallel transport the group element. But, because of flatness, the exact path of parallel transport is immaterial; only the origin (in this case, $\ell_2$) matters. Thus, the fused ribbon can be deformed to completely avoid the plaquette containing $l_7$, opening up a path for a ribbon to cross between the two components of $\bar{b}$,

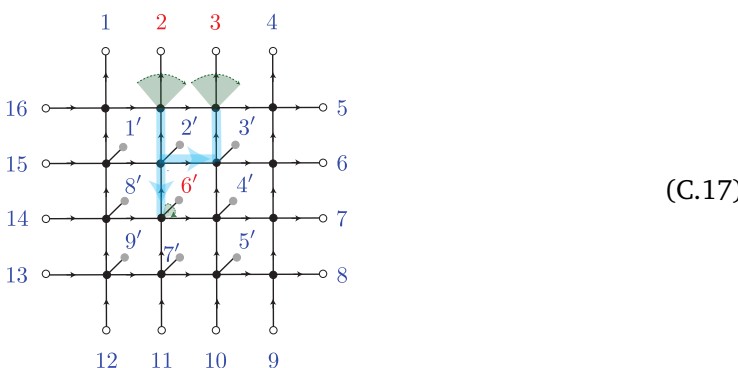

$$(C.17)$$

Let us sketch the general argument. For general matter, the reason that $\mathcal{A}_b$ does not include general ribbons from $\ell_2$ to $l_6$ is that they are not adjacent on the reduced lattice. But this is somewhat arbitrary, since different lattice deformations can result in different reduced lattices.

The ordering of the boundary vertices is of course fixed, but the ordering of the lollipops is not so. We state without proof that we can move the lollipop around on the reduced lattice with a braiding unitary.[40] A braiding of a quantum double charge (which general matter that modifies both types of constraints carries) and electric charge is not trivial, but the braiding of electric charges with each other is [35]. Denoting quantum double charge with R, $R \otimes \mu \neq \mu \otimes R$.[41] Thus, if the matter is purely electric, then we can braid all the matter lollipops to be adjacent to one of the boundary links we began with, and the unitary is trivial.

## D  Tripartite information in the DG model

We show that there are four-party states such that the tripartite information (80) satisfies (81). Consider a reduced lattice with four links $b_{1...4}$, in a state $|\psi\rangle$ such that the states $F_{\delta b_i}(\mu)|\psi\rangle$ are factorised between $b_i$ and the complement, so that all the entanglement entropy is edge mode entropy.

Assume that the four links have fixed irreps $\mu_{1...4}$. This is a reasonable approximation for holography, where we can make the fluctuations of four non-overlapping extremal surfaces small to leading order in $G_N$.

Let us then calculate a bound on the tripartite information. The positive contributions are

$$S_r = \log d_{\mu_r}, \quad r = 1, 2, 3, \qquad S_{123} = S_4 = \log d_{\mu_4}. \tag{D.1}$$

The negative contributions can be bounded above as

$$S_{r2} \leq \log d_{\mu_r} d_{\mu_2}, \qquad r = 1, 3, 4, \tag{D.2}$$

since the fusion of the two irreps $\mu_2, \mu_r$ necessarily gives a subspace of $\mu_r \otimes \mu_2$. Using $S_{13} = S_{24}$, the tripartite information can be bounded below

$$I_3(b_1 : b_2 : b_3) = \sum_{r=1}^{4} S_r - \sum_{r=1,3,4} S_{2r} \geq -2\log d_{\mu_2}. \tag{D.3}$$

Thus, (81) must be satisfied, as long as $I_3$ is negative.

We have to fine-tune the group and class of states to make $I_3 \leq 0$ and match holography. We have not done so in this work, and hope to do so in later work. However, this argument shows that once we make the relevant choices to make $I_3 \leq 0$, the tripartite information automatically has the right limit.

## E  Uniqueness of the factorization map

Our factorization map might seem like an obvious consequence of the gauge-theoretic description, but we should be more careful. This obviousness comes from the fact that we defined the physical theory using unphysical gauge degrees of freedom, and the factorization map consists of re-introducing some of these. However, this is not a unique affair; for example, there are dualities in which the same physical system can arise from gauging different groups on the two sides of the duality.

---

[40]An example of this can be found in the appendix of [43] for an example.

[41]This is not the tensor product physicists are used to. The notation $\boxtimes$ is common in the Hopf algebra literature for this abstract tensor product.

Furthermore, in the bipartite case, there is in fact an irreducible ambiguity. The center is a commutative algebra, and so there is, a priori, no constraint on the edge modes [40]. [40] also found that, in JT gravity, the inclusion of matter resolves this ambiguity. [54] showed that, once we fix the edge mode von Neumann entropy, the entanglement spectrum of each $|\chi_\alpha\rangle$ in (84) is completely fixed by the form of holographic Rényi entropies. In this section, we show a similar uniqueness theorem for the edge modes we have introduced in this work. Our basic tool is the consistency of the multi-party factorization.

The fusion multiplicities of the irreps are defined by $\mu \otimes \nu = \oplus_\rho \rho^{\oplus N_{\mu\nu}^\rho}$. The identity irrep $\mathbf{1}$ satisfies $N_{\mu\mathbf{1}}^\nu = \delta_\mu^\nu$. There is an important relation between the fusion multiplicities $N_{\mu\nu}^\rho$ and the quantum dimensions, see [79] for a physicists' explanation,

**Proposition E.1.** [80,81] Regard the fusion multiplicities with one index $\mu$ fixed as a matrix,

$$(\mathbb{N}_\mu)_\nu^\rho := N_{\mu\nu}^\rho. \tag{E.1}$$

Then, the quantum dimension is the largest eigenvalue of this matrix. In other words, there is a set $(a_\rho)$, such that

$$(\mathbb{N}_\mu a)_\nu = d_\mu a_\nu, \quad \text{and} \quad \lim_{n\to\infty}(\mathbb{N}_\mu^n)_\nu^\rho = d_\mu^n a_\nu a^\rho. \tag{E.2}$$

Our uniqueness theorem is the following:

**Theorem E.2.** *Consider the DG model (without matter) on a reduced lattice for $D^2$ with m links, and call its Hilbert space $\mathcal{H}_m$. Assume the existence of a factorization map $J : \mathcal{H}_m \hookrightarrow \mathcal{H}_\ell^{\otimes m}$, for some $\mathcal{H}_\ell$, satisfying the following two properties:*

1. *For any subset b, define $\mathcal{H}_b$ as the tensor factor of $\mathcal{H}_\ell^{\otimes m}$ on which $\mathcal{A}_b$ lives. The first condition is that for any state $|\psi\rangle \in \mathcal{H}_m$,*

$$J|\psi\rangle = \sum_\mu F_{\delta b}(\mu)|\psi\rangle \otimes UU'|\chi_\mu\rangle, \qquad U \in \mathcal{L}(\mathcal{H}_b),\ U' \in \mathcal{L}(\mathcal{H}_{\bar{b}}), \tag{E.3}$$

   *where $\mu$ is valued in the irreps of G and $|\chi_\mu\rangle$ is a state in an auxiliary bipartite Hilbert space $\mathcal{H}_{\mu,l} \otimes \mathcal{H}_{\mu,r}$, such that both the state and auxiliary Hilbert space are completely fixed by $\mu$. $U, U'$ are isometries that embed this abstract bipartite state into the $\mathcal{H}_\ell^{\otimes m}$.*

2. *$|\chi_\mathbf{1}\rangle$ is a factorized state.*

   *Then,*

$$|\chi_\mu\rangle = \frac{1}{\sqrt{d_\mu}} \sum_{i=1}^{d_\mu} |i\rangle|i\rangle. \tag{E.4}$$

*Proof.* Since our final aim is to prove a statement about the edge mode state $|\chi_\mu\rangle$, we can choose a specific lattice and state $|\psi\rangle$ that are most convenient for us.

Take a reduced lattice with $2n$ links. We will work in the limit $n \to \infty$. Call the links $\ell_{1...n}, \ell_{\tilde{1}...\tilde{n}}$, and let them all be oriented out of the central vertex. Denote by $\gamma_r$ the path $\bar{\ell}_r \ell_{\tilde{r}}$. Define the subregions $b := \ell_1 \cup \ldots \ell_n$ and $b_r := \ell_r \cup \ell_{\tilde{r}}$.

The state we work with is the following:

$$|\psi_0\rangle := \prod_{r=1}^n \left[ \sum_{g\in G} D_{ij}^\mu(g) F_{\gamma_r}(e,g) \right] |\mathbf{1}\rangle^{\otimes 2n}. \tag{E.5}$$

Here $i, j$ are some fixed indices in $\mathcal{H}_\mu$ that won't play a role below. It is useful to abstract away the lattice and keep track of only the fusion structure,

$$|\psi_0\rangle = \left| \begin{array}{c} \mu \underline{\quad 1 \quad} \mu \\ \mu \underline{\quad 2 \quad} \mu \\ \vdots \\ \mu \underline{\quad n \quad} \mu \end{array} \right\rangle . \tag{E.6}$$

The state can also be written as

$$\left| \begin{array}{c} \mu \underline{\quad 1 \quad} \mu \\ \mu \underline{\quad 2 \quad} \mu \\ \vdots \\ \mu \underline{\quad n \quad} \mu \end{array} \right\rangle = \sum_\nu \sqrt{\frac{d_\nu}{d_\mu^n}} \sum_{a=1}^{\left[\mathbb{N}_\mu^n\right]_\mu^\nu} \left| \begin{array}{c} \mu \diagdown 1 \quad \mu \\ \mu \diagup 2 \searrow \nu, a \diagup \mu \\ \vdots \\ \mu \diagup n \qquad \mu \end{array} \right\rangle , \tag{E.7}$$

$a$ denotes the various copies of $\nu$ that appear in the fusion, and each copy is orthogonal.

We calculate the reduced density matrix $\rho$, and more specifically $\operatorname{tr} \rho^q$, of the region $b$ in two ways, using the two representations of the state. We have implicitly used $J$ (acting on the original $2n$ links) to define $\rho$. Similarly, we define $\rho_r$ as the density matrix for $b_r$.

The first calculation follows simply from the fact that the irrep flowing out of the region $b_r$ is the identity irrep. This is because $F_{\gamma_r} \in \mathcal{A}_{b_r}$, and the operator $F_{\delta b_r}(\mu')$ that measures the irrep is in the center of $\mathcal{A}_{b_r}$, so

$$F_{\delta b_r}(\mu')|\psi_0\rangle = \delta_{\mu', \mathbf{1}}|\psi_0\rangle . \tag{E.8}$$

As a result, $\rho_r$ is a pure state, meaning that

$$\rho = \bigotimes_{r=1}^n \rho_{\ell_r} = \bigotimes_{r=1}^n U \chi_\mu U^\dagger . \tag{E.9}$$

Finally, this means that

$$\operatorname{tr} \rho^q = \left[ \operatorname{tr} \chi_\mu^q \right]^n \quad \Longrightarrow \quad [\operatorname{tr} \rho^q]^{1/n} = \operatorname{tr} \chi_\mu^q , \tag{E.10}$$

where we have defined $\chi_\mu$ as the reduced density matrix of $|\chi_\mu\rangle$ on one of the factors.

The second calculation uses the second representation of the state in (E.7). Using the fact that each total irrep $\nu$ and each copy of $\nu$ is orthogonal, the reduced density matrix is

$$\rho = \sum_\nu \frac{d_\nu \left[\mathbb{N}_\mu^n\right]_\mu^\nu}{d_\mu^n} \left\{ \sum_{a=1}^{\left[\mathbb{N}_\mu^n\right]_\mu^\nu} \frac{1}{\left[\mathbb{N}_\mu^n\right]_\mu^\nu} \left| \begin{array}{c} \mu \diagdown 1 \\ \mu \diagup 2 \searrow \nu, a \\ \vdots \\ \mu \diagup n \end{array} \right\rangle \left\langle \begin{array}{c} 1 \diagup \mu \\ \nu, a \searrow 2 \diagup \mu \\ \vdots \\ n \diagdown \mu \end{array} \right| \right\} \otimes \chi_\nu \tag{E.11}$$

$$:= \bigoplus_\nu p_\nu \rho_\nu \otimes U \chi_\nu U^\dagger ,$$

where $\rho_\nu$ is the object in square brackets and $p_\nu$ is the scalar prefactor. In the limit $n \to \infty$, we can use Proposition E.1 to simplify

$$p_\nu = \frac{d_\nu \left[\mathbb{N}_\mu^n\right]_\mu^\nu}{d_\mu^n} \xrightarrow{n \to \infty} d_\nu \mathsf{a}^\nu \mathsf{a}_\mu ,$$

$$\rho_\nu = \frac{\mathbb{1}}{\left[\mathbb{N}_\mu^n\right]_\mu^\nu} \xrightarrow{n \to \infty} \frac{\mathbb{1}}{d_\mu^n \mathsf{a}^\nu \mathsf{a}_\mu} . \tag{E.12}$$

Then,

$$\operatorname{tr}\rho^q = \sum_\nu p_\nu^q \operatorname{tr}\rho_\nu^q \operatorname{tr}\chi_\nu^q = \sum_\nu \frac{d_\nu^q (a^\nu a_\mu) \operatorname{tr}\chi_\nu^q}{d_\mu^{n(q-1)}} \quad \Longrightarrow \quad [\operatorname{tr}\rho^q]^{1/n} = \frac{1}{d_\mu^{q-1}}. \tag{E.13}$$

In the first line, we have used $\operatorname{tr}(\mathbb{1}_d/d)^q = d^{1-q}$, and in the second line we have used the fact that the numerator in the rightmost expression on the first line does not scale with $n$ in any way.

Comparing (E.10) and (E.13), we find

$$\operatorname{tr}\chi_\mu^q = \frac{1}{d_\mu^{q-1}} \quad \Longrightarrow \quad \chi_\mu = \frac{\mathbb{1}_{d_\mu}}{d_\mu}. \tag{E.14}$$

This means that $\left|\chi_\mu\right\rangle$ can be written as (E.4) in some basis, proving our claim. $\qquad\square$

We expect that a version of this theorem holds in a much more general class of topological field theories than doubly gauged models. Topological field theories (including the DG model) are believed to be specified by unitary fusion categories, see [61, 82] for an introduction. We have not assumed any particular braiding relations between the excitations, and so we can apply it to any unitary fusion category. In general, quantum dimensions need not be integers, and the trace on the edge mode Hilbert space might be a quantum trace. Our theorem allows this possibility (since we never used cyclicity of the trace). The result that generalizes is the fact that $\operatorname{tr}\chi_\mu^q = d_\mu^{1-q}$. There is some evidence that quantum traces are relevant in gravity, with edge modes satisfying this statement [30, 31, 83].

Finally, this result should be compared to the result of [40], which states that adding matter fixes the entropy of the edge modes. In our model, our result states that the entropy of the edge modes is fixed by the structure of the multipartite algebra. We can also consider our result to be a theorem fixing the edge mode entropy by incorporating matter effects, if we imagine that all the excitations live on bulk matter degrees of freedom.

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
