# Peer review of "Multipartite edge modes and tensor networks"

_SciPost Physics, doi:SciPost Phys. Core 7, 070 (2024)_

## Round 1 · Referee Report · Anonymous (Referee 1) · 2024-7-26

Strengths

The paper is very clearly written and contains substantial review.
Under the assumptions made in the paper, I believe the computations are correct.

Weaknesses

The assumptions which form the very foundation of the paper are not very justified from our point of view. Taking a topological wave-function as the starting point of constructing a holographic tensor network and then attempting to fit the result of entanglement entropy to the Ryu-Takayanagi result seems very difficult to justify.
The authors also appear to mix up target space geometry from world-volume geometry dependence which forms an important part of the paper.
We will explain these issues in detail in the report.

Report

The paper proposes a tensor network that follows from the wave-function of a class of topological phase. This network is proposed as a candidate for a holographic tensor network that remains independent of discretization, thus preserving diffeomorphism invariance. Additionally, the paper discusses incorporating matter into this topological wave-function and presents an area operator with properties aligning with the entanglement entropy results expected in semi-classical AdS/CFT.

The paper is one of the few papers in the field that explores graph independence in topological wave-functions to construct holographic tensor networks, an important direction for bulk reconstruction. However, there are significant issues that raise concerns about the main arguments of the construction.

Firstly, extensive arguments in Sections 1 and 4 are made to justify the choice of factorization map. The authors critique the area term and the subleading term in the entanglement entropy obtained from topological models like the Kitaev/Dijkgraaf-Witten models. They argue that allowing Wilson lines to take arbitrary shapes undermines the reduction of entanglement entropy to a counting of cuts. (we would return to this point later)

Furthermore, the paper contends that the sub-extensive term, a topological component in the entanglement entropy of a topological wave-function, is unexpected in the semi-classical AdS/CFT limit.

These points are central to the construction of the topological phase and the area operator throughout the paper. We have serious reservations about these arguments. The Ryu-Takayanagi formula is a semi-classical result applicable in the large N/large c limit. The paper attempts to match the entanglement entropy of a topological wavefunction to the RT result before taking this limit, which seems questionable for two reasons:

  1. The large N limit is intricate and affects the relative importance of terms. It's a complex interplay of variables, reaching its semiclassical value only at the saddle point from a superposition of geometries. Before defining a proper large N/large central charge limit, the dependence on the central charge in each term is unclear, making strong constraints on the presence or absence of a term seem poorly motivated.

  2. More critically, the entanglement entropy cannot be derived solely from the topological part of the tensor network. Topological wave-functions have zero correlation length, which contradicts the entanglement properties of a CFT. The entanglement must significantly depend on the boundary condition (defining a CFT) of the bulk theory/tensor network. The topological component lacks sufficient entanglement, which must ultimately come from the boundary condition providing infinite correlation length. Excluding the CFT raises doubts about any resemblance of the topological wave-function to AdS.

Thus, without considering a CFT and without a large N limit, attempting to match the semi-classical AdS/CFT results to such detail seems unreasonable.

Other concerns include:

  1. The assertion that discretization breaks symmetries is incorrect. You can see this fact not necessarily considering TQFTs. In the simplest situation, imagine doing quantum mechanics. Time evolution is given by the exp(i H t) and this is trivially equal to exp(i H t1) exp(i Ht2) ... exp( i H tn) for t1 + t2 + ... tn = t. This ``discretization'' does not break any time translation symmetry, as long as we sum over the complete set of states when we glue the operators together. In tensor networks, the breaking of symmetry occurs due to the truncation of bond dimensions so that we are not summing over all the states in any cutting and gluing. Therefore a tensor network that is graph independent is not really restricted to topological models (as is evident in the 0+1 dimensional case), even though topological model is an example where indeed symmetry is not broken under discretization. We however agree that since 3D gravity is topological, the study of topological wavefunctions is particularly pertinent.

  2. The paper places significant emphasis on the idea that entanglement depends on the counting of cuts, and yet in computations of AdS3 entanglement it involves insertion of Wilson lines in the bulk, which is independent its actual shape. This comparison is misleading.

Topological theories, including Chern-Simons theories, are independent of the background metric but not of the gauge connection, especially with a boundary. In AdS3/CFT2, where the bulk theory can be described as an SL2(R) Chern-Simons theory, the action DOES depend on the SL2(R) connection. This "target space" corresponds to the AdS3 metric, not the worldvolume metric of the gauge theory. The Wilson line’s shape in the worldvolume is unimportant, but its expectation values depend on the gauge connection encoding the target AdS3 metric. This has been well-discussed in literature, such as in arXiv:1603.07317 and arXiv:1612.06385. Distances in the topological bulk are encoded in the SL2(R) connections, indicating target space shapes rather than world-volume geometry. The authors seem to confuse target space geometry with world-volume geometry. In loop quantum gravity literature which consider similar models as the authors, the edge labels represent quantum group representations interpreted as geodesic lengths/dihedral angles (see also recent Virasoro TQFT discussions), which are gauge charges unrelated to the world-volume metric. Ribbon operator insertion would ultimately depend on the saddle point of these gauge labels in the semi-classical large c limit.

The difference from Chern Simons theories of generic gauge groups is that the target gauge connection does not have a geometric interpretation as the metric of some space -- they are just connections in some generic non-abelian groups. The expectation value of the wilson line still depends on the gauge connection, but do not depend on the world-volume metric, in precisely the same way as the SL2R CS theory.

For that matter, when the bulk topological theory is some generic CS theory, there is no reason to think that the gauge connection must contain some geometric interpretation as metric of some target space -- and yet the construction of the current paper takes up an arbitrary Dijkgraaf-Witten model and asked that one defines an area operator in these models that would mimic behaviour of the RT surface -- We find this starting point hard to justify and the complaint about shape independence in the world-volume metric irrelevant.

  1. The paper suggests that an important improvement to conventional tensor network model is that one relaxes the bi-paritite requirement and introduces multi-partite entanglement. This comment is misleading. Generic tensor networks inherently exhibit multi-partite entanglement, and entanglement entropy is not determined by counting graph cuts. Only in very specially engineered circumstance such as the HaPPY code would entanglement entropy by obtained by counting cuts in a graph. The notion of bi-partite has always been an approximation only. Entanglement entropy estimates the number of distillable Bell pairs, but the state itself is not necessarily entangled through Bell pairs. Therefore multipartite entanglement just seems to be the norm rather than the exception in the world of tensor networks, and it is by no means special to these topological models. For this reason, the ad hoc way the authors construct the tensor networks cannot be justified.

  2. TQFT tensor networks are a well-established field, with contributions from researchers like Vidal, Verstraete, Gu, and Wen. Many claims in this paper have appeared with more details in earlier literature.

  3. The review section is lengthy, and the distinction with string net models may be overstated, considering the paper’s focus on discrete groups.

  4. The tensor networks with matter in string net models have also been discussed in arXiv:1502.03433.

Recommendation

Reject

  • validity: low
  • significance: low
  • originality: ok
  • clarity: high
  • formatting: good
  • grammar: excellent

Author:  Ronak Soni  on 2024-09-25  [id 4801]

(in reply to Report 1 on 2024-07-26)
Category:
reply to objection

We thank the reviewer for their detailed engagement. We worry there might have been some miscommunication, so we will start by trying to clarify the big picture, and then we will respond to each point in order.

We completely agree with what the reviewer says in (4). In fact, it is perhaps the main point of the paper! The "space" in our tensor network should be understood as the worldvolume of the gauge theory. This is very different from previous holographic tensor networks, whose "space" is a slice of $AdS_3$. This paper can be understood as making this new choice and figuring out how it might be made to work.

Why make this choice, working with the worldvolume? Doing so allows us to solve problems with holographic tensor networks that we do not otherwise know how to solve, such as the problem of "commuting area operators.”

Having the tensors act in the worldvolume space raises many new subtleties, which we are attempting to address in the paper. One of the main subtleties has to do with the "area operator". If we were to straightforwardly use (say, random) tensors as is typical, then boundary entropies would satisfy a "Ryu-Takayanagi formula” in which the area operator counts an area in the worldvolume! This is what we do not want. We instead would like it to count area in the target space.

That’s the puzzle we want to solve: how do we make a tensor network that both (1) has the tensors act in the worldvolume space and (2) boundary entropies satisfy an RT-like formula with the "area" a feature of target space.

This paper takes a step towards solving this problem. Our tensor network has the tensors acting in the worldvolume space while at the same time the "area operator" is blind to the worldvolume metric. That said, we by no means completely solve the problem, because we are working with the toy model of a finite group. Closer to "the real thing" would be to use full SL2R CS, or Virasoro TQFT. However, that brings in even more complications and this paper already demonstrates a possible resolution to the problem. We agree that generic gauge groups do not have a geometric interpretation, and did not intend to make such a claim. We merely take discrete groups as a toy model for the string-net model built from the Virasoro TQFT.

  1. We agree that the large-$N$ limit is intricate and again we don't carefully take this limit, leaving it for future work. We took pains to clarify in our introduction that the 'correct' thing to do is to start from the Virasoro TQFT, and we are only exploring qualitative aspects that should be comparable to that. This is very much in the spirit of the tensor network literature, which is all about toy models. While previous tensor networks exhibit a quantum minimal surface formula, our tensor networks exhibit that as well as "non-commuting area operators."

    The one place where this issue is most pressing is in the argument about the undesirability of the total quantum dimension term. However, by virtue of it being the total quantum dimension, it scales with the central charge of the boundary CFT. We have added a note clarifying this point, in the relevant paragraph in section 4.1.

    In fact, in Virasoro TQFT, $S^0_0 = 0$ and so this term in fact diverges; this was discussed in [30]. An interesting subtlety is that in that case it diverges with the other sign.

  2. It is not true that the CFT has been excluded. Our open boundary conditions do not lead to a gapped boundary theory. Further, the central operator is in fact an operator in the boundary CFT, as can be seen by deforming it to the boundary. We discuss this in the last subsection of section 5. The vanishing of bulk correlation length is, rather than being a hindrance to the physics we need, exactly the source of the entropies agreeing with CFT calculations. We mentioned this around (4.2).

  3. We agree that restriction to a finite-dimensional Hilbert space is the important part that break symmetries, and have added a footnote in the introduction clarifying the point.

  4. As we mentioned above, we completely agree with the reviewer here, and we believe that is already reflected in the paper.

    One possible source of confusion here is the usage of the term "area"; while we call the operator that measures the total outgoing flux the area operator, we do not mean to imply that it necessarily has a geometric interpretation; we only mean that it is the operator that appears in e.g. (3.16). We have added notes at various places clarifying this point.

  5. There might be another miscommunication here. We do not claim that multipartite entanglement is the new ingredient, but multipartite edge modes. In the context of holographic TNs, edge modes are the link states that are added by the holographic map before projection with random tensors. Where previous work added only bipartite edge modes, we are forced to add multipartite edge modes. Since the area operator is definitionally an operator in the edge mode Hilbert space, this has important consequences for concepts like non-commuting areas. Heuristically, the multipartite entanglement in conventional RTNs is either due to the bulk state or due to the minimization, so that if we take a factorized bulk state, the multipartite entanglement is present due to the random tensors; whereas in our tensor network it is there before we act with random tensors. We have added a footnote in the introduction clarifying this.

  6. We agree that there are few substantially new condensed-matter results. We have added the citations that the reviewer has mentioned at various places, and thank the reviewer for these.

    However, we should clarify that our tensor networks are not the same as the TQFT tensor networks of previous literature, since we use a different factorisation map introduced by Dittrich, Delcamp and Riello.

    Also, the relationship is different between our tensor network and the TQFT. The reviewer mentions references where the tensor network is used to prepare a state of a TQFT. For us, the tensor network is a map from the Hilbert space of a TQFT to a different "boundary" Hilbert space.

  7. We wrote a lengthy review section since we wanted to introduce the technology of these models to high-energy researchers in an accessible way. This is (understandably) boring for experts on the lattice TQFT literature. We have added a sentence to the beginning of section 2 clarifying where the new results begin.

---

## Round 1 · Referee Report · Anonymous (Referee 2) · 2024-8-12

Report

Dear editor,

In this work, the authors study lattice gauge theory and its application to factorisation in gravitational systems. They point out a tension between conventional tensor models and factorisation properties of gravity. Using Kitaev's quantum double model, they then propose (in section 3) a novel class of tensor network models that have properties better in line with gravitational systems. This requires some rather dramatic changes compared to the usual set-up. The paper is well-written, rather lengthy and contains many ideas and arguments that would be of current interest to the community. The paper contains a lot of background and review material, and it is not always clear where the review ends, and the new results start. Moreover, a lot of the technical computations are contained in appendices, which makes the main text sometimes feel a bit qualitative. Also, due to its length, it is not easy for the reader to remember the notational conventions introduced in earlier sections. I believe the authors could still improve the readability along these lines.

I also have some specific comments on the work:

  • Around eq. (3.15) a reference to ref. 42 at that point would be useful.

  • It would be beneficial for the reader if the authors could provide more (physical) motivation for introducing the random tensors in the matter sectors in section 3.1, since at the moment this feels rather ad hoc. Relatedly, the result (3.34) is crucial, but seems to rely on quite a bit of technical assumptions to get there (random tensors, replica symmetry, saddle point approximation). Given the universality of equations like (3.34), it is a bit unsatisfactory that it requires this many assumptions. It would be good if the authors could comment on this.

  • It would help the reader if the authors would comment on how the F_\partial b(\mu) operators appear starting from eq (3.15). In the later equation just below eq (3.34), these F_\partial b(\mu) do not appear, being replaced by simple projection operator expressions instead. This also suggests that, although initially these central ribbon operators require an involved construction and discussion, they are actually extremely simple operators in the representation basis. It could be useful to mention this.

  • For explicitness, it might be good to put footnote 17 in the main text, and to refer to that paragraph (and notation) later on around eqns (3.19)-(3.27).

  • In eq (3.34), where is the contribution from b_2? It would also be good to explain where the \chi factors are coming from.

  • The results of this work seem to be related to ref. 30 in the following way. The authors there propose a factorisation map in 3d gravity that can be viewed as projecting onto the primary CFT state at the entanglement surface. This projection onto the zero-mode seems to be the same picture as the one the current authors present in terms of only the total flux being associated to the edge mode degrees of freedom. Do the authors have any additional insight on this possible direct relation?

  • Related to the previous comment, it might be worthwhile mentioning explicitly that the result of only keeping S_nonloc of eq (4.2) as the entanglement entropy matches in 3d gravity to the result first stated in ref. 27, and studied further in ref. 30.

After these clarifications and improvements, I think the paper would be suitable for publication.

Recommendation

Ask for minor revision

  • validity: high
  • significance: high
  • originality: high
  • clarity: good
  • formatting: excellent
  • grammar: perfect

Author:  Ronak Soni  on 2024-09-25  [id 4802]

(in reply to Report 2 on 2024-08-12)
Category:
answer to question

We thank the reviewer for the detailed suggestions!

Regarding the point about it being unclear where the review ends, we have added a sentence to the beginning of section 2 clarifying this point. Secondly, we agree that the main text is a bit qualitative; this is by design, since (a) we have a lot of ground to cover, so it would get unbearably wrong and (b) our main focus is in explaining why this set of techniques is worthwhile for holographers, so we wanted to foreground these points rather than the technical work (which is not very non-trivial).

  1. We have added this reference.

  2. The reason for the random tensors is simply that this is the only way we know to get a quantum minimal surface formula. We have no additional insight over the original random tensor network paper. We have mentioned this near the end of section 3.1.

  3. Thank you, we will clarify this. In the equation below (3.34), that projection operator is meant to be diagonal in the basis of the irrep ribbons (equation 2.27). These start showing up around (3.15) essentially because we have chosen to insert ''edge modes'' whose entanglement depends on the total flux through the cut, and these ribbon operators are what measure that flux. We have clarified this near (3.15), and also right after (2.30).

  4. Thank you for the suggestion. We have done this.

  5. The contribution from b_2 has been repackaged as the area term <psi|A|psi>. Note that we can do this repackaging into the expectation value of a linear operator because of the special form of the states (3.32).

  6. We agree with the stated connection to reference 30. This relation was in fact our original motivation to take this factorization map seriously.

  7. Yes this is a good point. We make those citations elsewhere but it is well worth mentioning it around (4.2) as well! We have added them.

---

## Round 1 · Referee Report · Anonymous (Referee 3) · 2024-8-28

Strengths

Gives a clear exposition on discretization of topological field theories, which is usually presented in a rather technical way in the literature.

Brings together ideas from different fields, ranging from quantum information, quantum gravity, to topological field theory.

Significant paradigm shift from standard holographic tensor networks.

Weaknesses

While the conceptual analogy to 3d gravity is clear, especially the way in which gauge constraints mimics effects of gravitational constraints, the details of the model are not always well motivated.

Report

This paper proposes a holographic tensor network (most relevant to 3d gravity) that has a number of advantages over existing models. In particular, it captures aspects of the gravitational constraints that are essential for reproducing the bulk gravity theory. The model is a well known discrete realization of a topological gauge theory called Kitaev's quantum double model. While these models are well known in condensed matter physics, this work attempts to draw lessons about gravity by identifying certain defect operators as "area operators". In particular, a novel factorization map is defined that produces an analogue of the holographic entanglement entropy formula associated this notion of area. In this context, the "area" is really measuring the entanglement entropy of edge modes. Another interesting aspect of the factorization map is that it produces multi-partite entangled edge modes. This property is then related to the non commutativity of the area operator for over lapping regions, which is another aspect of bulk gravity that is missed by conventional tensor networks.

The paper addresses some important conceptual problems with holographic tensor networks as they are currently formulated. It also brings together ideas from a diverse set of disciplines, and there is a clear potential for further development. For these reasons I think it should be published. However, there are also a number of drawbacks to the current presentation, and I would request that these be addressed before publication.

1.) First, I think the original contributions of the authors should be more clearly separated from results that already exist in the literature. For example, as far as I understand, the “doubly gauged model" used to produce the tensor network is just the Kitaev quantum double model, restricted to the ground state. The ribbon operators were also defined in Kitaev's work. So is the new ingredient in this paper is really nonlocal factorization map?

2.) Along the same lines, the introduction cites reference 43 as origin of the non-local factorization map. This is supposed to be generalized in this work, but how? For example, did reference 43 define the same subregion algebra and associated center? If so, is the bulk-boundary map the main new ingredient in this work?

3.) One reason I ask about the subregion algebra is that I find it a bit strange that the algebra assigned to a subregion acts on its complement. Can the authors comment on whether there are potential inconsistencies that might arise? Is this property related to the failure of Haag duality in theories with superselection sectors, as described in https://arxiv.org/abs/1905.10487 ?

3.) I find the actual derivation of the holographic entanglement entropy formula quite convoluted: there are many seemingly ad hoc constructions and assumptions that need to be better motivated. First, what is the motivation behind the projection on to random tensors, aside from the fact that it seems to produce the desired answer? Can the authors give a better motivation for the many assumptions and steps used in the replica trick ( are these standard manipulations in the tensor network literature?) At a more basic level, was the replica trick was essential in the first place? Given that the lattice can be heavily reduced, is it out of the question to just diagonalize the density matrix and compute the entanglement EE?

4.) A comment related to the point above is that the EE in the absence of matter is quite simple: formula 3.15 is well known from the gauge theory literature ( a citation is needed here) and seems to be more fundamental: each term has a clear meaning, e.g. the “area” contribution is just the entanglement entropy of edge modes. Isn’t it more natural to view this formula as the fundamental result, and the area operator as an artifice. Indeed, couldn’t we have just defined the area operator to be the operator with eigen value log dim R in the representation basis in the Hilbert space of total fluxes?

I believe the paper should be published after the authors address these points.

Recommendation

Ask for major revision

  • validity: good
  • significance: high
  • originality: high
  • clarity: top
  • formatting: excellent
  • grammar: perfect

Author:  Ronak Soni  on 2024-09-25  [id 4803]

(in reply to Report 3 on 2024-08-28)
Category:
answer to question

We thank the reviewer for the kind comments.

  1. We agree with the criticism that new and old results are not explicitly identified, and have added statements in section 2 differentiating them; most importantly, the sentences just before section 2.1.

  2. We have differentiated our work from [43] in an added sentence after (2.28).

  3. There are two possible things the referee might be referring to here, so let us address both of them.

    The first case where an algebra acts on the complement is that the central ribbon can be deformed into the complement. This is not a problem because the definition of a center is the intersection of two commuting algebras; thus, central operators always act on both a region and its complement. The reason this doesn't violate locality is that the center is a commutative algebra, and so it is merely a statement of classical correlation (as explained in arXiv:1510.07455 by one of the authors).

    It is very much related to the violation of Haag duality in arXiv:1905.10487; this is the point of the "Gauge Theory of Intertwiners" subsection of section 5.

    The second case where something like this happens is in the fused subalgebras of appendix C, where dressings between disconnected bulk subregions cross from one connected component to the other through the complement. Here, we can say for sure that there are no inconsistencies, since the algebra is perfectly well-defined; for example, it is a natural subregion algebra in the reduced lattice. However, it is physically undesirable for modelling holography.

  4. The motivation for introducing the random tensors is the same as in conventional random tensors: it is the best way we know to 'transmit' the local bulk state to the boundary and make a holographic map.

    The need for the replica trick is simply that it is the simplest way to prove the quantum minimal surface formula given that we added random tensors. The fact that the set of matter lollipops included is not fixed by us but by the state is most easily seen in the replica calculation. This last is also the reason that we don't just diagonalise the density matrix by hand; while it is certainly possible for fixed bulk region, the replica trick minimises the entropy over different bulk regions.

    The rest of the assumptions are included for simplicity; they are indeed standard in the random tensor network literature. To be clear, there are interesting contexts in which replica symmetry is broken, but those are orthogonal to the aspects we wish to elucidate.

  5. We have added a reference near (3.15); we thank the referee for pointing out this oversight.

    The referee is completely correct about what the area operator is, and has very much stated our construction in words! We have added some explanation of this after (2.30), where we introduce the central ribbons.

    That said, one important caveat: given a tensor network model, when we compute the entropy S(B) of some boundary region B, we will get an answer S(B) = A + S(bulk), where A is the expectation of some "area" operator. The "hard" part of our construction was figuring out a way to make the "area" operator we get specifically the one we wanted, written below (3.34).

---

## Round 2 · Referee Report · Anonymous (Referee 3) · 2024-10-14

Report

The revisions in this new version addresses the concerns raised before. I am happy to recommend this manuscript for publication now .

Recommendation

Publish (meets expectations and criteria for this Journal)

---

## Round 2 · Referee Report · Anonymous (Referee 1) · 2024-10-15

Report

The authors have made several clarifications and improvements to the paper.
One issue remains unresolved and continues to be a concern: these proposals are based on expectations of a finite rank discrete gauge group, and where no large N limit has to be taken. Therefore, it is still unclear to us why we can take the area term/topological term ratio very seriously as a "problem" to solve. While the reply explains that the boundary is not expected to be gapped, and that some operators in the bulk (the central operator) carries an interpretation in the boundary CFT, that is a very weak reference of the boundary CFT -- for example boundary conditions do not play any role in this computation. The factorization map also seems indifferent to the precise boundary conditions.
Even comparing with the usual semi-classical AdS/CFT, this is odd because the asymptotic boundary conditions do make a huge difference to the computation, without which everything changes, including the Brown-Henneaux asymptotic symmetries of the boundary theory.
Inspecting the wave-function, it is unclear how this tensor network would not describing a gapped state at the boundary rather than a CFT. If we can not be certain of a difference (for example seeing the c/3 log L term showing up), then can we really be certain that the entanglement entropy is given by the gapped bulk alone? The discussion at the end of section 5/ or near equation 4.2 do not seem to resolve this tension or justify why we do not need to worry about the CFT more, or take more care in the large N limit. Without clear justification and reasoning, we find the factorization map overly ad hoc and lacking sufficient grounding to be considered physically meaningful.
For these reasons, while we acknowledge that there is a proposal based on the factorization map that resolves some issues that might be there, it remains inconclusive whether these are real issues, and to what extent the modified tensor network carries more resemblance to a CFT ground state.
Nevertheless, we appreciate the clarifications and improvements, and we defer a decision to the editor.

Recommendation

Reject

---

## Round 2 · Referee Report · Anonymous (Referee 2) · 2024-10-15

Report

The authors have addressed my comments, and I therefore recommend publication.

Recommendation

Publish (easily meets expectations and criteria for this Journal; among top 50%)

---

## Editorial Decision

published